# Transformed Latent Variable Multi-Output Gaussian Processes

**Xiaoyu Jiang** [1]  **Xinxing Shi** [1]  **Sokratia Georgaka** [2]  **Magnus Rattray** [2]  **Mauricio A. Álvarez** [1]

## Abstract

Multi-Output Gaussian Processes (MOGPs) provide a principled probabilistic framework for modelling correlated outputs but face scalability bottlenecks when applied to datasets with high-dimensional output spaces. To maintain tractability, existing methods typically resort to restrictive assumptions, such as employing low-rank or sum-of-separable kernels, which can limit expressiveness. We propose the Transformed Latent Variable MOGP (T-LVMOGP), a novel framework that scales MOGPs to a massive number of outputs while preserving the capacity to capture meaningful inter-output dependencies. T-LVMOGP constructs a flexible multi-output deep kernel by mapping inputs and output-specific latent variables into an embedding space using a Lipschitz-regularised neural network. Combined with stochastic variational inference, our model effectively scales to high-dimensional output settings. Across diverse benchmarks, including climate modelling with over $10,000$ outputs and zero-inflated spatial transcriptomics data, T-LVMOGP outperforms baselines in both predictive accuracy and computational efficiency.

## 1. Introduction

Gaussian Processes (GPs) offer a principled and flexible Bayesian nonparametric framework for learning unknown functions (Williams & Rasmussen, 2006). Multi-Output Gaussian Processes (MOGPs) extend this framework to model correlations between output functions, proving effective across domains ranging from medical time-series prediction (Cheng et al., 2020) to dynamical system modelling (Tang et al., 2025). However, standard MOGP models incur

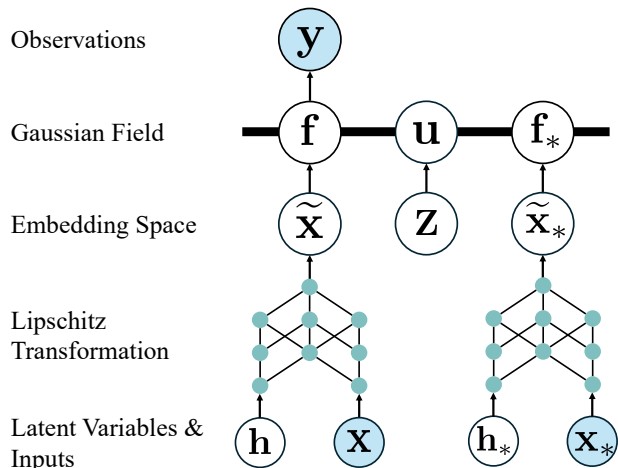

**Figure 1.** Schematic overview of the proposed T-LVMOGP framework. Inducing points **Z** are placed in the embedding space.

a cubic computational cost with respect to the number of outputs $P$. This constitutes a bottleneck for high-dimensional applications such as climate modelling and spatial transcriptomics, where $P$ can easily reach the order of thousands (Bonilla et al., 2007b; Van der Wilk et al., 2020).

To mitigate this complexity, existing research largely focuses on imposing structural constraints, potentially sacrificing expressiveness for scalability. For instance, low-rank approaches assume output functions are linear combinations of shared latent functions (Wackernagel, 2003; Higdon et al., 2008; Xing et al., 2016; Bruinsma et al., 2020). While computationally efficient, these linear assumptions often lack the expressiveness required to model intricate inter-output dependencies. For tensor-based observations, the outputs can be shaped into a multidimensional array so that the Kronecker and tensor algebra are exploited for efficient computation (Zhe et al., 2019; Li et al., 2020). However, this approach is predicated on rigid structural assumptions, which are frequently violated in real-world scenarios. Dai et al. (2017) introduce a latent variable for each output, and construct separable kernels for MOGP. Jiang et al. (2025) generalise this idea to sum-of-separable kernels, and they apply stochastic variational inference techniques (Hoffman et al., 2013; Hensman et al., 2013) to the output space, facilitating better scalability with respect to an increasing number of outputs. However, their expressivity remains bound by

[1]Department of Computer Science, University of Manchester, Manchester, UK [2]Faculty of Biology, Medicine and Health, University of Manchester, Manchester, UK. Correspondence to: Xiaoyu Jiang <xiaoyu.jiang@postgrad.manchester.ac.uk>, Mauricio A. Álvarez <mauricio.alvarezlopez@manchester.ac.uk>.

*Proceedings of the $43^{rd}$ International Conference on Machine Learning*, Seoul, South Korea. PMLR 306, 2026. Copyright 2026 by the author(s).

the fixed structural properties of the kernel class.

In this work, we propose the Transformed Latent Variable MOGP (T-LVMOGP), illustrated in Figure 1, a framework that scales MOGPs to large output dimensions while preserving high expressiveness. To this end, we introduce a multi-output deep kernel that captures intricate correlations among outputs without imposing rigid structural constraints. We associate each output with a latent variable, serving as a compact representation (Bonilla et al., 2007a). To ensure high expressiveness, we employ a neural network to map the concatenated input-latent representations into an embedding space, where correlations are defined (Hinton & Salakhutdinov, 2007; Calandra et al., 2016; Wilson et al., 2016a). This architecture enables seamless integration with existing scalable GP frameworks to handle a large output space. We employ the Sparse Variational GP (SVGP) framework (Titsias, 2009), placing inducing points in the embedding space and performing stochastic variational inference following Hensman et al. (2013). Furthermore, recently proposed tighter variational bounds (Titsias, 2025; Bui et al., 2025) can be readily incorporated into this framework, which potentially enhances predictive performance.

While deep kernels provide the requisite flexibility, naive implementations suffer from overfitting (Ober et al., 2021). In this work, we regularise the neural network via residual connections (He et al., 2016) and spectral normalisation (Miyato et al., 2018). This encourages the transformation to be Lipschitz continuous, and its distance awareness promotes locally smooth and globally stable mappings. This is critical for the reliable performance of the multi-output deep kernel proposed in this work.

**Contributions** This work makes the following contributions: (1) We propose the Transformed Latent Variable MOGP, a scalable framework for building flexible multi-output deep kernels that scale to high-dimensional outputs while preserving rich cross-output dependencies. (2) To improve generalisation, we adopt a Lipschitz continuous neural network using residual connections and spectral normalisation. (3) We develop a stochastic variational inference procedure within the SVGP framework, supporting mini-batch training over both inputs and outputs. Our formulation naturally accommodates non-Gaussian likelihoods and tighter variational bounds. (4) Empirical experiments demonstrate that our model outperforms MOGP baselines in both predictive accuracy and computational efficiency.

## 2. Background

### 2.1. Gaussian Processes

Consider a regression problem with a dataset of $N$ input-output pairs $(\mathbf{X}, \mathbf{y}) = \{(\mathbf{x}_n, y_n)\}_{n=1}^N$, where $\mathbf{X} \in \mathbb{R}^{N \times D_X}$ and $\mathbf{y} \in \mathbb{R}^N$. We assume the outputs are generated by a

latent function $f$ with observation noise, $y_n = f(\mathbf{x}_n) + \epsilon_n$. A zero-mean GP places a prior over the function $f \sim \mathcal{GP}(0, k(\cdot, \cdot'))$. The function values $\mathbf{f} = f(\mathbf{X})$ follow a multivariate Gaussian distribution $p(\mathbf{f}) = \mathcal{N}(\mathbf{f} \mid \mathbf{0}, \mathbf{K_{XX}})$, where $[\mathbf{K_{XX}}]_{ij} = k(\mathbf{x}_i, \mathbf{x}_j)$. Exact GPs incur an $\mathcal{O}(N^3)$ computational cost due to the determinant and inversion computation of an $N \times N$ matrix. To scale to large datasets, SVGP (Titsias, 2009) introduces a set of $M \ll N$ inducing inputs $\mathbf{Z} \in \mathbb{R}^{M \times D_X}$ and corresponding inducing variables $\mathbf{u} = f(\mathbf{Z})$. The joint posterior is approximated using a variational distribution $q(f, \mathbf{u}) = p(f \mid \mathbf{u})q(\mathbf{u})$, where $q(\mathbf{u}) = \mathcal{N}(\mathbf{u} \mid \mathbf{m}, \mathbf{S})$ and $\mathbf{m}, \mathbf{S}$ are variational parameters. We denote $\mathbf{K_{ZZ}} = k(\mathbf{Z}, \mathbf{Z})$ and $\mathbf{K_{XZ}} = k(\mathbf{X}, \mathbf{Z})$. The standard SVGP is trained by maximising the Evidence Lower Bound (ELBO):

$$\mathcal{L}_1 = \sum_{n=1}^N \mathbb{E}_{q(f(\mathbf{x}_n))} \left[ \log p(y_n \mid f(\mathbf{x}_n)) \right] - \mathrm{KL}[q(\mathbf{u}) \| p(\mathbf{u})].$$

Recently, Titsias (2025) and Bui et al. (2025) propose a refined variational posterior that relaxes the prior conditional assumption. They define $q(\mathbf{f}, \mathbf{u}) = q(\mathbf{f} \mid \mathbf{u})q(\mathbf{u})$ with

$$q(\mathbf{f} \mid \mathbf{u}) = \mathcal{N}(\mathbf{f} \mid \mathbf{K_{XZ}}\mathbf{K_{ZZ}}^{-1}\mathbf{u}, \mathbf{D_{XX}}^{1/2}\mathbf{V}\mathbf{D_{XX}}^{\top/2}),$$

where $\mathbf{V}$ is a positive diagonal matrix, $\mathbf{Q_{XX}} = \mathbf{K_{XZ}}\mathbf{K_{ZZ}}^{-1}\mathbf{K_{ZX}}$ represents the Nyström approximation, and $\mathbf{D_{XX}} = \mathbf{K_{XX}} - \mathbf{Q_{XX}}$. This parameterisation leads to a tighter lower bound $\mathcal{L}_2 = \mathcal{L}_1 + \Delta$ after optimising out $\mathbf{V}$, where $\Delta \geq 0$ serves as a correction term. For the Gaussian likelihood with noise variance $\sigma_y^2$, this term is given analytically by:

$$\Delta = \frac{1}{2} \sum_{n=1}^N \left[ \frac{d_n}{\sigma_y^2} - \log\left(1 + \frac{d_n}{\sigma_y^2}\right) \right], \quad (1)$$

where $d_n = [\mathbf{D_{XX}}]_{nn}$ is the $n$-th diagonal element of $\mathbf{D_{XX}}$. Further details are provided in Appendix G.

### 2.2. Multi-Output Gaussian Processes

MOGPs are a natural extension of GPs to vector-valued observations $\mathbf{Y} \in \mathbb{R}^{N \times P}$ where $\mathbf{y}_n \in \mathbb{R}^P$ denotes the observation at input $\mathbf{x}_n$ (Alvarez et al., 2012). A classic approach is the *Linear Model of Coregionalisation* (LMC), which expresses outputs as linear combinations of independent latent GPs (Journel & Huijbregts, 1976). Specifically, the $p$-th output function $f_p$ is modelled as:

$$f_p(\mathbf{x}_n) = \sum_{q=1}^Q \sum_{r=1}^{R_q} \alpha_{p,r}^{(q)} g_r^{(q)}(\mathbf{x}_n), \; y_{n,p} = f_p(\mathbf{x}_n) + \epsilon_{n,p},$$

where $\epsilon_{n,p} \sim \mathcal{N}(0, \sigma_y^2)$, $\{\alpha_{p,r}^{(q)}\}$ are mixing weights and the functions $\{g_r^{(q)}(\mathbf{x})\}_{r=1}^{R_q}$ are latent GPs sharing covariance

function $k_X^{(q)}(\mathbf{x}, \mathbf{x}')$, but sampled independently. The cross-covariance between $f_p(\mathbf{x})$ and $f_{p'}(\mathbf{x}')$ is given by

$$\mathrm{cov}[f_p(\mathbf{x}), f_{p'}(\mathbf{x}')] = \sum_{q=1}^{Q} c_{p,p'}^{(q)} k_X^{(q)}(\mathbf{x}, \mathbf{x}'),$$

with $c_{p,p'}^{(q)} = \sum_{r=1}^{R_q} \alpha_{p,r}^{(q)} \alpha_{p',r}^{(q)}$. $\mathbf{K}_{\mathbf{XX}}^{(q)} = k_X^{(q)}(\mathbf{X}, \mathbf{X})$ is from the $q$-th input kernel. Let $\mathbf{f} = [\mathbf{f}_1^\top, \ldots, \mathbf{f}_P^\top]^\top \in \mathbb{R}^{PN}$ be the vectorised latent function values with $\mathbf{f}_p = f_p(\mathbf{X})$. The covariance matrix for $\mathbf{f}$ is

$$\mathrm{cov}[\mathbf{f}, \mathbf{f}] = \sum_{q=1}^{Q} \mathbf{A}^{(q)} \mathbf{A}^{(q)\top} \otimes \mathbf{K}_{\mathbf{XX}}^{(q)} = \sum_{q=1}^{Q} \mathbf{C}^{(q)} \otimes \mathbf{K}_{\mathbf{XX}}^{(q)},$$

where $\otimes$ denotes the Kronecker product, $\mathbf{A}^{(q)} \in \mathbb{R}^{P \times R_q}$ collects $\alpha_{p,r}^{(q)}$, and the *coregionalisation matrix* $\mathbf{C}^{(q)} \in \mathbb{R}^{P \times P}$ contains $c_{p,p'}^{(q)}$. Notably, LMC implies that $\mathbf{C}^{(q)}$ is a low-rank matrix derived from a *linear kernel* applied to the latent output embeddings (the rows of $\mathbf{A}^{(q)}$).

The Latent Variable MOGP (LV-MOGP) (Dai et al., 2017; Jiang et al., 2025) generalises this framework by assigning latent variables to each output and constructing $\mathbf{C}^{(q)}$ by applying any valid kernel function to them. This formulation enhances the model's capacity to capture more complex output correlations. Let $\mathbf{H} = \{\mathbf{H}^{(q)}\}_{q=1}^{Q}$ denote the collection of latent variables, where $\mathbf{H}^{(q)} \in \mathbb{R}^{P \times D_H}$ contains the latent vector $\mathbf{h}_{p,q} \in \mathbb{R}^{D_H}$ for the $p$-th output in the $q$-th group. We have $\mathbf{C}^{(q)} = \mathbf{K}_{\mathbf{H}^{(q)}\mathbf{H}^{(q)}}^{(q)} = k_H^{(q)}(\mathbf{H}^{(q)}, \mathbf{H}^{(q)})$, where $k_H^{(q)}$ denotes the $q$-th kernel function on the latent space. Under a Bayesian framework, generative distributions are

$$p(\mathbf{h}_{p,q}) = \mathcal{N}(\mathbf{h}_{p,q} \mid \mathbf{0}, \mathbf{I}),$$

$$p(\mathbf{f} \mid \mathbf{X}, \mathbf{H}) = \mathcal{N}\left(\mathbf{f} \mid \mathbf{0}, \sum_{q=1}^{Q} \mathbf{K}_{\mathbf{H}^{(q)}\mathbf{H}^{(q)}}^{(q)} \otimes \mathbf{K}_{\mathbf{XX}}^{(q)}\right),$$

$$p(\mathrm{vec}(\mathbf{Y}) \mid \mathbf{f}) = \mathcal{N}(\mathrm{vec}(\mathbf{Y}) \mid \mathbf{f}, \sigma_y^2 \mathbf{I}),$$

where $\mathrm{vec}(\cdot)$ denotes the vectorisation operation. Scalable inference is achieved via a sparse variational method that places inducing points in both the input and latent spaces. For latent variables, LV-MOGP introduces variational distributions $q(\mathbf{H})$; unlike point estimation, this Bayesian treatment mitigates the risk of overfitting (Titsias & Lawrence, 2010; Lalchand et al., 2022).

### 2.3. Deep Kernels and Lipschitz Regularisation

Deep Kernel Learning (Wilson et al., 2016a;b) unifies the representation power of neural networks with GPs by defining a kernel over learned embeddings. However, unconstrained neural networks can lead to pathological behaviour, such as feature collapse and the loss of distance awareness (Liu et al., 2020; Ober et al., 2021). This degradation

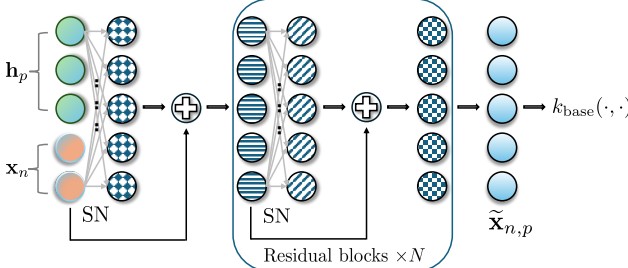

**Figure 2.** Illustration of the proposed multi-output deep kernel with Lipschitz-regularised residual connected neural network (RCNN). SN: spectral normalisation.

undermines the inherent benefits of GPs for quantifying uncertainty, leading to overconfident predictions on new data points (Ober et al., 2021; Van Amersfoort et al., 2021). To mitigate this, Van Amersfoort et al. (2021) propose enforcing a Lipschitz constraint on the neural network, recovering the principled uncertainty of deep kernel GPs. In this work, we leverage these Lipschitz-regularised deep kernels with appropriate expressiveness and extend them to LV-MOGPs.

## 3. Transformed Latent Variable MOGP

Building on LV-MOGP, our approach introduces a novel, expressive multi-output deep kernel that explicitly transforms complex MOGPs into more tractable scalar GPs in a learned embedding space. Formulated within the SVGP framework (Hensman et al., 2013), our model scales to datasets with high-dimensional outputs while effectively capturing essential inter-output correlations.

### 3.1. Model Definition

Let $\mathbf{H} = \{\mathbf{h}_p\}_{p=1}^{P} \in \mathbb{R}^{P \times D_H}$ represent latent variables assigned to every output. In MOGP, the central modelling choice is the specification of the cross-output covariances, i.e. the construction of $k_{p,p'}(\mathbf{x}, \mathbf{x}')$ for $p, p' \in \{1, ..., P\}, \mathbf{x}, \mathbf{x}' \in \mathbf{X}$. We build it in two stages: (1) map input and latent variable pairs to an embedding space, and (2) define covariances via a base kernel on that space.

Concretely, we introduce a neural network $\Phi_\theta : \mathbb{R}^{D_X} \times \mathbb{R}^{D_H} \to \mathbb{R}^{D_T}$ that takes the concatenated pair $(\mathbf{x}_n, \mathbf{h}_p)$ and returns an embedding $\widetilde{\mathbf{x}}_{n,p} \in \mathbb{R}^{D_T}$, where $D_T$ is the dimensionality of the embedding space. Given a stationary kernel $k_{\mathrm{base}}(\cdot, \cdot)$ on the embedding space (e.g., the RBF kernel), we define the cross-covariances as

$$\mathrm{cov}[f_p(\mathbf{x}_n), f_{p'}(\mathbf{x}_{n'})] = k_{\mathrm{base}}(\widetilde{\mathbf{x}}_{n,p}, \widetilde{\mathbf{x}}_{n',p'}).$$

Figure 2 depicts the construction of our kernel. Notably, we do not impose restrictive assumptions, such as low-rank or the sum-of-separable structures, on our kernel. Instead, correlations are induced implicitly through a learned embed-

ding and a base kernel. [1] As demonstrated in our experiments, this approach enables greater capacity in capturing intricate correlations among high-dimensional outputs. To mitigate the overfitting issues common in deep kernels (Ober et al., 2021; Matias et al., 2024), we parameterise $\Phi_\theta$ as a Residual Connected Neural Network (RCNN) (He et al., 2016) and apply spectral normalisation (Miyato et al., 2018) to the weight matrices of each layer. Specifically, we adopt the power iteration algorithm (Golub & Van Loan, 2013), as outlined in Algorithm 1 in Appendix C. By setting a proper Spectral Norm Upper Bound (SN-UB) of the weight matrices, $\Phi_\theta$ is Lipschitz continuous with the desired Lipschitz constant $\left(\text{that is } (1 + \text{SN-UB})^L \text{ for } L \text{ layer network}\right)$ (Liu et al., 2020), which is demonstrated to be significant for achieving improved generalisation ability in our experiments. Moreover, this parameterisation offers the expressive capacity to model a wide range of smooth Lipschitz mappings, suggesting the versatility of this approach (Bartlett et al., 2018). Further details regarding Lipschitz continuity and spectral normalisation can be found in Appendix C.

### 3.2. Variational Inference

We introduce prior and variational distributions for the latent variables $\mathbf{H}$ across outputs:

$$p(\mathbf{H}) = \prod_{p=1}^{P} p(\mathbf{h}_p) = \prod_{p=1}^{P} \mathcal{N}(\mathbf{h}_p \mid 0, \mathbf{I}),$$

$$q(\mathbf{H}) = \prod_{p=1}^{P} q(\mathbf{h}_p) = \prod_{p=1}^{P} \mathcal{N}(\mathbf{h}_p \mid \mathbf{m}_p, \boldsymbol{\Sigma}_p).$$

Here, $\mathbf{m}_p \in \mathbb{R}^{D_H}$ and $\boldsymbol{\Sigma}_p \in \mathbb{R}^{D_H \times D_H}$ are variational parameters. Following Hensman et al. (2013), we introduce inducing points $\mathbf{Z} \in \mathbb{R}^{M \times D_T}$ in the embedding space. The variational distribution of inducing variables $\mathbf{u}$ is $q(\mathbf{u}) = \mathcal{N}(\mathbf{u} \mid \mathbf{m}, \mathbf{S})$. Given input $\mathbf{x}_n$ and a sample of the latent variable $\mathbf{h}_p$, the embedding is computed as $\widetilde{\mathbf{x}}_{n,p} = \Phi_\theta(\mathbf{x}_n, \mathbf{h}_p)$. The variational distribution for $f_p(\mathbf{x}_n)$ can be computed as

$$q(f_p(\mathbf{x}_n) \mid \mathbf{x}_n, \mathbf{h}_p) = \int q(\mathbf{u}) p(f_p(\mathbf{x}_n) \mid \mathbf{u}) \, d\mathbf{u}$$

$$= \mathcal{N}(f_p(\mathbf{x}_n) \mid \mu_{n,p}, \sigma_{n,p}^2), \quad (2)$$

where $\mu_{n,p} = \mathbf{k}_{\widetilde{\mathbf{x}}_{n,p}}^\top \mathbf{K}_{\mathbf{ZZ}}^{-1} \mathbf{m}$ and $\sigma_{n,p}^2 = k_{\widetilde{\mathbf{x}}_{n,p}} - \mathbf{k}_{\widetilde{\mathbf{x}}_{n,p}}^\top \mathbf{K}_{\mathbf{ZZ}}^{-1} \mathbf{k}_{\widetilde{\mathbf{x}}_{n,p}} + \mathbf{k}_{\widetilde{\mathbf{x}}_{n,p}}^\top \mathbf{K}_{\mathbf{ZZ}}^{-1} \mathbf{S} \mathbf{K}_{\mathbf{ZZ}}^{-1} \mathbf{k}_{\widetilde{\mathbf{x}}_{n,p}}$. Here, $\mathbf{k}_{\widetilde{\mathbf{x}}_{n,p}} = k_{\text{base}}(\mathbf{Z}, \widetilde{\mathbf{x}}_{n,p}) \in \mathbb{R}^M$, $\mathbf{K}_{\mathbf{ZZ}} = k_{\text{base}}(\mathbf{Z}, \mathbf{Z}) \in \mathbb{R}^{M \times M}$ and $k_{\widetilde{\mathbf{x}}_{n,p}} = k_{\text{base}}(\widetilde{\mathbf{x}}_{n,p}, \widetilde{\mathbf{x}}_{n,p}) \in \mathbb{R}$. Consider $y_{n,p} \sim p(y_{n,p} \mid$

---

[1]Our kernel generalises the separable and sum-of-separable structures employed in existing LV-MOGP models, as detailed in Appendix D.

$f_p(\mathbf{x}_n))$ as the likelihood of the data; the ELBO is

$$\mathcal{L}_3 = \sum_{n=1}^{N} \sum_{p=1}^{P} \mathbb{E}_{q(\mathbf{h}_p)q(f_p(\mathbf{x}_n)|\mathbf{x}_n,\mathbf{h}_p)} \left[\log p(y_{n,p} \mid f_p(\mathbf{x}_n))\right]$$

$$- \text{KL}[q(\mathbf{u}) \,||\, p(\mathbf{u})] - \sum_{p=1}^{P} \text{KL}[q(\mathbf{h}_p) \,||\, p(\mathbf{h}_p)].$$

The derivation of the ELBO is provided in Appendix B. We compute the inner expectation $\mathcal{V}_{n,p}(\mathbf{h}_p) = \mathbb{E}_{q(f_p(\mathbf{x}_n)|\mathbf{x}_n,\mathbf{h}_p)}[\log p(y_{n,p} \mid f_p(\mathbf{x}_n))]$ based on the likelihood type. For the Gaussian likelihood, this term is available analytically. For non-Gaussian likelihoods, we approximate it via Gauss-Hermite quadrature or Monte Carlo sampling (Ramchandran et al., 2021). To handle the outer expectation over $q(\mathbf{h}_p)$, we employ the reparametrisation trick (Kingma & Welling, 2014; Lalchand et al., 2022). Specifically, we sample $\mathbf{h}_p^{(j)} = \mathbf{m}_p + \boldsymbol{\Sigma}_p^{1/2}\boldsymbol{\epsilon}^{(j)}$ with $\boldsymbol{\epsilon}^{(j)} \sim \mathcal{N}(0, \mathbf{I})$. The expected log-likelihood is then approximated as

$$\mathbb{E}_{q(\mathbf{h}_p)}[\mathcal{V}_{n,p}(\mathbf{h}_p)] \approx \frac{1}{J} \sum_{j=1}^{J} \mathcal{V}_{n,p}(\mathbf{h}_p^{(j)}).$$

For the Gaussian likelihood, we obtain a tighter variational bound by adding $NP$ correction terms defined as in Equation (1) to $\mathcal{L}_3$.

**Scalable Optimisation** To scale to datasets with a large number of inputs and outputs, we employ stochastic mini-batch training. At each iteration, we sample input and output mini-batches uniformly, denoted as $\mathcal{B}_N$ and $\mathcal{B}_P$ with $|\mathcal{B}_N| = N_b$ and $|\mathcal{B}_P| = P_b$. The ELBO is estimated as

$$\widetilde{\mathcal{L}}_3 = \frac{NP}{N_b P_b} \sum_{n \in \mathcal{B}_N} \sum_{p \in \mathcal{B}_P} \mathbb{E}_{q(\mathbf{h}_p)}[\mathcal{V}_{n,p}(\mathbf{h}_p)]$$

$$- \text{KL}[q(\mathbf{u}) \,||\, p(\mathbf{u})] - \frac{P}{P_b} \sum_{p \in \mathcal{B}_P} \text{KL}[q(\mathbf{h}_p) \,||\, p(\mathbf{h}_p)].$$

### 3.3. Predictive Posterior

For a new input $\mathbf{x}_*$ and a target output dimension $p_* \in \{1, 2, ..., P\}$, the predictive distribution of $f_{p_*}(\mathbf{x}_*)$ is

$$q(f_{p_*}(\mathbf{x}_*)) = \int q(\mathbf{h}_{p_*}) q(f_{p_*}(\mathbf{x}_*) \mid \mathbf{x}_*, \mathbf{h}_{p_*}) \, d\mathbf{h}_{p_*}. \quad (3)$$

Since this integration is analytically intractable, we employ a Monte Carlo approximation. We draw $S$ samples from $q(\mathbf{h}_{p_*})$, approximating Equation (3) as a mixture of Gaussian distributions:

$$q(f_{p_*}(\mathbf{x}_*)) \approx \frac{1}{S} \sum_{s=1}^{S} \mathcal{N}(f_{p_*}(\mathbf{x}_*) \mid \mu_*^{(s)}, \sigma_*^{2(s)}),$$

$\mu_*^{(s)}$ and $\sigma_*^{2(s)}$ are computed according to Equation (2). If we are interested in the predictive distribution of observation $\widetilde{y}_*$, we further convolve $q(f_{p_*}(\mathbf{x}_*))$ with the likelihood.

## 3.4. Computational Complexity

The computational complexity of our model stems from two components: the SVGP computations, which scale as $\mathcal{O}(N_b P_b M^2 + M^3)$, and the spectral normalisation operations. Regarding the latter, enforcing Lipschitz constraints via power iteration incurs $\mathcal{O}(Tmn)$ for an $m \times n$ matrix over $T$ iterations. However, given the relatively small width ($\sim 10$) and depth ($\sim 5$) of the RCNNs used in our experiments, the computational overhead of spectral normalisation is often negligible compared to the SVGP computations.

## 4. Related Work

A wide class of MOGP models have been proposed in the literature. Convolved GP framework (Boyle & Frean, 2004; Alvarez & Lawrence, 2008; Alvarez et al., 2012) employs convolution integrals between smoothing kernels and latent GPs to generate dependent outputs. For isotopic data, the Kronecker product structure between the output and input covariance matrices can be exploited for efficient computation (Bonilla et al., 2007b; Stegle et al., 2011; Rakitsch et al., 2013). In a different vein, Leroy et al. (2023) shift the focus from covariance to mean functions, modelling multiple tasks through clustering and latent mean processes. Dai et al. (2024) propose the Graphical MOGP (G-MOGP), which incorporates attention-based graphical models to construct more expressive multi-output priors. Moreno-Muñoz et al. (2018) extend LMC to heterogeneous prediction tasks, where each output is associated with a (possibly) different likelihood function. Liu et al. (2022) enhance the LMC by incorporating neural embeddings. By projecting latent GPs into a higher-dimensional feature space, they claim this approach augments the model's representational capacity. Jeong & Kim (2023) represent the latent GPs in the LMC as factorial stochastic differential equations for efficiently modelling temporal data. More recently, Rooijakkers et al. (2025) investigate the sensitivity of MOGPs to misspecification and outliers, proposing robust and conjugate MOGPs.

Despite these developments, current methods do not adequately address scalability for high-dimensional output spaces. Bridging this gap is the primary focus of this paper. Further related work is provided in Appendix H.

## 5. Experiments

In this section, we evaluate the proposed T-LVMOGP on several real-world datasets. We benchmark our method against various MOGP baselines, focusing on scalable models such as SV-LMC (Van der Wilk et al., 2020), SGPRN (Li et al., 2020), OILMM (Bruinsma et al., 2020), G-MOGP (Dai et al., 2024), and GS-LVMOGP (Jiang et al., 2025). When applicable, Ind-GP (Hensman et al., 2013) is also included for comparison. We provide more details about baseline models in Appendix I, Appendix J and Appendix K. We adopt the ARD-RBF kernel for all models. Performance is assessed using Mean Squared Error (MSE) and Negative Log-Likelihood (NLL) on held-out test data. The wall-clock training time is used to estimate computational efficiency. All reported results represent the mean and standard deviation across 10 random trials, and the best mean values are highlighted in the tables below. A list of acronyms can be found in Appendix A, and further experimental setups are described in Appendix F.

### 5.1. Electroencephalogram (EEG) Prediction

This dataset consists of 256 voltage measurements recorded from 7 scalp electrodes during a visual stimulus experiment (Zhang et al., 1995). Following the experimental protocol of Requeima et al. (2019) and Bruinsma et al. (2020), the task is to predict the final 100 time steps for three target electrodes, conditioned on the remaining observed data.

**Table 1.** Performance of different models on the EEG dataset. The $\downarrow$ symbol indicates that lower values are better.

| Model | MSE $\downarrow$ | NLL $\downarrow$ |
|---|---|---|
| *Baselines* | | |
| Ind-GP | $0.466 \pm 0.000$ | $1.227 \pm 0.000$ |
| SGPRN | $0.473 \pm 0.003$ | $7.105 \pm 2.110$ |
| G-MOGP | $0.604 \pm 0.099$ | $1.337 \pm 0.038$ |
| OILMM | $0.372 \pm 0.098$ | $0.979 \pm 0.159$ |
| GS-LVMOGP | $0.366 \pm 0.064$ | $0.924 \pm 0.081$ |
| SV-LMC | $0.282 \pm 0.193$ | $0.857 \pm 0.613$ |
| T-LVMOGP (ours) | $\mathbf{0.115 \pm 0.025}$ | $\mathbf{0.814 \pm 0.310}$ |

As presented in Table 1, our model consistently surpasses the baseline methods in terms of both MSE and NLL. A visualisation of the predictions for a selected electrode is shown in Figure 3. We investigate the impact of spectral normalisation on T-LVMOGP by varying the SN-UB of each layer. As summarised in Table 2, predictive performance peaks at an intermediate SN-UB value, with degradation observed when deviating from this optimum. This pattern highlights the trade-off in regulating network expressiveness: neural networks with relaxed spectral constraints (high SN-UB) do not necessarily yield better generalisation, whereas overly restrictive ones (low SN-UB) lack the capacity to capture essential correlations for making accurate predictions.

**Table 2.** Performance of T-LVMOGP with different SN-UB values on the EEG experiment.

| SN-UB | MSE $\downarrow$ | NLL $\downarrow$ |
|---|---|---|
| 0.1 | $0.128 \pm 0.087$ | $1.339 \pm 2.065$ |
| 0.005 | $\mathbf{0.115 \pm 0.025}$ | $\mathbf{0.814 \pm 0.310}$ |
| 0.001 | $0.145 \pm 0.053$ | $1.371 \pm 1.120$ |

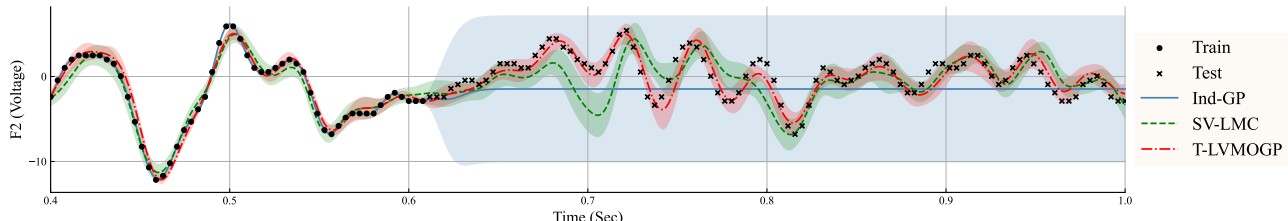

**Figure 3.** Predictions of Ind-GP, SV-LMC and T-LVMOGP for F2 electrode with 95% confidence region in the EEG experiment.

**Table 3.** Performance comparison of different models on the SARCOS dataset.

| Model | MSE ↓ | NLL ↓ | Training Time (s/epoch) ↓ |
|---|---|---|---|
| *Baselines* | | | |
| OILMM (Bruinsma et al., 2020) | $0.140 \pm 0.010$ | $0.864 \pm 0.090$ | $8.951 \pm 0.070$ |
| GS-LVMOGP (Jiang et al., 2025) | $0.037 \pm 0.001$ | $-0.220 \pm 0.011$ | $10.241 \pm 0.096$ |
| SV-LMC (Van der Wilk et al., 2020) | $0.033 \pm 0.000$ | $-0.297 \pm 0.003$ | $6.185 \pm 0.086$ |
| G-MOGP (Dai et al., 2024) | $0.023 \pm 0.001$ | $-0.483 \pm 0.017$ | $5.892 \pm 0.084$ |
| T-LVMOGP (ours) | $\mathbf{0.022 \pm 0.000}$ | $\mathbf{-0.485 \pm 0.009}$ | $\mathbf{5.263 \pm 0.194}$ |

## 5.2. Inverse Dynamics Prediction

In this experiment, we investigate the prediction of inverse dynamics for a SARCOS anthropomorphic robot arm. The dataset comprises $N = 48,933$ samples with $P = 7$ and $D_X = 21$. The $NP$ observations are randomly partitioned into training and testing sets using a 50% split, yielding $171,266$ training observations and $171,265$ test points.

Table 3 reports the predictive performance of the baseline methods and our approach. Note that Ind-GP and SGPRN are excluded as they are computationally prohibitive given the large sample size $N$. Across MSE, NLL, and wall-clock training time, our method consistently outperforms OILMM, SV-LMC, and GS-LVMOGP. G-MOGP performs slightly worse than our model but incurs a longer training time. Therefore, our approach maintains high expressiveness without compromising computational scalability.

**Table 4.** Performance of T-LVMOGP with different SN-UB values on the SARCOS experiment.

| SN-UB | MSE ↓ | NLL ↓ |
|---|---|---|
| 1.5 | $0.023 \pm 0.001$ | $-0.470 \pm 0.016$ |
| 1.0 | $\mathbf{0.022 \pm 0.000}$ | $\mathbf{-0.485 \pm 0.009}$ |
| 0.1 | $0.028 \pm 0.000$ | $-0.363 \pm 0.005$ |

We evaluate several per-layer SN-UB choices in this experiment. As shown in Table 4, predictive performance is maximised at a moderate SN-UB value, diminishing when the spectral norm upper bound is either tightened or relaxed. This confirms that optimal generalisation requires carefully calibrating the trade-off between model capacity and regularisation strength.

## 5.3. Climate Modelling

In this section, we consider modelling monthly temperatures at different spatial locations (treated as outputs) on two datasets: ERA5 and Copernicus Marine.

**ERA5** The ERA5 dataset (Copernicus Climate Change Service, 2024) combines model data with global observations via data assimilation. In this study, we focus on the monthly $2\,\mathrm{m}$ air temperature over the United Kingdom. Our dataset comprises $3,395$ spatial locations spanning a 30-month period from June 2023 to November 2025. All temperature data are partitioned into training (70%) and test (30%) sets. The task is to predict the temperature on the test split given the train split. We evaluate using two distinct splitting mechanisms: random and block-wise. For clarity, we present the random split results in this section, while the block-wise split results are detailed in Appendix F.4.1.

Table 5 provides the experimental results. As shown in the table, our model surpasses the baselines in both MSE and NLL. This result is further validated by Figure 4. Regarding computational efficiency, our method requires less training time than SGPRN, OILMM, and GS-LVMOGP. While G-MOGP and SV-LMC exhibit lower training times than T-LVMOGP, our model outperforms them significantly in predictive performance, measured from MSE and NLL. Consequently, our model achieves the most favourable trade-off between model expressiveness and computational efficiency.

**Copernicus Marine** In this section, we utilise sea water potential temperature data acquired from the Copernicus Marine Service (Mercator Ocean International, 2024). For this study, we extracted monthly-averaged temperature fields for the marine region surrounding the United Kingdom, covering a 24-month period from November 2023 to October

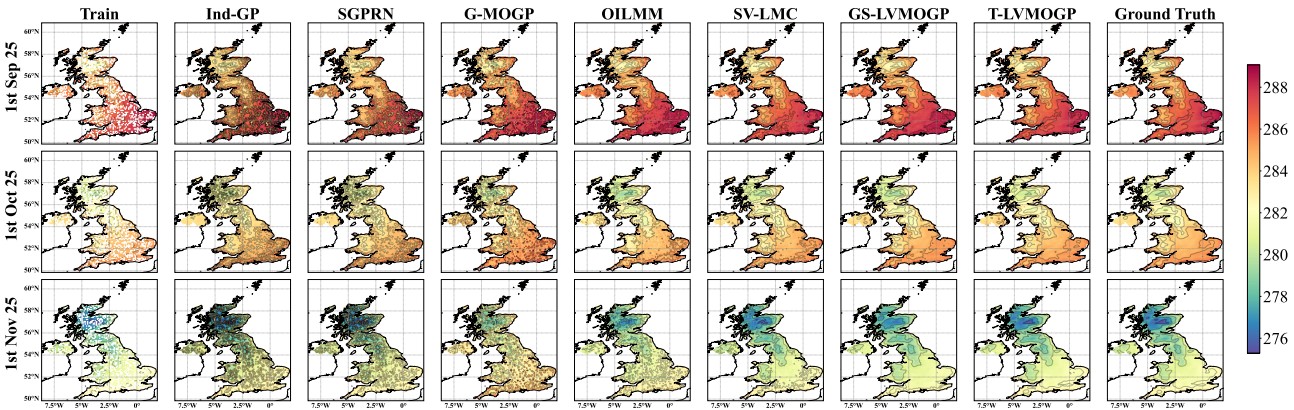

**Figure 4.** Predictions of different models on the ERA5 dataset with random splitting. The temperature is measured in Kelvin units.

**Table 5.** Performance comparison of different models on the ERA5 dataset with random splitting.

| Model | MSE ↓ | NLL ↓ | Training Time (s/epoch) ↓ |
|---|---|---|---|
| *Baselines* | | | |
| Ind-GP (Hensman et al., 2013) | $0.997 \pm 0.000$ | $1.418 \pm 0.000$ | $31.518 \pm 0.366$ |
| SGPRN (Li et al., 2020) | $0.897 \pm 0.124$ | $51.005 \pm 7.411$ | $0.780 \pm 0.001$ |
| G-MOGP (Dai et al., 2024) | $0.316 \pm 0.035$ | $0.910 \pm 0.058$ | $0.685 \pm 0.018$ |
| OILMM (Bruinsma et al., 2020) | $0.123 \pm 0.052$ | $0.483 \pm 0.403$ | $2.332 \pm 0.001$ |
| SV-LMC (Van der Wilk et al., 2020) | $0.012 \pm 0.004$ | $0.758 \pm 0.428$ | $\mathbf{0.570 \pm 0.015}$ |
| GS-LVMOGP (Jiang et al., 2025) | $0.014 \pm 0.003$ | $-0.699 \pm 0.084$ | $1.323 \pm 0.028$ |
| T-LVMOGP (ours) | $\mathbf{0.002 \pm 0.000}$ | $\mathbf{-1.564 \pm 0.024}$ | $0.708 \pm 0.013$ |

2025. Distinct from the previous experiment, we consider the extrapolation-of-outputs setting as in (Jiang et al., 2025), where spatial coordinates serve as the prior mean for the latent variables. The model is trained on a subset of spatial locations to generate predictions for the entire 24-month period at new locations. This dataset contains $21,679$ outputs, with $10,873$ observed and $10,806$ held out for test.

We consider GS-LVMOGP as the baseline for this experiment, and Table 6 compares their predictive performance on this task. The prediction plots are shown in Figure 8 in Appendix F.4.2. For the baseline, we evaluate the number of coregionalisation matrices ranging from $Q = 1$ to 3. Consistent with findings in (Jiang et al., 2025), $Q = 3$ yields the best performance; however, it remains inferior to our proposed method. Furthermore, T-LVMOGP demonstrates superior computational efficiency, achieving lower training times compared to the baselines.

### 5.4. Spatial Transcriptomics

Spatial transcriptomics enables high-resolution gene expression profiling while simultaneously preserving the spatial context of tissue samples (Ståhl et al., 2016). In this experiment, we consider a 10x Genomics Visium human prostate cancer dataset (10XGenomics2025), comprising gene expression counts for $5,000$ genes across $4,352$ spatial lo-

cations. As illustrated in Figure 10 in Appendix F.5, the data is characterised by high sparsity and significant overdispersion. To accommodate these characteristics, we employ the Zero-Inflated Negative Binomial (ZINB) likelihood following BinTayyash et al. (2021). Details regarding the ZINB likelihood are provided in Appendix E. By formulating spatial spots as inputs and genes as outputs, we obtain a total of $21,760,000$ observations. We implement a random $90\%/10\%$ train-test split to benchmark T-LVMOGP against GS-LVMOGP, both utilising the ZINB likelihood.

The experimental results are summarised in Table 7. Our model outperforms GS-LVMOGP with $Q = 1, 2, 3$ in terms of MSE, and achieves comparable NLL values to GS-LVMOGP with $Q = 2, 3$. Although GS-LVMOGP with $Q = 1$ requires the lowest training time, our model surpasses it in terms of both MSE and NLL.

### 5.5. Ablation Study and Analysis

**Impact of Spectral Normalisation** We present a performance comparison between the proposed T-LVMOGP with and without spectral normalisation in Table 8 and Table 23 in Appendix F.6. As shown in Table 8, the model incorporating spectral normalisation consistently outperforms the variant without it, validating the necessity of this component for our framework.

*Table 6.* Performance comparison on the Copernicus Marine dataset for the output extrapolation task.

| Metric | GS-LVMOGP (Jiang et al., 2025) | | | T-LVMOGP (ours) |
|---|---|---|---|---|
| | $Q$=1 | $Q$=2 | $Q$=3 | RCNN |
| **MSE** ↓ | $0.040 \pm 0.002$ | $0.036 \pm 0.001$ | $0.035 \pm 0.002$ | $\mathbf{0.029 \pm 0.000}$ |
| **NLL** ↓ | $5.230 \pm 0.640$ | $5.086 \pm 0.297$ | $4.975 \pm 0.923$ | $\mathbf{-0.439 \pm 0.011}$ |
| **Training Time (s/epoch)** ↓ | $1.321 \pm 0.043$ | $1.750 \pm 0.045$ | $2.076 \pm 0.048$ | $\mathbf{1.230 \pm 0.012}$ |

*Table 7.* Predictive performance of GS-LVMOGP and T-LVMOGP on a Spatial Transcriptomics dataset.

| Metric | GS-LVMOGP (Jiang et al., 2025) | | | T-LVMOGP (ours) |
|---|---|---|---|---|
| | $Q$=1 | $Q$=2 | $Q$=3 | RCNN |
| **MSE** ↓ | $11.616 \pm 0.773$ | $11.396 \pm 0.314$ | $11.024 \pm 0.350$ | $\mathbf{9.189 \pm 0.342}$ |
| **NLL** ↓ | $0.677 \pm 0.002$ | $0.675 \pm 0.001$ | $\mathbf{0.674 \pm 0.001}$ | $\mathbf{0.674 \pm 0.001}$ |
| **Training Time (s/epoch)** ↓ | $\mathbf{65.507 \pm 0.820}$ | $73.639 \pm 0.505$ | $78.245 \pm 1.131$ | $73.047 \pm 0.318$ |

*Table 8.* Test NLL comparison of T-LVMOGP with and without spectral normalisation.

| Dataset | w/o SN | w/ SN |
|---|---|---|
| EEG | $4.109 \pm 2.141$ | $\mathbf{0.814 \pm 0.310}$ |
| SARCOS | $0.112 \pm 0.856$ | $\mathbf{-0.485 \pm 0.01}$ |
| ERA5 (random) | $-1.401 \pm 0.13$ | $\mathbf{-1.564 \pm 0.03}$ |
| ERA5 (block-wise) | $-0.895 \pm 0.85$ | $\mathbf{-1.503 \pm 0.03}$ |
| Copernicus Marine | $-0.400 \pm 0.08$ | $\mathbf{-0.439 \pm 0.01}$ |

**Impact of Neural Network** The T-LVMOGP framework does not strictly require the neural network component; replacing it with an identity mapping preserves the model's validity as an MOGP. However, the results presented in Table 9 and Table 24 in Appendix F.6 indicate that incorporating neural networks yields performance that matches or exceeds that of the identity mapping. This validates the benefits of integrating neural networks into our framework.

*Table 9.* Test NLL comparison of T-LVMOGP with and without neural network architecture.

| Dataset | w/o NN | w/ NN |
|---|---|---|
| EEG | $1.153 \pm 0.437$ | $\mathbf{0.814 \pm 0.310}$ |
| SARCOS | $-0.336 \pm 0.01$ | $\mathbf{-0.485 \pm 0.01}$ |
| ERA5 (random) | $-1.554 \pm 0.03$ | $\mathbf{-1.564 \pm 0.03}$ |
| ERA5 (block-wise) | $-1.474 \pm 0.06$ | $\mathbf{-1.503 \pm 0.03}$ |
| Copernicus Marine | $-0.420 \pm 0.01$ | $\mathbf{-0.439 \pm 0.01}$ |

### 5.6. Optimisation with Tighter Variational Bounds

As shown in Titsias (2025) and Bui et al. (2025), tighter variational bounds yield performance gains for single-output GP. Our framework naturally extends this advantage to multi-output settings. As reported in Table 10 and Table 27 in Appendix F.7, optimising our model with tighter variational bounds results in performance that is comparable to or better

than the standard bound. These findings empirically validate the effectiveness of employing tighter variational bounds for multi-output learning within our T-LVMOGP framework.

*Table 10.* Test NLL comparison of T-LVMOGP with standard and tighter variational bounds. †: for non-Gaussian likelihood, tighter bounds might not be effectively tighter in practice; more explanations are provided in Appendix F.7.

| Dataset | Standard | Tighter |
|---|---|---|
| SARCOS | $-0.485 \pm 0.01$ | $\mathbf{-0.502 \pm 0.01}$ |
| Copernicus Marine | $-0.439 \pm 0.01$ | $\mathbf{-0.443 \pm 0.01}$ |
| Spatial Transcriptomics† | $0.674 \pm 0.001$ | $0.674 \pm 0.002$ |

## 6. Conclusion

In this work, we introduce a new MOGP framework that alleviates the computational bottleneck induced by high-dimensional outputs while retaining sufficient modelling capacity to capture rich inter-output dependencies. Central to our approach is a Lipschitz-regularised multi-output deep kernel that reformulates MOGPs as scalar GPs defined on a learned embedding space. This transformation enables a direct reuse of scalable inference methods developed for single-output GPs, thereby yielding scalability with respect to the number of outputs. We instantiate the proposed model within the SVGP framework, considering both standard and tighter variational bounds. Future work includes integrating other scalability techniques like structured kernel interpolation (Wilson & Nickisch, 2015) and nearest-neighbour approximations (Tran et al., 2021; Wu et al., 2022).

Our model relies on a factorised variational distribution for latent variables, which scales linearly with the number of outputs but limits the capture of posterior output couplings. Future work could address this via structured variational distributions (Mishkin et al., 2018) or amortised inference (Kingma & Welling, 2014).

## Acknowledgements

We would like to thank the anonymous reviewers for their insightful comments and constructive suggestions. This work was supported by computational resources from the University of Manchester's CSF3 facility. XJ is supported by a UoM Departmental Studentship; XS is supported by the UoM-CSC Joint Scholarship; MR acknowledges funding from the Wellcome Trust (227415/Z/23/Z) and UKRI-EPSRC (EP/Y028805/1); MA has been financed by the EPSRC Research Projects EP/R034303/1, EP/T00343X/2, EP/V029045/1, the UKRI cross-council grant MR/Z505468/1, and the Wellcome Trust project 217068/Z/19/Z.

## Impact Statement

This paper presents work whose goal is to advance the field of Machine Learning. There are many potential societal consequences of our work, none of which we feel must be specifically highlighted here.

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

# Transformed Latent Variable Multi-Output Gaussian Processes: Supplementary Materials

## Table of Contents

# A. Notations and Acronyms

**Table 11.** Summary of notations and their descriptions.

| Notation | Description |
|---|---|
| $D_X$ | Input dimensionality |
| $D_H$ | Latent variable dimensionality |
| $D_T$ | Dimensionality of the embedding space |
| $N$ | Number of inputs |
| $P$ | Number of outputs |
| $N_b$ | Number of inputs in a mini-batch, i.e. $|\mathcal{B}_N|$ |
| $P_b$ | Number of outputs in a mini-batch, i.e. $|\mathcal{B}_P|$ |
| $M$ | Number of inducing variables |
| $\mathbf{Z}$ | Inducing points / inputs |
| $\mathbf{u}$ | Inducing variables |
| $L$ | Number of latent SVGPs (used in SV-LMC, SGPRN and OILMM) |
| $Q$ | Number of Coregionalisation matrices (used in GS-LVMOGP) |
| $M_X$ | Number of inducing points in the input space (used in GS-LVMOGP) |
| $M_H$ | Number of inducing points in the latent space (used in GS-LVMOGP) |

**Table 12.** Acronym summary.

| Acronym | Meaning |
|---|---|
| GP | Gaussian Process |
| MOGP | Multi-Output Gaussian Process |
| SVGP | Sparse Variational Gaussian Process |
| NN | Neural Network |
| RCNN | Residual Connected Neural Network |
| Ind-GP | Independent Gaussian Process |
| LMC | Linear Model of Coregionalisation |
| SV-LMC | Sparse Variational Linear Model of Coregionalisation |
| G-MOGP | Graphical Multi-Output Gaussian Process |
| GPRN | Gaussian Process Regression Network |
| SGPRN | Scalable Gaussian Process Regression Network |
| OILMM | Orthogonal Instantaneous Linear Mixing Model |
| LV-MOGP | Latent Variable Multi-Output Gaussian Process |
| T-LVMOGP | Transformed Latent Variable Multi-Output Gaussian Process |
| GS-LVMOGP | Generalised Scalable Latent Variable Multi-Output Gaussian Process |
| SN | Spectral Normalisation |
| SN-UB | Spectral Norm Upper Bound |
| ZINB | Zero-Inflated Negative Binomial |
| NLL | Negative Log-Likelihood |
| MSE | Mean Squared Error |

## B. ELBO Derivation

In this section, we derive the ELBO of the T-LVMOGP model. Consider the latent variables $\{\mathbf{H}, \mathbf{f}, \mathbf{u}\}$ and observations $\mathbf{Y}$. We introduce the variational distribution $q(\mathbf{H}, \mathbf{u}, \mathbf{f}) = q(\mathbf{H})q(\mathbf{u})p(\mathbf{f} \mid \mathbf{u}, \mathbf{H})$, and the joint distribution is $p(\mathbf{Y}, \mathbf{H}, \mathbf{u}, \mathbf{f}) = p(\mathbf{Y} \mid \mathbf{f})p(\mathbf{H})p(\mathbf{u})p(\mathbf{f} \mid \mathbf{u}, \mathbf{H})$.

$$
\begin{aligned}
\log p(\mathbf{Y}) &= \log \int p(\mathbf{Y}, \mathbf{H}, \mathbf{u}, \mathbf{f}) d\mathbf{H} \, d\mathbf{u} \, d\mathbf{f} \\
&= \log \int q(\mathbf{H}, \mathbf{u}, \mathbf{f}) \frac{p(\mathbf{Y}, \mathbf{H}, \mathbf{u}, \mathbf{f})}{q(\mathbf{H}, \mathbf{u}, \mathbf{f})} d\mathbf{H} \, d\mathbf{u} \, d\mathbf{f} \\
&\geq \int q(\mathbf{H}, \mathbf{u}, \mathbf{f}) \log \left[ \frac{p(\mathbf{Y}, \mathbf{H}, \mathbf{u}, \mathbf{f})}{q(\mathbf{H}, \mathbf{u}, \mathbf{f})} \right] d\mathbf{H} \, d\mathbf{u} \, d\mathbf{f} \\
&= \int q(\mathbf{H}, \mathbf{u}, \mathbf{f}) \log \frac{p(\mathbf{H})p(\mathbf{u})p(\mathbf{f} \mid \mathbf{u}, \mathbf{H})p(\mathbf{Y} \mid \mathbf{f})}{q(\mathbf{H}, \mathbf{u}, \mathbf{f})} d\mathbf{H} \, d\mathbf{u} \, d\mathbf{f} \\
&= \int q(\mathbf{H})q(\mathbf{u})p(\mathbf{f} \mid \mathbf{u}, \mathbf{H}) \log \frac{p(\mathbf{H})p(\mathbf{u})p(\mathbf{Y} \mid \mathbf{f})}{q(\mathbf{H})q(\mathbf{u})} d\mathbf{H} \, d\mathbf{u} \, d\mathbf{f} \\
&= \int q(\mathbf{H})q(\mathbf{f} \mid \mathbf{H}) \left[ \log p(\mathbf{Y} \mid \mathbf{f}) \right] - \mathrm{KL}[q(\mathbf{u}) \mid\mid p(\mathbf{u})] - \mathrm{KL}[q(\mathbf{H}) \mid\mid p(\mathbf{H})] \\
&= \sum_{n=1}^{N} \sum_{p=1}^{P} \mathbb{E}_{q(\mathbf{h}_p)q(f_p(\mathbf{x}_n) \mid \mathbf{x}_n, \mathbf{h}_p)} \left[ \log p(y_{n,p} \mid f_p(\mathbf{x}_n)) \right] - \mathrm{KL}[q(\mathbf{u}) \mid\mid p(\mathbf{u})] - \sum_{p=1}^{P} \mathrm{KL}[q(\mathbf{h}_p) \mid\mid p(\mathbf{h}_p)] \\
&= \sum_{n=1}^{N} \sum_{p=1}^{P} \mathbb{E}_{q(\mathbf{h}_p)} \left[ \mathcal{V}_{n,p}(\mathbf{h}_p) \right] - \mathrm{KL}[q(\mathbf{u}) \mid\mid p(\mathbf{u})] - \sum_{p=1}^{P} \mathrm{KL}[q(\mathbf{h}_p) \mid\mid p(\mathbf{h}_p)] \\
&:= \mathcal{L}_3,
\end{aligned}
$$

where the inequality follows from Jensen's inequality, and

$$
\mathrm{KL}[q(\mathbf{H}) \mid\mid p(\mathbf{H})] = \int \left( \prod_{p=1}^{P} q(\mathbf{h}_p) \right) \log \left[ \frac{\prod_{p=1}^{P} q(\mathbf{h}_p)}{\prod_{p=1}^{P} p(\mathbf{h}_p)} \right] d\mathbf{h}_1 ... d\mathbf{h}_P = \sum_{p=1}^{P} \int q(\mathbf{h}_p) \log \left[ \frac{q(\mathbf{h}_p)}{p(\mathbf{h}_p)} \right] d\mathbf{h}_p = \sum_{p=1}^{P} \mathrm{KL}[q(\mathbf{h}_p) \mid\mid p(\mathbf{h}_p)].
$$

For the Gaussian likelihood, the inner expectation $\mathcal{V}_{n,p}(\mathbf{h}_p)$ can be analytically expressed as

$$
\mathcal{V}_{n,p}(\mathbf{h}_p) = \log \mathcal{N}(y_{n,p} \mid \mathbf{k}_{\widetilde{\mathbf{x}}_{n,p}}^{\top} \mathbf{K}_{\mathbf{ZZ}}^{-1} \mathbf{m}, \sigma_y^2) - \frac{1}{2\sigma_y^2} \mathrm{tr}(\mathbf{S}\mathbf{\Lambda}_{n,p}) - \frac{1}{2\sigma_y^2}(k_{\widetilde{\mathbf{x}}_{n,p}} - q_{\widetilde{\mathbf{x}}_{n,p}}),
$$

where $\mathbf{\Lambda}_{n,p} = \mathbf{K}_{\mathbf{ZZ}}^{-1} \mathbf{k}_{\widetilde{\mathbf{x}}_{n,p}} \mathbf{k}_{\widetilde{\mathbf{x}}_{n,p}}^{\top} \mathbf{K}_{\mathbf{ZZ}}^{-1}$ and $q_{\widetilde{\mathbf{x}}_{n,p}} = \mathbf{k}_{\widetilde{\mathbf{x}}_{n,p}}^{\top} \mathbf{K}_{\mathbf{ZZ}}^{-1} \mathbf{k}_{\widetilde{\mathbf{x}}_{n,p}}$.

## C. Lipschitz Continuity of Residual Connected Neural Network (RCNN)

**Definition C.1** (Lipschitz continuous). Given two metric spaces $(\mathcal{X}, d_X)$ and $(\mathcal{Y}, d_Y)$, where $d_X$ denotes the metric on the set $\mathcal{X}$ and $d_Y$ is the metric on the set $\mathcal{Y}$, a function $g: \mathcal{X} \to \mathcal{Y}$ is called Lipschitz continuous if there exists a real constant $C \geq 0$ such that, for all $x_1$ and $x_2$ in $\mathcal{X}$, $d_Y(g(x_1), g(x_2)) \leq C d_X(x_1, x_2)$. The smallest constant $C$ is sometimes called the (best) Lipschitz constant of $g$.

*Remark* C.2. The Lipschitz constant of a function $g(x)$ can be denoted as

$$
C = \sup_{x_1 \neq x_2} \frac{d_Y(g(x_1), g(x_2))}{d_X(x_1, x_2)}.
$$

**Definition C.3** (C-bilipschitz). For a real number $C \geq 1$, if $\frac{1}{C} d_X(x_1, x_2) \leq d_Y(g(x_1), g(x_2)) \leq C d_X(x_1, x_2)$ for all $x_1, x_2 \in \mathcal{X}$, then $g$ is called C-bilipschitz.

**Proposition C.4** (informal, Theorem 1 of Bartlett et al. (2018)). *Any smooth bilipschitz function $g$ can be represented as a composition $g_m \circ g_{m-1} \circ ... \circ g_1$ of functions $g_1(\cdot), ..., g_m(\cdot)$ that are close to the identity in the sense that each $(g_i - Id)$ is Lipschitz continuous.*

**Proposition C.5** (Proposition 1 of Liu et al. (2020)). *Consider a mapping $g : \mathcal{X} \to \mathcal{Y}$ with residual architecture $g = g_L \circ g_{L-1} \circ ... \circ g_2 \circ g_1$ where $g_l(\mathbf{x}) = \mathbf{x} + h_l(\mathbf{x})$. Both $\mathcal{X}$ and $\mathcal{Y}$ spaces are equipped with the Euclidean norm induced distance $|| \cdot ||$, i.e. $d_X(a,b) = d_Y(a,b) = ||a - b||$, which will be used interchangeably below. If for $0 \leq \alpha < 1$, all $h_l$ are $\alpha$-Lipschitz, i.e, $d_Y(h_l(\mathbf{x}), h_l(\mathbf{x}')) \leq \alpha \cdot d_X(\mathbf{x}, \mathbf{x}') \quad \forall \mathbf{x}, \mathbf{x}' \in \mathcal{X}$. Then:*

$$L_1 \cdot d_X(\mathbf{x}, \mathbf{x}') \leq d_Y(g(\mathbf{x}), g(\mathbf{x}')) \leq L_2 \cdot d_X(\mathbf{x}, \mathbf{x}'),$$

*where $L_1 = (1 - \alpha)^L$ and $L_2 = (1 + \alpha)^L$, i.e., $g$ is confined to have bounded distortion.*

*Proof.* This proof is an adaptation of the result of Liu et al. (2020) to our context. We first prove the following proposition:

$$\forall l, \quad (1 - \alpha)d_X(\mathbf{x}, \mathbf{x}') \leq d_Y(g_l(\mathbf{x}), g_l(\mathbf{x}')) \leq (1 + \alpha)d_X(\mathbf{x}, \mathbf{x}').$$

First, prove the left-hand side:

$$\begin{aligned}
d_X(\mathbf{x}, \mathbf{x}') = ||\mathbf{x} - \mathbf{x}'|| &= ||\mathbf{x} - \mathbf{x}' - (g_l(\mathbf{x}) - g_l(\mathbf{x}')) + (g_l(\mathbf{x}) - g_l(\mathbf{x}'))|| \\
&\leq ||(g_l(\mathbf{x}') - \mathbf{x}') - (g_l(\mathbf{x}) - \mathbf{x})|| + ||g_l(\mathbf{x}) - g_l(\mathbf{x}')|| \\
&\leq ||h_l(\mathbf{x}') - h_l(\mathbf{x})|| + ||g_l(\mathbf{x}) - g_l(\mathbf{x}')|| \\
&\leq \alpha \cdot ||\mathbf{x}' - \mathbf{x}|| + ||g_l(\mathbf{x}) - g_l(\mathbf{x}')|| \quad (h_l \text{ are } \alpha\text{-Lipschitz}) \\
&= \alpha \cdot d_X(\mathbf{x}, \mathbf{x}') + d_Y(g_l(\mathbf{x}), g_l(\mathbf{x}')).
\end{aligned}$$

Rearranging, we get

$$(1 - \alpha)d_X(\mathbf{x}, \mathbf{x}') \leq d_Y(g_l(\mathbf{x}), g_l(\mathbf{x}')).$$

Now show the right-hand side:

$$\begin{aligned}
d_Y(g_l(\mathbf{x}), g_l(\mathbf{x}')) = ||g_l(\mathbf{x}) - g_l(\mathbf{x}')|| \\
= ||\mathbf{x} + h_l(\mathbf{x}) - (\mathbf{x}' + h_l(\mathbf{x}'))|| \\
\leq ||\mathbf{x} - \mathbf{x}'|| + ||h_l(\mathbf{x}) - h_l(\mathbf{x}')|| \\
\leq (1 + \alpha)||\mathbf{x} - \mathbf{x}'|| \quad (h_l \text{ are } \alpha\text{-Lipschitz}) \\
= (1 + \alpha)d_X(\mathbf{x}, \mathbf{x}').
\end{aligned}$$

Now show the bi-Lipschitz condition for an $L$-layer residual connected network function $g = g_L \circ g_{L-1} \circ ... \circ g_2 \circ g_1$. It is easy to see that by induction, we get:

$$(1 - \alpha)^L d_X(\mathbf{x}, \mathbf{x}') \leq d_Y(g(\mathbf{x}), g(\mathbf{x}')) \leq (1 + \alpha)^L d_X(\mathbf{x}, \mathbf{x}').$$

$\square$

*Remark* C.6. If $h_l(\mathbf{x}) = \phi(\mathbf{W}_l\mathbf{x} + \mathbf{b}_l)$, where $\phi$ denotes activation function, then $C_{h_l} = C_\phi \times ||\mathbf{W}_l||_2$, $C_{h_l}$ and $C_\phi$ refer to Lipschitz constants for functions $h_l(\cdot)$ and $\phi(\cdot)$ respectively, $||\mathbf{W}_l||_2$ is the spectral norm of $\mathbf{W}_l$. For common activation functions (such as ReLU, sigmoid, tanh), $C_\phi = 1$, thus $C_{h_l} = ||\mathbf{W}_l||_2$.

*Remark* C.7. Building on Prop. C.5 and Remark C.6, we realise that for residual connected neural networks comprising linear layers, the bi-Lipschitz property of the overall transformation is guaranteed by applying proper spectral normalisation to the weight matrix of each layer with SN-UB $< 1$. The Lipschitz property holds even when SN-UB $\geq 1$.

### C.1. Spectral normalisation

**Definition C.8** (Matrix Norm). Given a field $K$ of either real or complex numbers, the matrix norm is a function $|| \cdot || : K^{m \times n} \to \mathbb{R}^{0+}$ that must satisfy the following properties:

For all scalar $\alpha \in K$ and matrices $\mathbf{A}, \mathbf{B} \in K^{m \times n}$,

- $||\mathbf{A}|| \geq 0$ (positive-valued)

- $||\mathbf{A}|| = 0 \Longleftrightarrow \mathbf{A} = \mathbf{0}_{m,n}$ (definite)

- $||\alpha\mathbf{A}|| = |\alpha|\ ||\mathbf{A}||$ (absolutely homogeneous)

- $||\mathbf{A} + \mathbf{B}|| \leq ||\mathbf{A}|| + ||\mathbf{B}||$ (sub-additive or satisfying the triangle inequality)

- $||\mathbf{AB}|| \leq ||\mathbf{A}||\ ||\mathbf{B}||$ (sub-multiplicative)

**Definition C.9** (Matrix norms induced by vector norms). Suppose a vector norm $||\cdot||_\alpha$ on $K^n$ and a vector norm $||\cdot||_\beta$ on $K^m$ are given. Any $m \times n$ matrix $\mathbf{A}$ induces a linear operator from $K^n$ to $K^m$ with respect to the standard basis, and one defines the corresponding induced norm or operator norm on the space $K^{m \times n}$ of all $m \times n$ matrices as follows:

$$||\mathbf{A}||_{\alpha,\beta} = \sup\left\{||\mathbf{Ax}||_\beta : \mathbf{x} \in K^n \text{ such that } ||\mathbf{x}||_\alpha \leq 1\right\}.$$

**Definition C.10** (Matrix norms induced by vector p-norms). If the $p$-norm for vectors ($1 \leq p \leq \infty$) is used for both spaces $K^n$ and $K^m$, then the corresponding operator norm is

$$||\mathbf{A}||_p = \sup\left\{||\mathbf{Ax}||_p : \mathbf{x} \in K^n \text{ such that } ||\mathbf{x}||_p \leq 1\right\}.$$

*Remark* C.11. Geometrically speaking, one can imagine a $p$-norm unit ball $\mathbb{V}_{p,n} = \{\mathbf{x} \in K^n : ||\mathbf{x}||_p \leq 1\}$ in $K^n$, then apply the linear map $\mathbf{A}$ to the ball. It becomes a distorted convex shape $\mathbf{A}\mathbb{V}_{p,n} \subset K^m$, and $||\mathbf{A}||_p$ measures the longest "radius" of the distorted convex shape. In other words, we must take a $p$-norm unit ball $\mathbb{V}_{p,m}$ in $K^m$, then multiply it by at least $||\mathbf{A}||_p$, in order for it to be large enough to contain $\mathbf{A}\mathbb{V}_{p,n}$. Thus,

$$||\mathbf{A}||_p = \sup_{||\mathbf{x}||_p = 1}\left\{||\mathbf{Ax}||_p : \mathbf{x} \in \mathbb{R}^n\right\}.$$

**Definition C.12** (Spectral norm, $p = 2$). When $p = 2$ (the Euclidean norm), the induced matrix norm is the *spectral norm*.

**Proposition C.13.** *The spectral norm of a matrix $\mathbf{A}$ is the largest singular value of $\mathbf{A}$, i.e., the square root of the largest eigenvalue of the matrix $\mathbf{A}^*\mathbf{A}$, where $\mathbf{A}^*$ denotes the conjugate transpose of $\mathbf{A}$:*

$$||\mathbf{A}||_2 = \sqrt{\lambda_{max}(\mathbf{A}^*\mathbf{A})} = \sigma_{max}(\mathbf{A}).$$

*where $\sigma_{max}(\mathbf{A})$ represents the largest singular value of matrix $\mathbf{A}$.*

*Proof.* Using the Rayleigh quotient: $||\mathbf{A}||_2^2 = \sup\{R(\mathbf{A}^*\mathbf{A}, \mathbf{x}) : \mathbf{x} \in \mathbb{R}^n\}$. $\qquad\square$

**Proposition C.14** (Power / Von Mises iteration). *For $\mathbf{M} \in \mathbb{R}^{n \times n}$, if we assume $\mathbf{M}$ has an eigenvalue that is strictly greater in magnitude than that of its other eigenvalues and the starting vector $\mathbf{b}_0$ has a nonzero component in the direction of the eigenvector associated with the dominant eigenvalue, then an iterative procedure converge to dominant eigenvalue and corresponding eigenvector:*

$$\mathbf{b}_{k+1} = \frac{\mathbf{Mb}_k}{||\mathbf{Mb}_k||_2}, \quad \tilde{\lambda}_{max} = \lambda_k = \frac{\mathbf{b}_k^*\mathbf{Mb}_k}{\mathbf{b}_k^*\mathbf{b}_k}.$$

---

**Algorithm 1** Spectral Normalisation for $\mathbf{A} \in \mathbb{R}^{m \times n}$ using Power Iteration (Miyato et al., 2018)

---

1: **Input:** $\mathbf{A} \in \mathbb{R}^{m \times n}$; $c > 0$: spectral norm upper bound (SN-UB); number of power iterations $T$
2: **Output:** $\tilde{\mathbf{A}}$ {(approximately) spectral normalised $\mathbf{A}$}
3: Initialise $\tilde{\mathbf{v}}$ with random vectors {sampled from standard Gaussian distribution}
4: $i \leftarrow 0$
5: **while** $i < T$ **do**
6: $\quad$ $\tilde{\mathbf{u}} \leftarrow \mathbf{A}\tilde{\mathbf{v}} \, / \, \|\mathbf{A}\tilde{\mathbf{v}}\|_2$ {(approximation of) the first left singular vector}
7: $\quad$ $\tilde{\mathbf{v}} \leftarrow \mathbf{A}^\top\tilde{\mathbf{u}} \, / \, \|\mathbf{A}^\top\tilde{\mathbf{u}}\|_2$ {(approximation of) the first right singular vector}
8: $\quad$ $i \leftarrow i + 1$
9: **end while**
10: $\tilde{\sigma} \leftarrow \tilde{\mathbf{u}}^\top \mathbf{A}\tilde{\mathbf{v}}$ {(approximation of) the first singular value}
11: **if** $\tilde{\sigma} \leq c$ **then**
12: $\quad$ $\tilde{\mathbf{A}} \leftarrow \mathbf{A}$
13: **else**
14: $\quad$ $\tilde{\mathbf{A}} \leftarrow c\mathbf{A} \, / \, \tilde{\sigma}$
15: **end if**

---

## D. Connections to Prior LV-MOGP Models

In this section, we relate the kernel structure of T-LVMOGP to that of prior Latent Variable MOGP models (Dai et al., 2017; Jiang et al., 2025). We focus on the (ARD)-RBF kernels used throughout the paper.

**T-LVMOGP kernel** T-LVMOGP defines a valid covariance function on the augmented space $(\mathbf{x}, \mathbf{h}_p)$ via an embedding mapping $\Phi_\theta : \mathbb{R}^{D_X} \times \mathbb{R}^{D_H} \to \mathbb{R}^{D_T}$ and a stationary base kernel:

$$k_{\text{T-LVMOGP}}((\mathbf{x}, \mathbf{h}_p), (\mathbf{x}', \mathbf{h}_{p'})) = k_{\text{base}}(\Phi_\theta(\mathbf{x}, \mathbf{h}_p), \Phi_\theta(\mathbf{x}', \mathbf{h}_{p'})).$$

For an ARD-RBF kernel on $\mathbb{R}^{D_T}$,

$$k_{\text{RBF}}(\mathbf{z}, \mathbf{z}') = \sigma^2 \exp\Big( -\tfrac{1}{2}(\mathbf{z} - \mathbf{z}')^\top \mathbf{\Lambda}^{-1}(\mathbf{z} - \mathbf{z}') \Big), \quad \mathbf{\Lambda} = \text{diag}(\ell_1^2, \ldots, \ell_{D_T}^2), \quad \sigma^2 \text{ represents outputscale.}$$

**LV-MOGP ($Q = 1$) as a special case** Dai et al. (2017) introduce LV-MOGP ($Q = 1$) by associating each output $p$ with a latent vector $\mathbf{h}_p$ and using a separable kernel

$$k_{\text{LV-MOGP}}((\mathbf{x}, \mathbf{h}_p), (\mathbf{x}', \mathbf{h}_{p'})) = k_X^{(1)}(\mathbf{x}, \mathbf{x}') \, k_H^{(1)}(\mathbf{h}_p, \mathbf{h}_{p'}).$$

Consider T-LVMOGP with $k_{\text{base}} = k_{\text{RBF}}$ on $\mathbb{R}^{D_X + D_H}$ and a blockwise linear embedding

$$\Phi_\theta(\mathbf{x}, \mathbf{h}) = \begin{bmatrix} \mathbf{x} \oslash \boldsymbol{\ell}_X \\ \mathbf{h} \oslash \boldsymbol{\ell}_H \end{bmatrix},$$

where $\oslash$ denotes element-wise (Hadamard) division, $\boldsymbol{\ell}_X \in \mathbb{R}_{>0}^{D_X}$ and $\boldsymbol{\ell}_H \in \mathbb{R}_{>0}^{D_H}$. Then the exponent separates into input and latent parts, yielding the exact factorisation

$$k_{\text{T-LVMOGP}}((\mathbf{x}, \mathbf{h}_p), (\mathbf{x}', \mathbf{h}_{p'})) = \underbrace{\exp\Big( -\frac{1}{2}\sum_{d=1}^{D_X} \frac{(x_d - x_d')^2}{\ell_{X,d}^2} \Big)}_{k_X^{(1)}(\mathbf{x}, \mathbf{x}')} \cdot \underbrace{\exp\Big( -\frac{1}{2}\sum_{d=1}^{D_H} \frac{(h_{p,d} - h_{p',d})^2}{\ell_{H,d}^2} \Big)}_{k_H^{(1)}(\mathbf{h}_p, \mathbf{h}_{p'})}.$$

Therefore, under the (ARD)-RBF setting, the kernel employed in LV-MOGP is a special case of that of T-LVMOGP obtained by restricting $\Phi_\theta$ to a blockwise linear scaling.

**GS-LVMOGP** ($Q > 1$) **as a special case** GS-LVMOGP (Jiang et al., 2025) extends LV-MOGP to a sum-of-separable kernel:

$$k_{\text{GS-LVMOGP}}((\mathbf{x}, \mathbf{h}_p), (\mathbf{x}', \mathbf{h}_{p'})) = \sum_{q=1}^{Q} k_X^{(q)}(\mathbf{x}, \mathbf{x}') \, k_H^{(q)}(\mathbf{h}_{p,q}, \mathbf{h}_{p',q}),$$

where $\mathbf{h}_p = \{\mathbf{h}_{p,q}\}_{q=1}^{Q}$. This can be recovered by T-LVMOGP by using a block-structured embedding $\Phi_\theta(\mathbf{x}, \mathbf{h}_p) = \text{concat}\left(\Phi_\theta^{(1)}(\mathbf{x}, \mathbf{h}_{p,1}), \ldots, \Phi_\theta^{(Q)}(\mathbf{x}, \mathbf{h}_{p,Q})\right)$ and an additive base kernel

$$k_{\text{base}}(\mathbf{z}, \mathbf{z}') = \sum_{q=1}^{Q} k_{\text{RBF}}^{(q)}(\mathbf{z}^{(q)}, \mathbf{z}'^{(q)}), \quad \mathbf{z}^{(q)} = \Phi_\theta^{(q)}(\mathbf{x}, \mathbf{h}_{p,q}),$$

with $\Phi_\theta^{(q)}(\mathbf{x}, \mathbf{h}) = [\mathbf{x} \oslash \boldsymbol{\ell}_X^{(q)}; \mathbf{h} \oslash \boldsymbol{\ell}_H^{(q)}]$. Each additive component factorises as in the $Q = 1$ case, giving exactly the sum-of-separable GS-LVMOGP kernel. Consequently, GS-LVMOGP is also a special case of T-LVMOGP, while T-LVMOGP generalises these models by allowing a general (regularised) neural network $\Phi_\theta$ that induces non-separable covariances not representable as $\sum_q k_X^{(q)} k_H^{(q)}$.

Although T-LVMOGP can recover the kernel classes of LV-MOGP ($Q=1$) and GS-LVMOGP ($Q>1$) through appropriate restrictions on $\Phi_\theta$ and $k_{\text{base}}$, the resulting inference procedures are not equivalent. In particular, T-LVMOGP performs sparse variational inference by placing inducing inputs in the learned embedding space induced by $\Phi_\theta$, whereas LV-MOGP and GS-LVMOGP typically introduce inducing points separately in the input space and the latent space to exploit the separable (or sum-of-separable) structure. Consequently, our "special-case" statements should be interpreted at the kernel structure level rather than as an exact equivalence of the entire model.

## E. Zero-Inflated Negative Binomial (ZINB) Likelihood

**Negative Binomial (NB) Distribution** NB distribution can be regarded as a mixture of Poisson distributions with a Gamma prior:

$$\text{Poisson:} \quad y \mid \lambda \sim \text{Pois}(\lambda), \quad p(y \mid \lambda) = e^{-\lambda} \frac{\lambda^y}{y!},$$

$$\text{Gamma (shape-rate parametrisation):} \quad \lambda \sim \text{Gamma}(r, \beta), \quad p(\lambda) = \frac{\beta^r}{\Gamma(r)} \lambda^{r-1} e^{-\beta\lambda}, \ \lambda > 0,$$

$$\text{NB}(Y = y) = p_Y(y) = \int p(y \mid \lambda) p(\lambda) d\lambda = \binom{y + r - 1}{y} \left(\frac{1}{1+\beta}\right)^r \left(\frac{\beta}{1+\beta}\right)^y.$$

Recall the canonical NB form is (count of $y$ failures before $r$ successes, success probability $p$):

$$p_Y(y) = \binom{y + r - 1}{y} (1 - p)^y p^r, \quad y = 0, 1, 2, \ldots$$

We recognise that:

$$p = \frac{1}{1+\beta}, \quad 1 - p = \frac{\beta}{1+\beta}.$$

The negative binomial has a mean and a variance defined by

$$\mathbb{E}[Y] = \frac{r(1-p)}{p} \equiv m, \quad \text{Var}[Y] = \frac{r(1-p)}{p^2} = m + \frac{m^2}{r}.$$

Consider $r \to \infty$ while keeping $m$ fixed; $\text{Var}[Y] \to m$, and the NB converges to the Poisson distribution.

**Mean-Dispersion Parametrisation** NB can be alternatively parametrised by a mean parameter ($m$) and a dispersion parameter ($\alpha = 1/r$), defined as follows [2]:

$$\text{NB}(y \mid m, \alpha) = \frac{\Gamma(r + y)}{\Gamma(r) y!} \left(\frac{r}{r + m}\right)^{1/\alpha} \left(\frac{m}{r + m}\right)^y, \quad y \in \{0, 1, \ldots\}; \quad \text{with mean } m, \text{ variance } m + m^2 \alpha.$$

---

[2] Recall that $\Gamma(n) = (n - 1)!$.

**Extra-Zero Mechanism (Michaelis–Menten)**   A zero-inflation probability

$$\psi = 1 - \frac{m}{k_m + m},$$

with $k_m \geq 0$, it is borrowed from the Michaelis–Menten saturation curve: small $m \Rightarrow \psi \approx 1$ (many extra zeros); large $m \Rightarrow \psi \to 0$. If $k_m$ is chosen to be 0, $\psi$ is always 0, and there is no zero inflation.

**Zero-Inflated Negative Binomial**   In negative binomial regression, the mean of the distribution is related to the explanatory variables. For GP models, $m$ can be modelled as the positively transformed latent GP function $f$ times a positive scaling factor.

$$p(Y = 0 \mid f) = \psi + (1 - \psi) \, \mathrm{NB}(Y = 0 \mid m, \alpha),$$
$$p(Y = y > 0 \mid f) = (1 - \psi) \, \mathrm{NB}(Y = y \mid m, \alpha).$$

Recall that $\mathrm{NB}(0 \mid m, \alpha) = \left( \frac{r}{r+m} \right)^r = (1 + \alpha m)^{-1/\alpha}$. The summary of three key formulas:

$$\log P(Y = y \mid f) = \begin{cases} \log \left[ \psi + (1 - \psi)(1 + \alpha m)^{-1/\alpha} \right], & y = 0 \\ \log(1 - \psi) + \log \mathrm{NB}(y \mid m, \alpha), & y > 0; \end{cases}$$
$$\mathbb{E}[Y = y \mid f] = (1 - \psi)m;$$
$$\mathrm{Var}(Y = y \mid f) = m(1 - \psi)[1 + m(\alpha + \psi)].$$

# F. Experimental Details and Additional Results

The code implementation is available from `https://github.com/XiaoyuJiang17/T-LVMOGP-official`.

To ensure a fair comparison, we implement all models, including the proposed T-LVMOGP and the baselines (Ind-GP, SV-LMC, SGPRN, OILMM, G-MOGP, and GS-LVMOGP), using the `PyTorch` (Paszke et al., 2019) and `GPyTorch` (Gardner et al., 2018) frameworks. Experiments are conducted on a high-performance computing cluster utilising NVIDIA HGX A100-SXM4 or NVIDIA L40S-SXM4 GPUs, with all computations performed in double precision (`Float64`). For computational efficiency assessment, training times are benchmarked on an NVIDIA RTX-5090 GPU. Throughout this work, we employ the ARD-RBF kernel as the base kernel of T-LVMOGP for simplicity; and $J = 1$ Monte Carlo sample is adopted for model training. In terms of tighter variational bounds, the correction term $\Delta$ given by Equation (1) is evaluated conditionally on sampled $\mathbf{h}_p^{(j)}$. When data are missing, we denote $\Omega \subseteq \{1, ..., N\} \times \{1, ..., P\}$ as the index set for training data with $|\Omega| = N_{\text{train}}$. We draw input-output pairs from $\Omega$ and scale the mini-batch expected log-likelihood term in $\widetilde{\mathcal{L}}_3$ by $N_{\text{train}}/(NP)$, where $N_{\text{train}} < NP$ denotes the total number of observed training data, to ensure an unbiased weight for the expected log-likelihood term.

We choose $S = 20$ samples for latent variables when making predictions on the test set. The NLL is computed as follows:

$$\mathrm{NLL} = -\log q(y_* \mid \mathbf{x}_*) = -\log \int q(\mathbf{h}_{p_*}) q(f_{p_*}(\mathbf{x}_*) \mid \mathbf{x}_*, \mathbf{h}_{p_*}) p(y_* \mid f_{p_*}(\mathbf{x}_*)) \, d\mathbf{h}_{p_*} \, df_{p_*}(\mathbf{x}_*)$$

$$\approx -\log \frac{1}{S} \sum_s \int q(f_{p_*}(\mathbf{x}_*) \mid \mathbf{x}_*, \mathbf{h}_{p_*}^s) p(y_* \mid f_{p_*}(\mathbf{x}_*)) \, df_{p_*}(\mathbf{x}_*)$$

$$= \log S - \underset{s=1,...,S}{\mathrm{logsumexp}} \left[ \log \underbrace{\int q(f_{p_*}(\mathbf{x}_*) \mid \mathbf{x}_*, \mathbf{h}_{p_*}^s) p(y_* \mid f_{p_*}(\mathbf{x}_*)) \, df_{p_*}(\mathbf{x}_*)}_{q(y_* \mid \mathbf{h}_{p_*}^s)} \right].$$

For the Gaussian likelihood, the integral for $q(y_* \mid \mathbf{h}_{p_*}^s)$ can be analytically solved. For non-Gaussian likelihoods, we approximate it by plugging in the mean value of $q(f_{p_*}(\mathbf{x}_*) \mid \mathbf{x}_*, \mathbf{h}_{p_*}^s)$, i.e.

$$q(y_* \mid \mathbf{h}_{p_*}^s) \approx p(y_* \mid f_{\text{mean}*}), \text{ where } f_{\text{mean}*} \text{ represents the mean value of } q(f_{p_*}(\mathbf{x}_*) \mid \mathbf{x}_*, \mathbf{h}_{p_*}^s).$$

## F.1. Dataset Statistics

The summary of dataset statistics is presented in Table 13. We adopt the Gaussian likelihood for all experiments, except the Spatial Transcriptomics experiment, where we use a zero-inflated negative binomial likelihood (more details about this likelihood can be found in Appendix E).

**Table 13.** Summary of dataset statistics. $D_X$ and $P$ denote input and output dimensions, respectively. $N$ denotes the number of inputs, and $N_{train}$ and $N_{test}$ denote the total numbers of training and test data points, respectively.

| Dataset | $D_X$ | $N$ | $P$ | $N_{train}$ | $N_{test}$ |
|---|---|---|---|---|---|
| EEG | 1 | 256 | 7 | $1,492$ | 300 |
| SARCOS | 21 | $48,933$ | 7 | $171,266$ | $171,265$ |
| ERA5-random | 1 | 30 | $3,395$ | $71,850$ | $30,000$ |
| ERA5-block | 1 | 30 | $3,395$ | $71,850$ | $30,000$ |
| Copernicus Marine | 1 | 24 | $10,873$ | $260,952$ | $259,344$ |
| Spatial Transcriptomics | 2 | $4,352$ | $5,000$ | $19,582,941$ | $2,177,059$ |

## F.2. EEG Prediction

We summarise the experimental settings of our model for this experiment in Table 14. Additional prediction plots are provided in Figure 5. For baseline models involving latent functions (SGPRN, SV-LMC, and OILMM), we set $L = 3$, and $M = 200$ inducing points are allocated separately to each latent function for SV-LMC and OILMM. To ensure SGPRN training stability, gradient clipping is used with a maximum norm of 10. We set the coefficient of the regularisation term, $\beta_2$, to 1.6 to improve OILMM's predictive performance. For G-MOGP, $M = 200$ inducing points are used for each output. Finally, regarding GS-LVMOGP, we set $Q = 3$, $D_H = 3$, $M_H = 4$ and $M_X = 50$. Note that $Q = 3$ is more flexible than $Q = 1, 2$ according to Jiang et al. (2025). All baseline models are trained using Adam with a learning rate of 0.01 for 1000 epochs.

**Table 14.** Experimental settings for the EEG experiment.

| Setting | Value |
|---|---|
| Number of residual blocks | 3 |
| Total number of neural network parameters | 60 |
| Spectral norm upper bound (SN-UB) | 0.005 |
| Latent dimensionality ($D_H$) | 3 |
| Number of inducing points ($M$) | 200 |
| Likelihood type | Gaussian likelihood |
| Output batch size ($P_b$) | 7 |
| Input batch size ($N_b$) | 128 |
| Optimiser | Adam, $lr = 0.01$ |
| Training epochs | 1000 |

## F.3. Inverse Dynamics Prediction

The SARCOS dataset is publicly available [3], and a sketch of the anthropomorphic robot arm is shown in Figure 6. The experimental settings are listed in Table 15. We set $L = 7$ for OILMM and $L = 5$ for SV-LMC. For each latent function, we adopt $M = 200$ inducing points. For G-MOGP, we use $M = 100$ inducing points for each output function. Regarding GS-LVMOGP, we have $Q = 3$, $D_H = 3$, $M_H = 5$ and $M_X = 40$. All baseline models are trained with Adam at a learning rate of 0.01 for 1000 epochs.

---

[3] https://gaussianprocess.org/gpml/data/

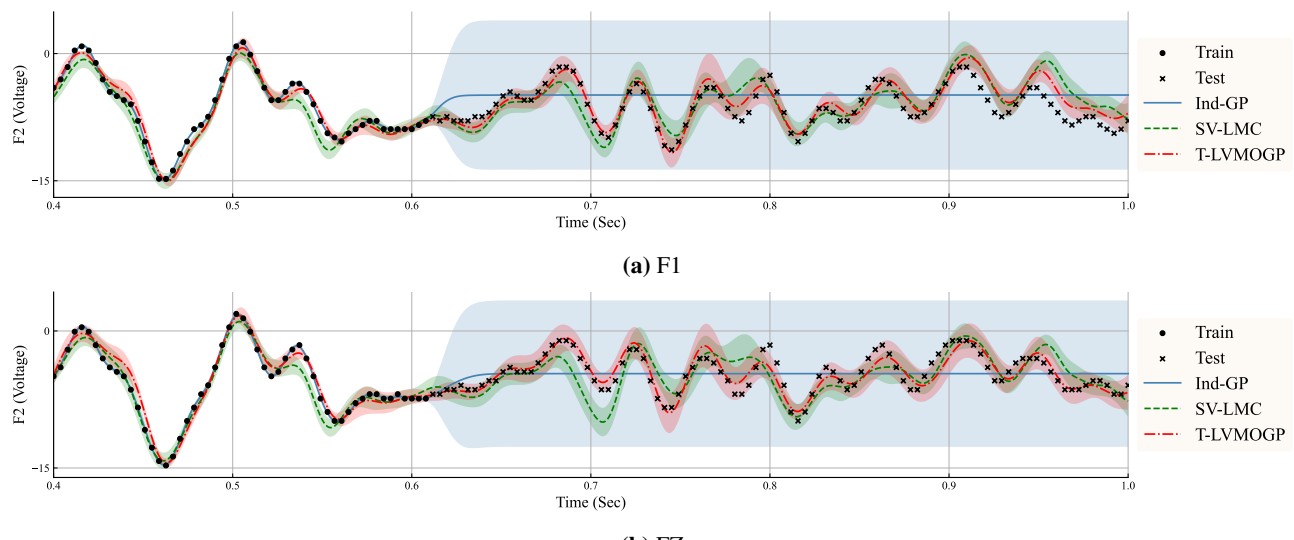

**(a)** F1

**(b)** FZ

**Figure 5.** Prediction plots for Ind-GP, SV-LMC and T-LVMOGP on EEG channels F1 and FZ.

**Table 15.** Experimental settings for the SARCOS experiment.

| Setting | Value |
|---|---|
| Number of residual blocks | 3 |
| Total number of neural network parameters | 2106 |
| Spectral norm upper bound (SN-UB) | 1.0 |
| Latent dimensionality ($D_H$) | 5 |
| Number of inducing points ($M$) | 200 |
| Likelihood type | Gaussian likelihood |
| Output batch size ($P_b$) | 7 |
| Input batch size ($N_b$) | 256 |
| Optimiser | Adam, $lr = 0.01$ |
| Training epochs | 1000 |

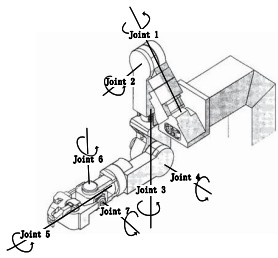

**Figure 6.** Sketch of the SARCOS anthropomorphic robot arm (Vijayakumar & Schaal, 2000; Zhao & Sun, 2016).

## F.4. Climate Modelling

### F.4.1. ERA5

The ERA5 dataset, a fifth-generation atmospheric reanalysis produced by the ECMWF, is a global reanalysis product that provides data on a $0.25° \times 0.25°$ regular latitude-longitude grid. This dataset is available from Climate Data Store [4]. The experimental settings for T-LVMOGP are provided in Table 16. In terms of baselines, SGPRN, SV-LMC and OILMM adopt $L = 100$ latent functions; SV-LMC and OILMM use $M = 10$ inducing points for each latent GP. G-MOGP also utilise $M = 10$ inducing points per output. For GS-LVMOGP, we use the following hyperparameters: $Q = 3$, $D_H = 3$, $M_H = 20$ and $M_X = 10$. All baseline models are trained with Adam at a learning rate of 0.01 for 1000 epochs, except for Ind-GP, which converged within 50 epochs. The experimental settings remain the same for both random and block-wise data splitting.

In this dataset, we additionally consider an alternative train/test partitioning strategy based on block-wise splitting. Concretely, we randomly select 1, 500 outputs and hold out their observations from the first 10 time points as the test set. Independently, we randomly select another 1, 500 outputs and hold out their observations from the last 10 time points for testing. This strategy induces block-wise missing data, where observations are missing over contiguous intervals. This poses a greater challenge for MOGP compared to random splitting.

---

[4]https://cds.climate.copernicus.eu/datasets

***Table 16.*** Experimental settings of the ERA5 experiment for both splitting mechanisms.

| Setting | Value |
|---|---|
| Number of residual blocks | 3 |
| Total number of neural network parameters | 60 |
| Spectral norm upper bound (SN-UB) | 0.1 |
| Latent dimensionality $(D_H)$ | 3 |
| Number of inducing points $(M)$ | 200 |
| Likelihood type | Gaussian likelihood |
| Output batch size $(P_b)$ | 128 |
| Input batch size $(N_b)$ | 30 |
| Optimiser | Adam, $lr = 0.01$ |
| Training epochs | 1000 |

***Table 17.*** Performance comparison of different models on the ERA5 dataset with block-wise splitting.

| Model | MSE ↓ | NLL ↓ | Training Time (s/epoch) ↓ |
|---|---|---|---|
| *Baselines* | | | |
| Ind-GP (Hensman et al., 2013) | $1.065 \pm 0.000$ | $1.459 \pm 0.000$ | $31.401 \pm 0.792$ |
| SGPRN (Li et al., 2020) | $1.039 \pm 0.089$ | $59.928 \pm 4.913$ | $0.780 \pm 0.001$ |
| G-MOGP (Dai et al., 2024) | $0.907 \pm 0.035$ | $1.353 \pm 0.021$ | $0.684 \pm 0.012$ |
| SV-LMC (Van der Wilk et al., 2020) | $0.162 \pm 0.112$ | $1.669 \pm 1.430$ | $0.552 \pm 0.009$ |
| OILMM (Bruinsma et al., 2020) | $0.142 \pm 0.100$ | $0.599 \pm 0.660$ | $\mathbf{0.267 \pm 0.001}$ |
| GS-LVMOGP (Jiang et al., 2025) | $0.019 \pm 0.011$ | $-0.550 \pm 0.230$ | $1.346 \pm 0.020$ |
| T-LVMOGP (ours) | $\mathbf{0.003 \pm 0.000}$ | $\mathbf{-1.503 \pm 0.026}$ | $0.703 \pm 0.014$ |

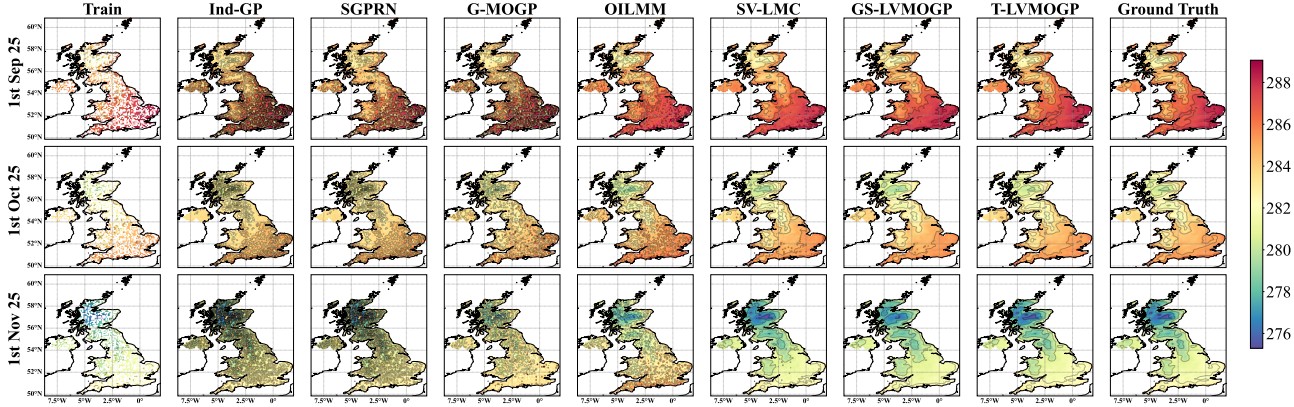

***Figure 7.*** Predictions of different models on the ERA5 dataset with block-wise splitting. The temperature is measured in Kelvin units.

The quantitative comparison between T-LVMOGP and the baseline methods is reported in Table 17, with predictive visualisations provided in Figure 7. As summarised in Table 17, our approach achieves the best performance across all methods under both MSE and NLL metrics. In terms of computational efficiency, OILMM is notably faster than in the random-split setting. This improvement arises because the projection from the observation space to the latent space is accelerated when inputs share similar missing patterns, which is the case for block-wise missing.

We study the impact of the per-layer SN-UB on the ERA5 dataset, with results reported in Table 18 and Table 19. While SN-UB variation only modestly affects MSE, the best NLL is achieved at medium SN-UB values. This trend aligns with our observations from earlier experiments (EEG and SARCOS), suggesting that T-LVMOGP benefits most from a balanced spectral normalisation regime.

**Table 18.** Performance of T-LVMOGP with different SN-UB values on ERA5 random.

| SN-UB | MSE ↓ | NLL ↓ |
|---|---|---|
| 0.5 | $0.0022 \pm 0.0002$ | $-1.556 \pm 0.029$ |
| 0.1 | $0.0022 \pm 0.0001$ | $\mathbf{-1.564 \pm 0.024}$ |
| 0.01 | $0.0022 \pm 0.0001$ | $-1.558 \pm 0.033$ |

**Table 19.** Performance of T-LVMOGP with different SN-UB values on ERA5 block-wise.

| SN-UB | MSE ↓ | NLL ↓ |
|---|---|---|
| 0.5 | $0.0029 \pm 0.0002$ | $-1.475 \pm 0.028$ |
| 0.1 | $\mathbf{0.0027 \pm 0.0002}$ | $\mathbf{-1.503 \pm 0.026}$ |
| 0.01 | $0.0028 \pm 0.0002$ | $-1.492 \pm 0.036$ |

**Table 20.** Experimental settings of the Copernicus Marine experiment.

| Setting | Value |
|---|---|
| Number of residual blocks | 3 |
| Total number of NN parameters | 36 |
| Spectral norm upper bound (SN-UB) | 0.7 |
| Latent dimensionality ($D_H$) | 2 |
| Number of inducing points ($M$) | 300 |
| Likelihood type | Gaussian |
| Output batch size ($P_b$) | 256 |
| Input batch size ($N_b$) | 24 |
| Optimiser | Adam, $lr = 0.01$ |
| Training epochs | 1000 |

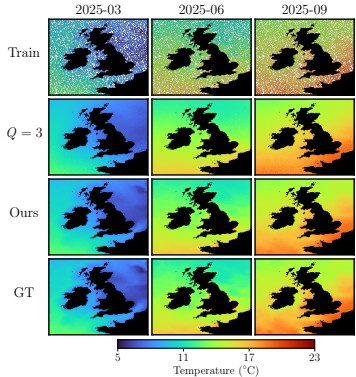

**Figure 8.** Predictions of GS-LVMOGP with $Q = 3$ and T-LVMOGP for output extrapolation task.

### F.4.2. COPERNICUS MARINE

The Copernicus Marine dataset [5] is derived from the Operational Mercator global ocean analysis and forecast system, which assimilates satellite and in-situ observations into numerical models at a horizontal resolution of $1/12°$. The experimental settings are listed in Table 20. We use GS-LVMOGP with different $Q$ values as baselines, and the hyperparameters are set to $D_H = 2$, $M_H = 30$, and $M_X = 10$. For both GS-LVMOGP and T-LVMOGP, we specify the prior mean of the latent variables as the associated spatial coordinates and fix the prior covariance at 0.01:

$$p(\mathbf{h}_p) = \mathcal{N}(\mathbf{h}_p \mid \mathrm{sc}_p, 0.01 \cdot \mathbf{I}), \text{ where } \mathrm{sc}_p \text{ refers to the spatial coordinates associated with the } p\text{-th output.}$$

The baseline models are trained using Adam with a learning rate of 0.01 for 1000 epochs. Regarding predictions on new outputs, we draw $S = 20$ samples $\mathbf{h}_{p_*} \sim \mathcal{N}(\mathbf{h}_{p_*} \mid \mathrm{sc}_{p_*}, 0.1 \cdot \mathbf{I})$, and use Monte Carlo approach for predicting on new spatial location with coordinates $\mathrm{sc}_{p_*}$. Note that we inflate prior variance at test time for robustness.

For T-LVMOGP, the prediction performance with different SN-UB values is listed in Table 21. The MSE remains invariant across different SN-UB settings. The optimal NLL is achieved at an SN-UB value of 0.7.

**Table 21.** Performance of T-LVMOGP with different SN-UB values on the Copernicus Marine dataset.

| SN-UB | MSE ↓ | NLL ↓ |
|---|---|---|
| 0.5 | $0.029 \pm 0.001$ | $-0.434 \pm 0.013$ |
| 0.6 | $0.029 \pm 0.000$ | $-0.438 \pm 0.010$ |
| 0.7 | $0.029 \pm 0.000$ | $\mathbf{-0.439 \pm 0.011}$ |
| 0.8 | $0.029 \pm 0.001$ | $-0.433 \pm 0.018$ |

### F.5. Spatial Transcriptomics

Spatial transcriptomics (Ståhl et al., 2016) provides a high-resolution view of gene expression while preserving tissue spatial topology, yielding rich yet challenging high-dimensional data. Applying machine learning models is vital to

---

[5] https://data.marine.copernicus.eu/product/GLOBAL_ANALYSISFORECAST_PHY_001_024/description

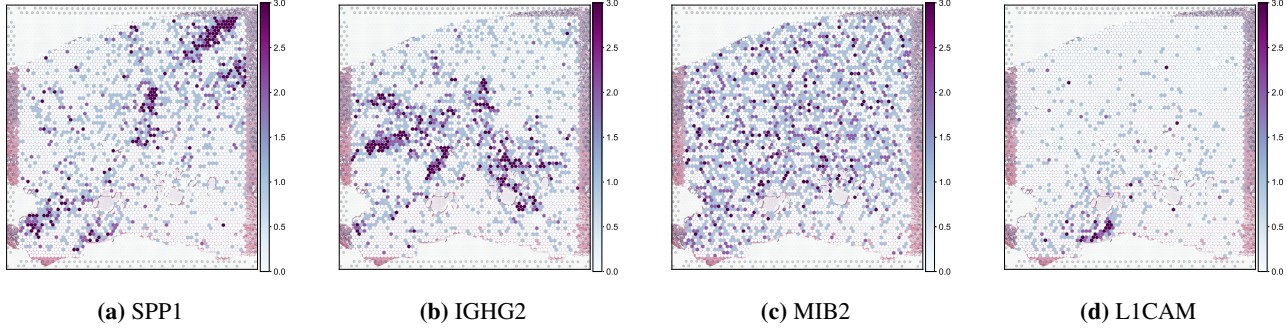

**(a)** SPP1      **(b)** IGHG2      **(c)** MIB2      **(d)** L1CAM

**Figure 9.** Spatial plots of gene expressions for 4 selected genes.

**Table 22.** Experimental settings of the Spatial Transcriptomics experiment.

| Setting | Value |
|---|---|
| Number of residual blocks | 3 |
| Total number of NN parameters | 90 |
| Latent dimensionality ($D_H$) | 3 |
| Number of inducing points ($M$) | 500 |
| Likelihood type | ZINB |
| Michaelis–Menten constant ($k_m$) | 0.1 |
| Scale factor | 1.0 |
| Output batch size ($P_b$) | 64 |
| Input batch size ($N_b$) | 256 |
| Optimiser | Adam, $lr = 0.0005$ |
| Training epochs | 300 |

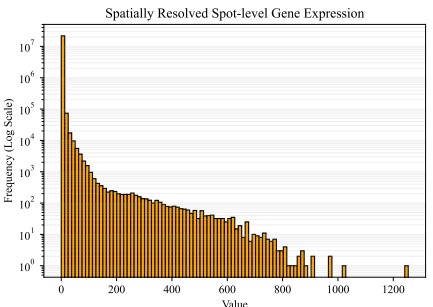

**Figure 10.** Histogram of the spatially resolved gene expression values in Log Scale. This dataset is highly sparse, with over $75\%$ of values being zero, and exhibits significant overdispersion, with extreme values exceeding $1, 200$.

disentangle the resulting spatial variability, enabling the discovery of distinct tissue sub-populations and disease markers. Figure 9 illustrates spatial gene expressions for 4 selected genes in the 10x Genomics Visium human prostate cancer dataset (10XGenomics2025). The experimental settings are listed in Table 22. We use GS-LVMOGP with varying $Q$ as baselines, with hyperparameters set to $D_H = 3$, $M_H = 20$, and $M_X = 25$. For both GS-LVMOGP and T-LVMOGP, the PCA vectors computed for each gene are established as the prior means of the latent variables. For ZINB likelihood, as mentioned in Appendix E, we utilise the positively transformed latent GP function $f$ times a positive scaling factor to model the distribution's mean, and in our experiments, we choose the `softplus` function for instantiating this positive transformation.

## F.6. Ablation Study and Analysis

Table 23 and Table 24 provide the test MSE of ablation studies regarding spectral normalisation and neural network, respectively. For the EEG experiment, we notice that T-LVMOGP with identity mapping achieved better MSE compared to neural networks. We attribute this phenomenon to the small size of the dataset (only $1, 492$ training data points), where simpler models often perform well.

**Table 23.** Test MSE comparison of T-LVMOGP with and without spectral normalisation.

| Dataset | w/o SN | w/ SN |
|---|---|---|
| EEG | $0.518 \pm 0.179$ | $\mathbf{0.115 \pm 0.025}$ |
| SARCOS | $0.317 \pm 0.447$ | $\mathbf{0.022 \pm 0.000}$ |
| ERA5 (random) | $0.003 \pm 0.001$ | $\mathbf{0.002 \pm 0.000}$ |
| ERA5 (block-wise) | $0.006 \pm 0.004$ | $\mathbf{0.003 \pm 0.000}$ |
| Copernicus Marine | $0.031 \pm 0.002$ | $\mathbf{0.029 \pm 0.000}$ |

**Table 24.** Test MSE comparison of T-LVMOGP with and without neural network.

| Dataset | w/o NN | w/ NN |
|---|---|---|
| EEG | $\mathbf{0.096 \pm 0.023}$ | $0.115 \pm 0.025$ |
| SARCOS | $0.030 \pm 0.000$ | $\mathbf{0.022 \pm 0.000}$ |
| ERA5 (random) | $0.002 \pm 0.002$ | $0.002 \pm 0.000$ |
| ERA5 (block-wise) | $0.003 \pm 0.001$ | $0.003 \pm 0.000$ |
| Copernicus Marine | $0.030 \pm 0.000$ | $\mathbf{0.029 \pm 0.000}$ |

We conduct additional ablation studies of T-LVMOGP on the SARCOS dataset by varying $D_H$ and $M$, while keeping all other hyperparameters the same as those used in Section 5.2 of the paper. The results are reported in Table 25 and Table 26. In terms of $D_H$, we find that varying $D_H$ has only a limited effect on test predictive performance, with $D_H = 7$ yielding the best overall result. As for $M$, varying the number of inducing points from 100 to 300 shows a consistent improvement in predictive performance as $M$ increases, consistent with our expectations.

**Table 25.** Performance comparison of T-LVMOGP on the SARCOS dataset by varying $D_H$ from 1 to 9.

| $D_H$ | 1 | 3 | 5 | 7 | 9 |
|---|---|---|---|---|---|
| MSE ($\downarrow$) | $0.024 \pm 0.001$ | $0.024 \pm 0.000$ | $\mathbf{0.022 \pm 0.000}$ | $\mathbf{0.022 \pm 0.000}$ | $0.023 \pm 0.000$ |
| NLL ($\downarrow$) | $-0.462 \pm 0.012$ | $-0.463 \pm 0.006$ | $-0.485 \pm 0.009$ | $\mathbf{-0.493 \pm 0.009}$ | $-0.481 \pm 0.008$ |

**Table 26.** Performance comparison of T-LVMOGP on the SARCOS dataset by varying $M$ from 100 to 300.

| $D_H$ | 100 | 150 | 200 | 250 | 300 |
|---|---|---|---|---|---|
| MSE ($\downarrow$) | $0.025 \pm 0.001$ | $0.024 \pm 0.001$ | $0.022 \pm 0.000$ | $0.022 \pm 0.000$ | $\mathbf{0.021 \pm 0.002}$ |
| NLL ($\downarrow$) | $-0.429 \pm 0.012$ | $-0.459 \pm 0.016$ | $-0.485 \pm 0.009$ | $-0.496 \pm 0.013$ | $\mathbf{-0.505 \pm 0.029}$ |

### F.7. Optimisation with Tighter Variational Bounds

Table 27 reports the test MSE comparison between T-LVMOGP models optimised using standard and tighter variational bounds. In the Spatial Transcriptomics experiment, we do not observe a clear benefit from optimising the "tighter" variational bound. We attribute this to the following reasons (1) the limited additional flexibility introduced for non-Gaussian likelihoods: relative to the standard SVGP bound, the new objective augments $q(f \mid \mathbf{u})$ with only a single extra parameter (Titsias, 2025) (the spherical $\mathbf{V}$, see Appendix G.2), which may be insufficient to yield a measurable improvement. (2) Moreover, there is no general theoretical guarantee that the new objective is strictly tighter than the standard SVGP bound, which is different from the Gaussian likelihood case. (3) Finally, because the expected log-likelihood term is typically intractable under non-Gaussian likelihoods and must be approximated (e.g., via Gaussian-Hermite quadrature or Monte Carlo), it becomes difficult in practice to ascertain which bound is effectively tighter throughout optimisation.

**Table 27.** Test MSE comparison of T-LVMOGP with standard and tighter variational bounds.

| **Dataset** | **Standard** | **Tighter** |
|---|---|---|
| SARCOS | $0.022 \pm 0.000$ | $0.022 \pm 0.000$ |
| Copernicus Marine | $0.029 \pm 0.000$ | $\mathbf{0.028 \pm 0.001}$ |
| Spatial Transcriptomics | $\mathbf{9.189 \pm 0.342}$ | $9.203 \pm 0.332$ |

## G. Review of SVGP with Tighter Variational Bounds

### G.1. Standard Bounds

SVGP leverages a small set of $M(\ll N)$ *inducing points* to approximate the true posterior. The inducing locations and the related inducing variables are denoted as follows, respectively

$$\mathbf{Z} = [\mathbf{z}_m]_{m=1}^M \in \mathbb{R}^{M \times D}, \quad \mathbf{u} = [u_m]_{m=1}^M \in \mathbb{R}^M.$$

To keep the GP prior untouched, the joint prior distribution of $\mathbf{f}$ and $\mathbf{u}$ is formulated by

$$\begin{bmatrix} \mathbf{f} \\ \mathbf{u} \end{bmatrix} \sim \mathcal{N}\left( \begin{bmatrix} \mathbf{f} \\ \mathbf{u} \end{bmatrix} \mid \mathbf{0}, \begin{bmatrix} \mathbf{K_{XX}} & \mathbf{K_{XZ}} \\ \mathbf{K_{ZX}} & \mathbf{K_{ZZ}} \end{bmatrix} \right).$$

Therefore, we have the marginal prior over $\mathbf{u}$ and the conditional prior $\mathbf{f}|\mathbf{u}$:

$$p(\mathbf{u}) = \mathcal{N}(\mathbf{u} \mid \mathbf{0}, \mathbf{K_{ZZ}}),$$
$$p(\mathbf{f} \mid \mathbf{u}) = \mathcal{N}(\mathbf{f} \mid \mathbf{K_{XZ}} \mathbf{K_{ZZ}}^{-1} \mathbf{u}, \mathbf{K_{XX}} - \mathbf{K_{XZ}} \mathbf{K_{ZZ}}^{-1} \mathbf{K_{ZX}}).$$

SVGP approximates the exact posterior $p(\mathbf{f}, \mathbf{u} \mid \mathbf{y})$ by a variational distribution $q(\mathbf{f}, \mathbf{u})$ through the minimisation of $\mathrm{KL}[q(\mathbf{f}, \mathbf{u}) \mid\mid p(\mathbf{f}, \mathbf{u} \mid \mathbf{y})]$. A critical assumption is the following choice:

$$q(\mathbf{f}, \mathbf{u}) = p(\mathbf{f} \mid \mathbf{u})q(\mathbf{u}), \quad q(\mathbf{u}) \text{ is an optimisable } M\text{-dimensional variational distribution.}$$

The ELBO is as follows:

$$
\begin{aligned}
\log p(\mathbf{y}) &\geq \int p(\mathbf{f} \mid \mathbf{u})q(\mathbf{u}) \log \frac{p(\mathbf{y} \mid \mathbf{f})p(\mathbf{f} \mid \mathbf{u})p(\mathbf{u})}{p(\mathbf{f} \mid \mathbf{u})q(\mathbf{u})} \, d\mathbf{f} \, d\mathbf{u} \\
&= \int q(\mathbf{u}) \log \frac{\exp\{\int p(\mathbf{f} \mid \mathbf{u}) \log p(\mathbf{y} \mid \mathbf{f}) \, d\mathbf{f}\}p(\mathbf{u})}{q(\mathbf{u})} \, d\mathbf{u} \\
&= -\mathrm{KL}\left[ q(\mathbf{u}) \mid\mid \frac{\exp\{\int p(\mathbf{f} \mid \mathbf{u}) \log p(\mathbf{y} \mid \mathbf{f}) \, d\mathbf{f}\}p(\mathbf{u})}{\int \exp\{\int p(\mathbf{f} \mid \mathbf{u}) \log p(\mathbf{y} \mid \mathbf{f}) \, d\mathbf{f}\}p(\mathbf{u}) \, d\mathbf{u}} \right] \\
&\quad + \log \int \exp\left\{ \int p(\mathbf{f} \mid \mathbf{u}) \log p(\mathbf{y} \mid \mathbf{f}) \, d\mathbf{f} \right\} p(\mathbf{u}) \, d\mathbf{u}.
\end{aligned}
$$

**Titsias' collapsed bounds**   If we optimise over $q(\mathbf{u})$ and obtain the optimal choice $q^*(\mathbf{u}) \propto \exp\{\int p(\mathbf{f} \mid \mathbf{u}) \log p(\mathbf{y} \mid \mathbf{f}) \, d\mathbf{f}\}p(\mathbf{u})$, and substitute this optimal distribution to the above ELBO, we arrive at the so-called *collapsed bound* (*Titsias, 2009*):

$$\log p(\mathbf{y}) \geq \mathcal{L} = \log \int \exp \left\{ \int p(\mathbf{f} \mid \mathbf{u}) \log p(\mathbf{y} \mid \mathbf{f}) d\mathbf{f} \right\} p(\mathbf{u}) \, d\mathbf{u}.$$

For the Gaussian likelihood, the above bound takes the form:

$$\mathcal{L} = \underbrace{\log \mathcal{N}\left(\mathbf{y} \mid \mathbf{0}, \mathbf{Q_{XX}} + \sigma_y^2 \mathbf{I}\right)}_{\text{DTC log lik}} - \underbrace{\frac{1}{2\sigma_y^2} \operatorname{tr}\left(\mathbf{K_{XX}} - \mathbf{Q_{XX}}\right)}_{\text{trace term}}, \tag{4}$$

where $\mathbf{Q_{XX}} = \mathbf{K_{XZ}}\mathbf{K_{ZZ}^{-1}}\mathbf{K_{ZX}}$ is the $M$-rank Nyström matrix. The inducing points $\mathbf{Z}$, kernel parameters $\theta$ and likelihood noise $\sigma_y^2$ can be optimised by maximising the bound, which requires $\mathcal{O}(NM^2)$.

**Hensman's uncollapsed bounds**   To support mini-batch training, instead of using "optimal" $q(\mathbf{u})$, Hensman et al. (2013) maintain an explicit representation of these inducing variables. They introduce a variational distribution $q(\mathbf{u}) = \mathcal{N}(\mathbf{u} \mid \mathbf{m}, \mathbf{S})$, where $\mathbf{m}$ and $\mathbf{S}$ are variational parameters. Thus,

$$
\begin{aligned}
\log p(\mathbf{y}) &\geq \int q(\mathbf{f}) \log p(\mathbf{y} \mid \mathbf{f}) \, d\mathbf{f} - \mathrm{KL}[q(\mathbf{u}) \mid\mid p(\mathbf{u})] \\
&= \sum_{n=1}^{N} \mathbb{E}_{q(f_i)}\left[\log p(y_i \mid f_i)\right] - \mathrm{KL}[q(\mathbf{u}) \mid\mid p(\mathbf{u})],
\end{aligned} \tag{5}
$$

where $q(f_i) = \int p(\mathbf{f} \mid \mathbf{u})q(\mathbf{u}) \, d\mathbf{f}_{-i} \, d\mathbf{u}$ is the marginal over $f(\mathbf{x}_i)$ with respect to $q(\mathbf{f}, \mathbf{u})$, calculated as follows:

$$q(f_i) = \mathcal{N}(f_i \mid \mathbf{k}_{\mathbf{x}_i \mathbf{Z}}\mathbf{K_{ZZ}^{-1}}\mathbf{m}, \, k_{\mathbf{x}_i \mathbf{x}_i} - q_{\mathbf{x}_i \mathbf{x}_i} + \mathbf{k}_{\mathbf{x}_i \mathbf{Z}}\mathbf{K_{ZZ}^{-1}}\mathbf{S}\mathbf{K_{ZZ}^{-1}}\mathbf{k}_{\mathbf{Z}\mathbf{x}_i}).$$

For the Gaussian likelihood, the log-likelihood term in Equation (5) has an analytical formula, which results in the following bound:

$$
\begin{aligned}
\log p(\mathbf{y}) &\geq \sum_{i=1}^{N} \left\{ \mathbb{E}_{q(\mathbf{u})}\left[\log \mathcal{N}(y_i \mid \mathbf{k}_{\mathbf{x}_i \mathbf{Z}}\mathbf{K_{ZZ}^{-1}}\mathbf{u}, \sigma_y^2)\right] - \frac{1}{2\sigma_y^2}\left(k_{\mathbf{x}_i \mathbf{x}_i} - q_{\mathbf{x}_i \mathbf{x}_i}\right) \right\} - \mathrm{KL}[q(\mathbf{u}) \mid\mid p(\mathbf{u})] \\
&= \sum_{n=1}^{N} \left\{ \log \mathcal{N}(y_i \mid \mathbf{k}_{\mathbf{x}_i \mathbf{Z}}\mathbf{K_{ZZ}^{-1}}\mathbf{m}, \sigma_y^2) - \frac{1}{2\sigma_y^2}\operatorname{tr}(\mathbf{S}\mathbf{\Lambda}_i) - \frac{1}{2\sigma_y^2}\left(k_{\mathbf{x}_i \mathbf{x}_i} - q_{\mathbf{x}_i \mathbf{x}_i}\right) \right\} - \mathrm{KL}[q(\mathbf{u}) \mid\mid p(\mathbf{u})],
\end{aligned} \tag{6}
$$

where $\mathbf{\Lambda}_i = \mathbf{K_{ZZ}^{-1}}\mathbf{k}_{\mathbf{Z}\mathbf{x}_i}\mathbf{k}_{\mathbf{x}_i \mathbf{Z}}\mathbf{K_{ZZ}^{-1}}$.

## G.2. Tighter Bounds

The essential design of the variational distribution in SVGP is that $q(\mathbf{f}, \mathbf{u}) = p(\mathbf{f} \mid \mathbf{u})q(\mathbf{u})$, where the variational conditional $q(\mathbf{f} \mid \mathbf{u})$ matches the prior conditional $p(\mathbf{f} \mid \mathbf{u})$. Recently, Titsias (2025) and Bui et al. (2025) have proposed a new form of variational distribution:

$$q(f_*) = p(f_* \mid \mathbf{f}, \mathbf{u})q(\mathbf{f} \mid \mathbf{u})q(\mathbf{u}), \quad q(\mathbf{f} \mid \mathbf{u}) = \mathcal{N}(\mathbf{f} \mid \mathbf{K_{XZ}K_{ZZ}^{-1}u}, \mathbf{D_{XX}^{1/2}VD_{XX}^{\top/2}}), \quad q(\mathbf{u}) = \mathcal{N}(\mathbf{u} \mid \mathbf{m}, \mathbf{S}),$$

where $\mathbf{V}$ is a diagonal matrix with positive diagonal elements $v_1, v_2, ..., v_N$ and $\mathbf{D_{XX}} = \mathbf{K_{XX}} - \mathbf{Q_{XX}}$.

$$\log p(\mathbf{y}) \geq \int q(\mathbf{f} \mid \mathbf{u})q(\mathbf{u}) \log \frac{p(\mathbf{y} \mid \mathbf{f})p(\mathbf{f} \mid \mathbf{u})p(\mathbf{u})}{q(\mathbf{f} \mid \mathbf{u})q(\mathbf{u})} \, d\mathbf{f} \, d\mathbf{u}$$

$$= -\operatorname{KL}[q(\mathbf{u}) \mid\mid p(\mathbf{u})] - \int q(\mathbf{u}) \operatorname{KL}[q(\mathbf{f} \mid \mathbf{u}) \mid\mid p(\mathbf{f} \mid \mathbf{u})] + \int q(\mathbf{f} \mid \mathbf{u})q(\mathbf{u}) \log p(\mathbf{y} \mid \mathbf{f}) \, d\mathbf{f} \, d\mathbf{u}. \quad (7)$$

Thanks to the structure of the variational distribution, the middle term can be simplified to

$$-\int q(\mathbf{u}) \operatorname{KL}[q(\mathbf{f} \mid \mathbf{u}) \mid\mid p(\mathbf{f} \mid \mathbf{u})] = -\frac{1}{2}\operatorname{tr}(\mathbf{V}) + \frac{1}{2}\log|\mathbf{V}| + \frac{N}{2} = \frac{1}{2}\sum_{n=1}^{N}\left[1 + \log(v_n) - v_n\right]. \quad (8)$$

**Proposition G.1** ((Titsias, 2025), Lemma 3.2). *For the Gaussian likelihood,*

$$\mathbb{E}_{q(\mathbf{f}\mid\mathbf{u})}\left[\log p(\mathbf{y} \mid \mathbf{f})\right] = \log \mathcal{N}(\mathbf{y} \mid \mathbf{K_{XZ}K_{ZZ}^{-1}u}, \sigma_y^2 \mathbf{I}) - \frac{1}{2}\sum_{n=1}^{N}\frac{v_n d_n}{\sigma_y^2}, \quad (9)$$

*where $d_n$ is the $n$-th element in the diagonal of $\mathbf{D_{XX}}$, $d_n = k_{\mathbf{x}_n \mathbf{x}_n} - \mathbf{k}_{\mathbf{x}_n \mathbf{Z}}\mathbf{K_{ZZ}^{-1}}\mathbf{k}_{\mathbf{Z}\mathbf{x}_n}$. By combining Equation (8) and Equation (9), the ELBO can be written as*

$$\log p(\mathbf{y}) \geq \int q(\mathbf{u}) \log \frac{\mathcal{N}(\mathbf{y} \mid \mathbf{K_{XZ}K_{ZZ}^{-1}u}, \sigma_y^2 \mathbf{I})p(\mathbf{u})}{q(\mathbf{u})} \, d\mathbf{u} - \frac{1}{2}\sum_{n=1}^{N}\left\{\frac{v_n d_n}{\sigma_y^2} - 1 - \log(v_n) + v_n\right\}. \quad (10)$$

**Collapsed Bound and Optimal V** For the Gaussian likelihood, similar to Titsias' collapsed bounds in SVGP (Titsias, 2009), we can obtain the optimal form for $q(\mathbf{u})$, thus

$$\log p(\mathbf{y}) \geq \log \mathcal{N}(\mathbf{y} \mid 0, \mathbf{Q_{XX}} + \sigma_y^2 \mathbf{I}) - \frac{1}{2}\sum_{n=1}^{N}\left[\frac{v_n d_n}{\sigma_y^2} - 1 - \log(v_n) + v_n\right]. \quad (11)$$

Setting the partial derivative w.r.t. $v_n$ to 0, we arrive at $v_n = \frac{\sigma_y^2}{d_n + \sigma_y^2} \leq 1$ and the following bound:

$$\log p(\mathbf{y}) \geq -\frac{N}{2}\log(2\pi) - \frac{1}{2}\mathbf{y}^\top\left(\mathbf{Q_{XX}} + \sigma_y^2 \mathbf{I}\right)^{-1}\mathbf{y} - \frac{1}{2}\log|\mathbf{Q_{XX}} + \sigma_y^2 \mathbf{I}| - \frac{1}{2}\sum_{n=1}^{N}\log\left(1 + \frac{d_n}{\sigma_y^2}\right). \quad (12)$$

*Remark* G.2. When $\mathbf{V}$ is the identity matrix, that is $v_n = 1 \; \forall n$, Equation (11) becomes identical to Titsias' collapsed bound Equation (4). It is noticeable that Equation (12) is tighter (larger) than Equation (4) due to the inequality $\log(1 + x) \leq x$.

**Uncollapsed Bound for the Gaussian likelihood** Starting from Equation (10), plugging in optimal $v_n = \frac{\sigma_y^2}{d_n + \sigma_y^2}$,

$$\log p(\mathbf{y}) \geq \sum_{n=1}^{N}\left\{\mathbb{E}_{q(\mathbf{u})}[\log \mathcal{N}(y_n \mid \mathbf{k}_{\mathbf{x}_n \mathbf{Z}}\mathbf{K_{ZZ}^{-1}u}, \sigma_y^2)] - \frac{1}{2}\log\left(1 + \frac{d_n}{\sigma_y^2}\right)\right\} - \operatorname{KL}[q(\mathbf{u}) \mid\mid p(\mathbf{u})]$$

$$\overset{(6)}{=} \underbrace{\sum_{n=1}^{N}\left\{\mathbb{E}_{q(\mathbf{u})}[\log \mathcal{N}(y_n \mid \mathbf{k}_{\mathbf{x}_n \mathbf{Z}}\mathbf{K_{ZZ}^{-1}u}, \sigma_y^2)] - \frac{d_n}{2\sigma_y^2}\right\} - \operatorname{KL}[q(\mathbf{u}) \mid\mid p(\mathbf{u})]}_{\text{Hensman's uncollapsed bound for the Gaussian likelihood}} + \underbrace{\frac{1}{2}\sum_{n=1}^{N}\left\{\frac{d_n}{\sigma_y^2} - \log\left(1 + \frac{d_n}{\sigma_y^2}\right)\right\}}_{\text{Correction Term } \Delta \geq 0}$$

$$= \underbrace{\sum_{n=1}^{N}\mathbb{E}_{p(f_n\mid\mathbf{u})q(\mathbf{u})}\left[\log \mathcal{N}(y_n \mid f_n, \sigma_y^2)\right] - \operatorname{KL}[q(\mathbf{u}) \mid\mid p(\mathbf{u})]}_{\text{Another notation}} + \underbrace{\frac{1}{2}\sum_{n=1}^{N}\left\{\frac{d_n}{\sigma_y^2} - \log\left(1 + \frac{d_n}{\sigma_y^2}\right)\right\}}_{\text{Correction Term } \Delta}.$$

**Uncollapsed Bound for non-Gaussian likelihood**   Based on Equation (7) and Equation (8), we have the following result:

$$\log p(\mathbf{y}) \geq - \text{KL}[q(\mathbf{u}) \,||\, p(\mathbf{u})] + \frac{1}{2} \sum_{n=1}^{N} [1 + \log(v_n) - v_n] + \sum_{n=1}^{N} \int q(f_n \mid \mathbf{u}) q(\mathbf{u}) \log p(y_n \mid f_n) \, df_n \, d\mathbf{u}.$$

The marginalisation $q(f_n \mid \mathbf{u}) = \int q(\mathbf{f} \mid \mathbf{u}) d\mathbf{f}_{-n}$ cannot be trivially expressed. Titsias (2025) use a simplified version of $q(\mathbf{f} \mid \mathbf{u})$, in which they choose a spherical $\mathbf{V} = v\mathbf{I}$ with $v > 0$. The ELBO becomes

$$\log p(\mathbf{y}) \geq - \text{KL}[q(\mathbf{u}) \,||\, p(\mathbf{u})] + \frac{N}{2}[1 + \log(v) - v] + \sum_{n=1}^{N} \underbrace{\mathbb{E}_{q(f_n)}[\log p(y_n \mid f_n)]}_{\text{Monte Carlo, Quadrature ...}},$$

where the marginal is $q(f_n) = \mathcal{N}(f_n \mid \mathbf{k}_{\mathbf{x}_n \mathbf{Z}} \mathbf{K}_{\mathbf{ZZ}}^{-1} \mathbf{m}, v d_n + \mathbf{k}_{\mathbf{x}_n \mathbf{Z}} \mathbf{K}_{\mathbf{ZZ}}^{-1} \mathbf{S} \mathbf{K}_{\mathbf{ZZ}}^{-1} \mathbf{k}_{\mathbf{Z} \mathbf{x}_n})$.

## H. Further Related Work

There are more works about MOGPs in the literature. Yu et al. (2005) exploit the equivalence between parametric linear models and GPs for multi-task learning in a hierarchical Bayesian framework. Semiparametric Latent Factor Models (SLFM) can be viewed as a special case of LMC (Teh et al., 2005). Latent Force Models (LFMs) (Alvarez et al., 2009) induce dependencies by passing latent GP forces through mechanistic differential equations, yielding physically interpretable multi-output kernels. Melkumyan & Ramos (2011) develop multi-task covariance functions by combining stationary kernels via Fourier arguments under the process convolution framework. Collaborative MOGP (Nguyen & Bonilla, 2014) represents output functions as a combination of two elements: projections from latent GPs shared across all outputs and individual GPs capturing the output-specific residuals. Kaiser et al. (2018) propose a hierarchical MOGP model, where Convolution Process layers are embedded in the middle of Deep Gaussian Processes (DGP) (Damianou & Lawrence, 2013) for explicitly modelling output dependencies. To enhance expressivity in multichannel signal modelling, the Multi-Output Spectral Mixture kernel (MOSM) (Parra & Tobar, 2017; Altamirano & Tobar, 2022) is introduced, which is effective for modelling phase differences and delays among outputs. Óscar García-Hinde et al. (2022) reformulate the MOGP problem as a sequence of conditioned univariate GPs by augmenting the input space with previous output values—a strategy similarly adopted in autoregressive GP modelling (Requeima et al., 2019). While MOGP literature predominantly focuses on modelling output correlations through complex covariance structures, Fortuin et al. (2019) demonstrate that meta-learning deep mean functions can be a competitive strategy for knowledge transfer in low-data regimes.

In terms of applications and specific extensions, Zhao & Sun (2016) propose a convolved MOGP model for complex dynamical systems such as motion capture, traffic flow and robot inverse dynamics. Yousefi et al. (2019) construct an MOGP model based on LMC for modelling data points that have been aggregated at different input scales. To improve heterogeneous modelling of MOGP, Giraldo & Alvarez (2021) propose a fully natural gradient scheme applicable for both LMC and process convolution. López-Lopera et al. (2022) apply an MOGP model with a separable kernel to surrogate modelling of coastal flood hazard assessment. Yoon et al. (2022) introduce a specialised variant of LMC that decomposes input effects on outputs into components shared across or specific to tasks and samples. Ma & Álvarez (2023) introduce a multi-output multi-class classification model based on LMC. Ma et al. (2023) extend LV-MOGP for modelling hierarchical datasets, where several replicates are observed for each output.

## I. Complexity Analysis of Baseline Models

We provide a comparative analysis of the baseline models (SV-LMC (Van der Wilk et al., 2020), SGPRN (Li et al., 2020), OILMM (Bruinsma et al., 2020), G-MOGP (Dai et al., 2024), and GS-LVMOGP (Jiang et al., 2025)), focusing on their respective model sizes and computational costs based on our implementation. Notably, in the following evaluation, kernel parameters are excluded from the parameter enumeration.

### I.1. SV-LMC

The total number of trainable parameters is given by

$$LP + LMD_X + LM + LM^2,$$

where $LP$ denotes the number of parameters in the LMC mixing coefficients (projection matrix). The remaining terms correspond to the $L$ latent SVGPs: $LMD_X$ represents the inducing locations (where $D_X$ is the input dimension), $LM$ accounts for the variational mean parameters, and $LM^2$ represents the elements of the Cholesky factor of the variational covariance matrix. The computational complexity is formulated as

$$\mathcal{O}(N_b LM^2 + LM^3 + N_b P_b L).$$

Here, the term $\mathcal{O}(N_b LM^2 + LM^3)$ captures the complexity associated with the $L$ latent SVGPs. The term $N_b P_b L$ corresponds to the computational cost of the linear projection from the latent space to the observation space for a mini-batch with $N_b$ inputs and $P_b$ output dimensions.

### I.2. SGPRN

The total number of trainable parameters consists of:

$$LN + 2N^2 + 2L^2 + LNP + P^2,$$

where the terms $LN + N^2 + L^2$ correspond to the parameters of the variational distribution $q(\mathbf{F})$, and $LNP + N^2 + L^2 + P^2$ correspond to those of the variational distribution $q(\mathcal{W})$. In cases where the outputs form a tensor of dimensions $P_1 \times P_2 \times \cdots \times P_t$ (such that $P = \prod_{i=1}^{t} P_i$), the $P^2$ term is replaced by $\sum_{i=1}^{t} P_i^2$. A comprehensive review of this model is provided in Appendix J.1. The computational complexity is given by

$$\mathcal{O}(N^3 + LN^2 P + N_b L^2 P).$$

Here, the $\mathcal{O}(N^3)$ term arises from the Cholesky decomposition of $\mathbf{L}_{\hat{f}}$; $\mathcal{O}(LN^2 P)$ results from computing $\mathbf{U}_1 \mathbf{U}_1^\top$; and $\mathcal{O}(N_b L^2 P)$ represents the cost of computing $\mathbb{E}_q[\mathbf{W}_n]^\top \mathbb{E}_q[\mathbf{W}_n]$ for a mini-batch of size $N_b$. The presence of the $\mathcal{O}(N^3)$ term limits the tractability of this model for datasets with a massive number of inputs.

### I.3. OILMM

The total number of trainable parameters is given by

$$2L + LMD_X + LM + LM^2.$$

Here, the two $L$ terms denote the sum of the number of parameters in the diagonal of the basis matrix $\mathbf{H}$ and the additive noise term for the latent GPs. The term $LMD_X$ corresponds to the inducing locations, while $LM + LM^2$ accounts for the variational parameters (specifically, the variational mean and the Cholesky factor of the variational covariance matrix). More details about this model are provided in Appendix J.2.

The computational complexity for OILMM is:

$$\mathcal{O}(N_b LM^2 + LM^3 + N_b P_b L + L^3 + L^2 P).$$

The initial term, $\mathcal{O}(N_b LM^2 + LM^3)$, represents the complexity associated with the latent SVGPs. The term $\mathcal{O}(N_b P_b L)$ quantifies the cost of projecting a mini-batch with $N_b$ inputs and $P_b$ outputs from the latent space to the observation space. The final terms, $\mathcal{O}(L^3 + L^2 P)$, arise from the construction of the projection matrix. In practice, outputs within each mini-batch are sorted and grouped by missing-data patterns prior to computation. Consequently, the effective computational cost is also influenced by the sparsity rate and the specific missingness patterns of the dataset.

### I.4. G-MOGP

The total number of trainable parameters is

$$2P^2 + MD_X P + MP + M^2 P.$$

Here, the two $P^2$ terms correspond to the attention weights and attention biases, respectively. The inducing points are defined specifically for every output, thus $MD_X P$ is the total number of parameters for inducing locations, $MP + M^2 P$ accounts for the variational distribution parameters. It is important to note that this approach necessitates the construction

of $P \times P$ matrices for attention weights and biases. This imposes a constraint on memory scalability as $P$ increases, particularly when compared to our proposed T-LVMOGP. The computational cost is

$$\mathcal{O}(N_b P_b M P + P_b M^3 + N_b P_b M^2).$$

The term $\mathcal{O}(N_b P_b M P)$ arises from computing the $P$ covariance matrices $\mathbf{K}_{\mathbf{X}_b \mathbf{Z}}$ and aggregating them via weighted averaging for the $P_b$ outputs. The final terms, $\mathcal{O}(P_b M^3 + N_b P_b M^2)$, represent the computational complexity associated with the $P_b$ independent SVGPs.

### I.5. GS-LVMOGP

The total number of model parameters is

$$Q D_H P + Q D_H^2 P + M_X M_H + M_X^2 + M_H^2 + Q M_H D_H + M_X D_X,$$

where $Q D_H P + Q D_H^2 P$ counts the variational parameters for latent variables, $M_X M_H$ is the number of parameters for the variational mean of inducing variables; $M_X^2$ (input space) and $M_H^2$ (latent space) are for the Cholesky factors of covariance matrices of variational distributions. The term $Q M_H D_H$ is the number of parameters of the inducing points in the latent space, while the $M_X D_X$ is for inducing points in the input space. The computational complexity of this model depends on the choice of $Q$. For $Q = 1$, the Cholesky factorisation of $\mathbf{K_{ff}} = \mathbf{K}_{\mathbf{H}^1 \mathbf{H}^1}^1 \otimes \mathbf{K}_{\mathbf{XX}}^1$ can be computed in $\mathcal{O}(M_X^3 + M_H^3)$ by exploiting the Kronecker product structure. For $Q > 1$, $\mathbf{K_{ff}}$ becomes $\sum_{q=1}^{Q} \mathbf{K}_{\mathbf{H}^q \mathbf{H}^q}^q \otimes \mathbf{K}_{\mathbf{XX}}^q \in \mathbb{R}^{M_X M_H \times M_X M_H}$, and the Cholesky has cost $\mathcal{O}(M_X^3 M_H^3)$. The overall computational complexity for $Q = 1$ is

$$\mathcal{O}(N_b P_b M_X^2 M_H^2 + M_X^3 + M_H^3),$$

the $\mathcal{O}(N_b P_b M_X^2 M_H^2)$ comes from $\mathbf{L}_{\mathbf{ZZ}}^{-1} \mathbf{K}_{\mathbf{ZX}_n}$, where $\mathbf{L}_{\mathbf{ZZ}}$ is the Cholesky factor of $\mathbf{K}_{\mathbf{ZZ}} \in \mathbb{R}^{M_X M_H \times M_X M_H}$, and $\mathbf{K}_{\mathbf{ZX}_n} \in \mathbb{R}^{M_X M_H \times N_b P_b}$ is the cross-covariance matrix between inducing points and mini-batch data points. For $Q > 1$, the computational cost is

$$\mathcal{O}(N_b P_b M_X^2 M_H^2 + M_X^3 M_H^3 + Q M_X^2 M_H^2).$$

The term $Q M_X^2 M_H^2$ is from summing $Q$ Kronecker products.

## J. Review of SGPRN and OILMM

In this section, we present a detailed review of the two scalable MOGP models adopted as baselines in this paper—namely, SGPRN (Li et al., 2020) and OILMM (Bruinsma et al., 2020)—to provide interested readers with additional background and context.

### J.1. Scalable Gaussian Process Regression Network (SGPRN)

Gaussian Process Regression Networks (GPRN) (Wilson et al., 2012) are flexible Bayesian models for multi-output regression. A critical bottleneck for GPRNs is the intractability of their inference. Existing methods use a fully factorised structure (Wilson et al., 2012) (or a mixture of such structures (Nguyen & Bonilla, 2013)) over all the outputs and latent functions for posterior approximation; however, this can miss the strong posterior dependencies among the latent variables (and projection weights) and hurt the inference quality. Li et al. (2020) propose tensorising the output space and introducing a matrix/tensor-normal posterior. This structure not only captures the strong posterior dependencies of the weights (or latent functions), but also requires significantly fewer covariance parameters. They train the model via ELBO, exploiting Kronecker product properties to accelerate the computation of the expensive log determinant and matrix inverse.

**GPRN** To model multiple outputs, GPRN first introduce a small set of $L$ latent functions, $\{f_1(\cdot), ..., f_L(\cdot)\}$, where each latent function $f_l(\cdot)$ is sampled from a GP prior. Next, GPRN introduces a $P \times L$ projection matrix $\mathbf{W}$, where each element $w_{ij}(1 \le i \le P, 1 \le j \le L)$ is also considered as a function of the input, and sampled from an independent GP prior. Given an input $\mathbf{x}$, the outputs are modelled by

$$\mathbf{y}(\mathbf{x}) = \mathbf{W}(\mathbf{x})[\mathbf{f}(\mathbf{x}) + \sigma_f \epsilon] + \sigma_y \mathbf{z},$$

where $\mathbf{f}(\mathbf{x}) = [f_1(\mathbf{x}), f_2(\mathbf{x}), ..., f_L(\mathbf{x})]^\top$, $\epsilon$ and $\mathbf{z}$ are random noise sampled from standard Gaussian distribution. GPRN accommodates input-dependent (i.e. non-stationary) correlations of the outputs. Given $\mathbf{W}(\cdot)$, the covariance of two outputs $y_i(\mathbf{x}_a)$ and $y_j(\mathbf{x}_b)$ is

$$k_{y_i, y_j}(\mathbf{x}_a, \mathbf{x}_b) = \sum_{l=1}^{L} w_{il}(\mathbf{x}_a) k_{\hat{f}_l}(\mathbf{x}_a, \mathbf{x}_b) w_{jl}(\mathbf{x}_b) + \delta_{ij} \delta_{ab} \sigma_y^2,$$

where $k_{\hat{f}_l}(\mathbf{x}_a, \mathbf{x}_b) = k_{f_l}(\mathbf{x}_a, \mathbf{x}_b) + \delta_{ab} \sigma_f^2$ and $k_{f_l}$ is the covariance (kernel) function for the latent function $f_l(\cdot)$. The output covariances are determined by the inputs via the projection weights $w_{il}(\mathbf{x}_a)$ and $w_{jl}(\mathbf{x}_b)$. Therefore, the model can adaptively capture the complex output correlations that vary across the input space. Following the original paper (Wilson et al., 2012), we assume all the latent functions share the same kernel $k_f(\cdot, \cdot)$ and parameters $\boldsymbol{\theta}_f$, and all the projection weights share the same kernel $k_w(\cdot, \cdot)$ and parameters $\boldsymbol{\theta}_w$. We write $\hat{f}_l(\mathbf{x}) = f_l(\mathbf{x}) + \sigma_f \epsilon$, where $\epsilon \sim \mathcal{N}(0, 1)$ and we denote $\hat{\mathbf{f}}_l = [\hat{f}_l(\mathbf{x}_1), \hat{f}_l(\mathbf{x}_2), ..., \hat{f}_l(\mathbf{x}_N)]^\top$ and projection matrix $\mathbf{w}_{ij} = [w_{ij}(\mathbf{x}_1), w_{ij}(\mathbf{x}_2), ..., w_{ij}(\mathbf{x}_N)]^\top$. $\mathbf{W}_n$ is $P \times L$, each $[\mathbf{W}_n]_{ij} = w_{ij}(\mathbf{x}_n)$, and $\mathbf{h}_n = [\hat{f}_1(\mathbf{x}_n), ..., \hat{f}_L(\mathbf{x}_n)]^\top$. The joint distribution then is

$$p(\mathbf{Y}, \{\mathbf{w}_{ij}\}, \{\hat{\mathbf{f}}_l\} \mid \mathbf{X}, \boldsymbol{\theta}_f, \boldsymbol{\theta}_w, \sigma_f^2, \sigma_y^2) = \prod_{l=1}^{L} \mathcal{N}(\hat{\mathbf{f}}_l \mid \mathbf{0}, \mathbf{L}_{\hat{f}}) \prod_{i=1}^{P} \prod_{j=1}^{L} \mathcal{N}(\mathbf{w}_{ij} \mid \mathbf{0}, \mathbf{L}_w) \prod_{n=1}^{N} \mathcal{N}(\mathbf{y}_n \mid \mathbf{W}_n \mathbf{h}_n, \sigma_y^2 \mathbf{I}).$$

**Matrix and Tensor Normal Posteriors**    To capture posterior dependency between the latent functions, they use a matrix normal distribution as the joint variational posterior for the $N \times L$ function values $\mathbf{F} = [\hat{\mathbf{f}}_1, ..., \hat{\mathbf{f}}_L]$,

$$q(\mathbf{F}) = \mathcal{MN}(\mathbf{F} \mid \mathbf{M}; \boldsymbol{\Sigma}, \boldsymbol{\Omega}) = \mathcal{N}(\text{vec}(\mathbf{F}) \mid \text{vec}(\mathbf{M}), \boldsymbol{\Sigma} \otimes \boldsymbol{\Omega}),$$

where $\boldsymbol{\Sigma} \in \mathbb{R}^{N \times N}$ and $\boldsymbol{\Omega} \in \mathbb{R}^{L \times L}$ are row and column covariance matrices, respectively.

Next, they consider the variational posterior of the projection weights $\{w_{ij}\}$. To obtain a joint posterior yet with compact parameterisation, they tensorise the $P$-dimensional output space into an $M$-mode tensor space [6], $p_1 \times p_2 \times \cdots \times p_M$ where $P = \prod_{m=1}^{M} p_m$. For simplicity, they set $p_1 = \cdots = p_M = p = \sqrt[M]{P}$. Then they organise all the weights into an $N \times L \times p_1 \times p_2 \times \cdots \times p_M$ tensor $\mathcal{W}$. They introduce the tensor normal distribution:

$$q(\mathcal{W}) = \mathcal{TN}(\mathcal{W} \mid \mathcal{U}; \boldsymbol{\Gamma}_1, \boldsymbol{\Gamma}_2, \cdots, \boldsymbol{\Gamma}_{M+2}) = \mathcal{N}(\text{vec}(\mathcal{W}) \mid \text{vec}(\mathcal{U}), \boldsymbol{\Gamma}_1 \otimes \boldsymbol{\Gamma}_2 \otimes \cdots \otimes \boldsymbol{\Gamma}_{M+2}),$$

where $\mathcal{U}$ is the mean tensor, $\{\boldsymbol{\Gamma}_1, \boldsymbol{\Gamma}_2, ..., \boldsymbol{\Gamma}_{M+2}\}$ are covariance matrices in each mode, $\boldsymbol{\Gamma}_1 \in \mathbb{R}^{N \times N}, \boldsymbol{\Gamma}_2 \in \mathbb{R}^{L \times L}$, all others are $p \times p$.

Finally, they choose their variational distribution as $q(\mathbf{F}, \mathcal{W}) = q(\mathbf{F}) q(\mathcal{W})$.

**Simplified ELBO**

$$\mathcal{L} = \mathbb{E}_q \left[ \log \frac{p\left(\mathbf{Y}, \mathbf{F}, \mathcal{W} \mid \boldsymbol{\theta}_f, \boldsymbol{\theta}_w, \sigma_f^2, \sigma_y^2\right)}{q(\mathbf{F}, \mathcal{W})} \right] = -\text{KL}[q(\mathbf{F}) \| p(\mathbf{F})] - \text{KL}[q(\mathcal{W}) \| p(\mathcal{W})] + \mathbb{E}_q[\log p(\mathbf{Y} \mid \mathbf{X}, \mathcal{W}, \mathbf{F})],$$

and we can get tensorised prior for $\mathbf{F}$ and $\mathcal{W}$ as follows:

$$p(\mathbf{F}) = \mathcal{MN}(\mathbf{F} \mid \mathbf{0}; \mathbf{L}_{\hat{f}}, \mathbf{I}_L) = \mathcal{N}(\text{vec}(\mathbf{F}) \mid \mathbf{0}, \mathbf{L}_{\hat{f}} \otimes \mathbf{I}_L),$$

$$p(\mathcal{W}) = \mathcal{TN}(\mathcal{W} \mid 0; \mathbf{L}_w, \mathbf{I}_L, \mathbf{I}_{p_1}, ..., \mathbf{I}_{p_M}) = \mathcal{N}(\text{vec}(\mathcal{W}) \mid 0, \mathbf{L}_w \otimes \mathbf{I}_L \otimes \mathbf{I}_{p_1} \otimes \cdots \otimes \mathbf{I}_{p_M}).$$

The KL terms are as follows:

$$\text{KL}[q(\mathbf{F}) \| p(\mathbf{F})] = \frac{1}{2} \left[ \text{tr}\left(\mathbf{L}_{\hat{f}}^{-1} \boldsymbol{\Sigma}\right) \text{tr}(\boldsymbol{\Omega}) - NL + \text{tr}\left(\mathbf{L}_{\hat{f}}^{-1} \mathbf{MM}^\top\right) + L \log \left|\mathbf{L}_{\hat{f}}\right| - (L \log |\boldsymbol{\Sigma}| + N \log |\boldsymbol{\Omega}|) \right],$$

$$\text{KL}[q(\mathcal{W}) \| p(\mathcal{W})] = \frac{1}{2} \left[ \text{tr}\left(\mathbf{L}_w^{-1} \boldsymbol{\Gamma}_1\right) \prod_{m=2}^{M+2} \text{tr}\left(\boldsymbol{\Gamma}_m\right) - NLP + PL \log |\mathbf{L}_w| + \text{tr}\left(\mathbf{L}_w^{-1} \mathbf{U}_1 \mathbf{U}_1^\top\right) - \sum_{m=1}^{M+2} \frac{NPL}{p_m} \log |\boldsymbol{\Gamma}_m| \right],$$

---

[6]In this section, we overload the notation $M$ to represent the number of tensor modes, distinct from the number of inducing points as defined in the main text.

where $\mathbf{U}_1$ is an $N \times PL$ matrix, obtained by unfolding the mean tensor $\mathcal{U}$ at mode 1.

For the Gaussian likelihood, the expected log likelihood term is

$$\mathbb{E}_q[\log p(\mathbf{Y} \mid \mathbf{X}, \mathcal{W}, \mathbf{F})] = -\frac{NP}{2}\log 2\pi - NP\log\sigma_y - \sum_{n=1}^{N}\frac{1}{2\sigma_y^2}\big[\mathbf{y}_n^\top\mathbf{y}_n - 2\mathbf{y}_n^\top\mathbb{E}_q[\mathbf{W}_n]\mathbb{E}_q[\mathbf{h}_n] + \mathrm{tr}(\mathbb{E}_q[\mathbf{W}_n^\top\mathbf{W}_n]\mathbb{E}_q[\mathbf{h}_n\mathbf{h}_n^\top])\big],$$

where $\mathbb{E}_q[\mathbf{W}_n]$ is obtained by taking the $n$-th slice of $\mathcal{U}$ at mode 1, and then reorganise it into an $P \times L$ matrix, $\mathbb{E}_q[\mathbf{h}_n]$ the $n$-th row vector of $\mathbf{M}$. The moments are calculated as follows:

$$\mathbb{E}_q[\mathbf{W}_n^\top\mathbf{W}_n] = \mathbb{E}_q[\mathbf{W}_n]^\top\mathbb{E}_q[\mathbf{W}_n] + \mathbf{\Gamma}_1[n,n]\mathbf{\Gamma}_2\prod_{m=3}^{M+2}\mathrm{tr}(\mathbf{\Gamma}_m),$$

$$\mathbb{E}_q[\mathbf{h}_n\mathbf{h}_n^\top] = \mathbb{E}_q[\mathbf{h}_n]\mathbb{E}_q[\mathbf{h}_n]^\top + \mathbf{\Sigma}[n,n]\mathbf{\Omega}.$$

In general, we do not have a closed-form expression for the expected log-likelihood, so we use Monte Carlo methods to approximate it. For diagonal "noise" models, the expected log likelihood is factorised across inputs. We sample $\mathcal{W}^{(s)}$ and $\mathbf{F}^{(s)}$ from $q(\mathcal{W})$ and $q(\mathbf{F})$, respectively. For a given input $\mathbf{x}_n$,

$$\mathbb{E}_q[\log p(\mathbf{y}_n \mid \mathbf{x}_n, \mathcal{W}, \mathbf{F})] \approx \log p(\mathbf{y}_n \mid \mathbf{x}_n, \mathcal{W}^{(s)}, \mathbf{F}^{(s)}) = \log p(\mathbf{y}_n \mid \mathbf{W}_n^{(s)}\mathbf{h}_n^{(s)}).$$

The sampling can be accelerated by exploiting the Kronecker product structure. Firstly, we sample tensors of i.i.d. standard Gaussian noises, $\epsilon_f$ of shape $N \times L$ and $\epsilon_w$ of shape $N \times L \times p_1 \times \cdots \times p_M$, respectively. Then we generate $\mathbf{F}^{(s)} = \mathbf{M} + \epsilon_f \times_1 \mathrm{chol}(\mathbf{\Sigma}) \times_2 \mathrm{chol}(\mathbf{\Omega})$; $\mathcal{W}^{(s)} = \mathcal{U} + \epsilon_w \times_1 \mathrm{chol}(\mathbf{\Gamma}_1) \times_2 \mathrm{chol}(\mathbf{\Gamma}_2) \times \cdots \times_{M+2} \mathrm{chol}(\mathbf{\Gamma}_{M+2})$, where $\mathrm{chol}(A)$ refers to the Cholesky factor of $A$. The computation of ELBO involves only matrices at each tensor mode, making the optimisation much easier.

**Prediction**  Given a new input $\mathbf{x}^*$, we aim to use the estimated variational posterior to predict the output $\mathbf{y}^*$. The posterior mean of $\mathbf{y}^*$ is computed by $\mathbb{E}[\mathbf{y}^* \mid \mathbf{x}^*, \mathbf{X}, \mathbf{Y}] = \mathbb{E}[\mathbf{W}(\mathbf{x}^*)]\mathbb{E}[\hat{\mathbf{f}}(\mathbf{x}^*)]$ where $\hat{\mathbf{f}}(\mathbf{x}^*) = [\hat{f}_1(\mathbf{x}^*), ..., \hat{f}_L(\mathbf{x}^*)]^\top$. Each $[\mathbb{E}[\mathbf{W}(\mathbf{x}^*)]]_{ij}$ is computed by $\mathbf{k}_w^*\mathbf{L}_w^{-1}\mathbf{v}_{ij}$ where $\mathbf{k}_w^* = [k_w(\mathbf{x}^*, \mathbf{x}_1), ..., k_w(\mathbf{x}^*, \mathbf{x}_N)]$ and $\mathbf{v}_{ij}$ is obtained by first organising $\mathcal{U}$ to an $N \times L \times P$ tensor $\hat{\mathcal{U}}$ and then taking the fiber $\hat{\mathcal{U}}(:,i,j)$. Each $\mathbb{E}[\hat{f}_l(\mathbf{x}^*)] = \mathbf{k}_{\hat{f}}^*\mathbf{L}_{\hat{f}}^{-1}\mathbf{M}(:,l)$ where $\mathbf{k}_{\hat{f}}^* = [k_{\hat{f}}(\mathbf{x}^*, \mathbf{x}_1), ..., k_{\hat{f}}(\mathbf{x}^*, \mathbf{x}_N)]$. The predictive distribution, however, does not have a closed form due to the likelihood being non-Gaussian w.r.t the projection weights and latent functions. We adopt Monte-Carlo approximations, i.e. we generate a set of i.i.d. posterior samples of $\mathbf{W}(\mathbf{x}^*)$ and $\hat{\mathbf{f}}(\mathbf{x}^*)$, denoted by $\{\widetilde{\mathbf{W}}^{(s)}, \widetilde{\mathbf{f}}^{(s)}\}_{s=1}^{S}$, and then approximate $p(\mathbf{y}^* \mid \mathbf{x}^*, \mathbf{X}, \mathbf{Y}) \approx \frac{1}{S}\sum_{s=1}^{S}\mathcal{N}(\mathbf{y}^* \mid \widetilde{\mathbf{W}}^{(s)}\widetilde{\mathbf{f}}^{(s)}, \sigma_y^2\mathbf{I})$.

For prediction, we discard cross-covariances among different test inputs by default. Therefore, w.l.o.g., we abuse the notation $\mathbf{x}^*$ to indicate only one test input.

$$q(\hat{\mathbf{f}}(\mathbf{x}^*)) = \mathcal{N}\left(\hat{\mathbf{f}}(\mathbf{x}^*) \mid \underbrace{\mathbf{k}_{\hat{f}}^*\mathbf{L}_{\hat{f}}^{-1}\mathbf{M}}_{\boldsymbol{\mu}_{\hat{f}} \in \mathbb{R}^{N_{\text{test}} \times L}}, \underbrace{\big(k_{\hat{f}}(\mathbf{x}^*, \mathbf{x}^*) - \mathbf{k}_{\hat{f}}^*\mathbf{L}_{\hat{f}}^{-1}\mathbf{k}_{\hat{f}}^{*\top}\big)}_{\mathbf{L}_{\hat{f}}^{\text{pred}} \in \mathbb{R}^{N_{\text{test}} \times N_{\text{test}}}} \otimes \mathbf{I}_L + \underbrace{\big(\mathbf{k}_{\hat{f}}^*\mathbf{L}_{\hat{f}}^{-1}\mathbf{\Sigma}\mathbf{L}_{\hat{f}}^{-1}\mathbf{k}_{\hat{f}}^{*\top}\big)}_{\mathbf{L}_{\hat{f}}^{\text{VI}} \in \mathbb{R}^{N_{\text{test}} \times N_{\text{test}}}} \otimes \mathbf{\Omega}\right).$$

$\mathbf{L}_{f}^{\text{pred}}$ and $\mathbf{L}_{f}^{\text{VI}}$ can be approximated by diagonal matrices for simplicity. First, generate two tensors of i.i.d. standard Gaussian noise with shape $s \times N_{\text{test}} \times L$, denoted as $\epsilon_f^{(1)}$ and $\epsilon_f^{(2)}$. Then, $\widetilde{\mathbf{F}} = \{\widetilde{\mathbf{f}}^{(s)}\}_{s=1}^{S} = \boldsymbol{\mu}_{\hat{f}} + \epsilon_f^{(1)} \times_2 \mathrm{chol}(\mathbf{L}_{\hat{f}}^{\text{pred}}) + \epsilon_f^{(2)} \times_2 \mathrm{chol}(\mathbf{L}_{\hat{f}}^{\text{VI}}) \times_3 \mathrm{chol}(\mathbf{\Omega})$, where $s$ denotes the number of Monte Carlo samples, and $\widetilde{\mathbf{F}} \in \mathbb{R}^{s \times N_{\text{test}} \times L}$.

$$q(\mathbf{W}(\mathbf{x}^*)) = \mathcal{N}\Big(\mathrm{vec}(\mathbf{W}(\mathbf{x}^*)) \mid \mathrm{vec}(\underbrace{\mathcal{U} \times_1 \mathbf{k}_w^*\mathbf{L}_w^{-1}}_{\boldsymbol{\mu}_w \in \mathbb{R}^{N_{\text{test}} \times L \times p_1 \times \cdots \times p_M}}); \underbrace{\big(k_w(\mathbf{x}^*, \mathbf{x}^*) - \mathbf{k}_w^*\mathbf{L}_w^{-1}\mathbf{k}_w^{*\top}\big)}_{\mathbf{L}_w^{\text{pred}} \in \mathbb{R}^{N_{\text{test}} \times N_{\text{test}}}} \otimes \mathbf{I}_L \otimes \mathbf{I}_P$$

$$+ \underbrace{\big(\mathbf{k}_w^*\mathbf{L}_w^{-1}\mathbf{\Gamma}_1\mathbf{L}_w^{-1}\mathbf{k}_w^{*\top}\big)}_{\mathbf{L}_w^{\text{VI}} \in \mathbb{R}^{N_{\text{test}} \times N_{\text{test}}}} \otimes \mathbf{\Gamma}_2 \otimes \mathbf{\Gamma}_3 \otimes \cdots \otimes \mathbf{\Gamma}_{M+2}\Big).$$

Similarly, generate two tensors of i.i.d. standard Gaussian noise with shape $s \times N_{\text{test}} \times p_1 \times p_2 \times \cdots \times p_M$, denoted as $\epsilon_w^{(1)}$ and $\epsilon_w^{(2)}$. Then, $\widetilde{\mathbf{W}} = \{\widetilde{\mathbf{W}}^{(s)}\}_{s=1}^{S} = \boldsymbol{\mu}_w + \epsilon_w^{(1)} \times_2 \mathrm{chol}(\mathbf{L}_w^{\text{pred}}) + \epsilon_w^{(2)} \times_2 \mathrm{chol}(\mathbf{L}_w^{\text{VI}}) \times_3 \mathrm{chol}(\mathbf{\Gamma}_2) \times \cdots \times_{M+2} \mathrm{chol}(\mathbf{\Gamma}_{M+2})$.

## J.2. Orthogonal Instantaneous Linear Mixing Model (OILMM)

Bruinsma et al. (2020) consider MOGP problems under the assumption that high-dimensional observations lie near a low-dimensional linear subspace. They first review the Instantaneous Linear Mixing Model (ILMM) framework, which is a special type of LMC but enjoys lower computational cost ($\mathcal{O}(N^3 L^3)$ instead of $\mathcal{O}(N^3 P^3)$) by using "sufficient statistic" to exploit the "low-rank" covariance structure. The key idea to achieve such a reduction in computational complexity is to use a "projection" matrix $\mathbf{T}$ to transform the original MOGP with $P$ outputs to another MOGP with $L$ outputs. The authors further note that if the transformed $L$-dimensional MOGP can be decoupled into $L$ independent GPs, the computational complexity can be further reduced to $\mathcal{O}(LN^3)$. They propose several assumptions on the mixing basis vectors and the observation noise matrix to achieve this goal. To support the handling of missing data, the authors illustrate several modifications to their method. One key limitation of their model is that it supports only the Gaussian likelihood, and it appears challenging to extend it to a general likelihood.

### Instantaneous Linear Mixing Model (ILMM)

**Definition J.1.** Consider $L$ independent GPs $z_l(\cdot)$ collectively denoted as $z(\cdot)$, with kernels $k_{\theta_l}$ where $l \in \{1, 2, ..., L\}$, $\mathbf{H}$ is a $P \times L$ matrix with linearly independent columns, and $\mathbf{\Sigma}$ is a covariance matrix of observation noise $P \times P$. Then the ILMM is given by the following generative model:

$$z_l \sim \mathcal{GP}(0, k_{\theta_l}(\mathbf{x}, \mathbf{x}')) \quad \text{(latent processes)}$$
$$f(\mathbf{x}) \mid \mathbf{H}, z(\mathbf{x}) = \mathbf{H}z(\mathbf{x}) \quad \text{(mixing mechanism)}$$
$$\mathbf{y} \mid f(\mathbf{x}) \sim \mathcal{GP}(f(\mathbf{x}), \delta[\mathbf{x} - \mathbf{x}']\mathbf{\Sigma}) \quad \text{(noise model)}$$

*Remark* J.2 (Connections to LMC). For a set of inputs $\mathbf{X}$, if we marginalise out $z(\mathbf{X})$, we find that

$$f(\mathbf{X}) \sim \mathcal{GP}\Big(0, \sum_{l=1}^{L} \mathbf{H}_{:,l}\mathbf{H}_{:,l}^{\top} \otimes k_{\theta_l}(\mathbf{X}, \mathbf{X})\Big),$$

where $\mathbf{H}_{:,l} \in \mathbb{R}^P$ denotes the $l$-th column of the mixing matrix. In LMC, observation noise covariance $\mathbf{\Sigma}$ is typically assumed to be diagonal.

*Remark* J.3 (Connections to Factor Analysis). Recall that Factor Analysis (FA) is a latent variable model having $\mathbf{z} \in \mathbb{R}^L$; $\mathbf{c}, \epsilon, \mathbf{y} \in \mathbb{R}^P$ and $\mathbf{H} \in \mathbb{R}^{P \times L}$, and $\mathbf{\Psi} \in \mathbb{R}^{P \times P}$ is a diagonal matrix. FA is defined as follows:

$$\mathbf{z} \sim \mathcal{N}(\mathbf{z} \mid 0, I_L), \quad \epsilon \sim \mathcal{N}(\epsilon \mid 0, \mathbf{\Psi}), \quad \mathbf{z} \perp\!\!\!\perp \epsilon, \quad \mathbf{y} = \mathbf{Hz} + \mathbf{c} + \epsilon,$$

where $\mathbf{c}$ is the bias term, which we assume is 0 for simplicity. In ILMM, if we choose a white noise kernel for each of the latent processes (i.e. $k_{\theta_l}(\mathbf{x}, \mathbf{x}') = \delta[\mathbf{x} - \mathbf{x}']$) and $\mathbf{\Sigma}$ diagonal, then the ILMM model recovers FA exactly. In other words, the ILMM is a time-varying generalisation of FA.

*Remark* J.4. The ILMM exhibits a low-rank covariance structure, which can be exploited by devising a low-dimensional "summary" or "projection" of the $P$-dimensional observations. They introduce a projection matrix $\mathbf{T} \in \mathbb{R}^{L \times P}$, such that $\mathbf{Ty} \in \mathbb{R}^L$ is minimally sufficient. Specifically:

$$\mathbf{T} = (\mathbf{H}^{\top}\mathbf{\Sigma}^{-1}\mathbf{H})^{-1}\mathbf{H}^{\top}\mathbf{\Sigma}^{-1}$$
$$p(\mathbf{z} \mid \mathbf{y}) = p(\mathbf{z} \mid \mathbf{Ty}),$$

that is, conditioning on $\mathbf{y}$ is equivalent to conditioning on $\mathbf{Ty}$, where $\mathbf{Ty}$ can be interpreted as a "summary" or "projection" of $\mathbf{y}$. The proof can be found in Prop. 3 Appendix D of the original paper (Bruinsma et al., 2020).

**Proposition J.5.** *For the ILMM model equipped with $\mathbf{T}$ defined above, consider the data matrix $\mathbf{Y} \in \mathbb{R}^{P \times N}$, we have the following properties:*

- $\mathbf{Ty} \mid z(\mathbf{x}) \sim \mathcal{GP}(z(\mathbf{x}), \delta[\mathbf{x} - \mathbf{x}']\mathbf{\Sigma}_T)$ *with* $\mathbf{\Sigma}_T = (\mathbf{H}^{\top}\mathbf{\Sigma}^{-1}\mathbf{H})^{-1} = \mathbf{T}\mathbf{\Sigma}\mathbf{T}^{\top}$.

- $p(\mathbf{Y}) = \Big[\prod_{n=1}^{N} \frac{\mathcal{N}(\mathbf{y}_n \mid 0, \mathbf{\Sigma})}{\mathcal{N}(\mathbf{Ty}_n \mid 0, \mathbf{\Sigma}_T)}\Big] \int p\left(z(\mathbf{X})\right) \prod_{n=1}^{N} \mathcal{N}(\mathbf{Ty}_n \mid z(\mathbf{x}_n), \mathbf{\Sigma}_T) dz(\mathbf{X})$.

*where the $n^{th}$ observation $\mathbf{y}_n$ is the $n^{th}$ column of $\mathbf{Y}$. Please refer to Appendix F of the original paper for the proof.*

*Remark* J.6. Crucially, $\mathbf{y}$ are $P$ dimensional, $\mathbf{Ty}$ are $L$ dimensional summaries, and generally, $L \ll P$. The "sufficiency" of statistic $\mathbf{Ty}$ can be used to make predictions on unseen $\mathbf{x}_*$:

$$
\begin{aligned}
p(f(\mathbf{x}_*) \mid \mathbf{Y}) &= \int p(f(\mathbf{x}_*) \mid f(\mathbf{X}))p(f(\mathbf{X}) \mid \mathbf{Y})df(\mathbf{X}) \\
&= \int p(f(\mathbf{x}_*) \mid f(\mathbf{X}))p(f(\mathbf{X}) \mid z(\mathbf{X}))p(z(\mathbf{X}) \mid \mathbf{Y})df(\mathbf{X})dz(\mathbf{X}) \\
&= \int p(f(\mathbf{x}_*) \mid f(\mathbf{X}))p(f(\mathbf{X}) \mid z(\mathbf{X}))p(z(\mathbf{X}) \mid \mathbf{TY})df(\mathbf{X})dz(\mathbf{X}) \\
&= \int p(f(\mathbf{x}_*) \mid z(\mathbf{x}_*))p(z(\mathbf{x}_*) \mid z(\mathbf{X}))p(z(\mathbf{X}) \mid \mathbf{TY})dz(\mathbf{x}_*)dz(\mathbf{X}) \\
&= \int p(f(\mathbf{x}_*) \mid z(\mathbf{x}_*))p(z(\mathbf{x}_*) \mid \mathbf{TY})dz(\mathbf{x}_*),
\end{aligned}
$$

where the last equation can be interpreted as follows: first, project the data $\mathbf{Y}$ and compute the predictive posterior $p(z(\mathbf{x}_*) \mid \mathbf{TY})$ in the $L$-dimensional space, and then map $z(\mathbf{x}_*)$ back to the $P$ dimensional $f(\mathbf{x}_*)$.

*Remark* J.7. The first property in Prop. J.5 gives the "likelihood" model of the latent processes. As the prior distributions of $z(\mathbf{x})$ are independent GPs, $p(z(\mathbf{x}) \mid \mathbf{Ty}) \propto p(z(\mathbf{x}))p(\mathbf{Ty} \mid z(\mathbf{x}))$ can be computed by the posterior equation of GP. Notably, $\mathbf{\Sigma}_T$ is the projected observation noise, which is important as it couples the latent processes upon observing data. In particular, if the latent processes are independent under the prior and $\mathbf{\Sigma}_T$ is diagonal, then the latent processes remain independent when data is observed. This observation forms the basis of the computational gains achieved by the Orthogonal Instantaneous Linear Mixing Model.

**Proposition J.8** (Interpretation of Log Likelihood). *Prop. J.5 shows that the log-probability of the data $\mathbf{Y}$ is equal to the log probability of the projected data $\mathbf{TY}$ plus, for every observation $\mathbf{y}_n$, a "correction" term of the form $\log\mathcal{N}(\mathbf{y}_n \mid 0, \mathbf{\Sigma}) - \log\mathcal{N}(\mathbf{Ty}_n \mid 0, \mathbf{\Sigma}_T)$. That is*

$$
\begin{aligned}
\log p(\mathbf{Y}) &= \log \int p(z(\mathbf{X})) \prod_{n=1}^{N} \mathcal{N}(\mathbf{Ty}_n \mid z(\mathbf{x}_n), \mathbf{\Sigma}_T)dz(\mathbf{x}_n) + \sum_{n=1}^{N} \left[ \log\mathcal{N}(\mathbf{y}_n \mid 0, \mathbf{\Sigma}) - \log\mathcal{N}(\mathbf{Ty}_n \mid 0, \mathbf{\Sigma}_T) \right] \\
&= \log \int p(z(\mathbf{X})) \prod_{n=1}^{N} \mathcal{N}(\mathbf{Ty}_n \mid z(\mathbf{x}_n), \mathbf{\Sigma}_T)dz(\mathbf{x}_n) \underbrace{- \frac{1}{2}\sum_{n=1}^{N} ||\mathbf{y}_n - \mathbf{HT}\mathbf{y}_n||_{\mathbf{\Sigma}}^2}_{\text{data "lost" by projection}} \underbrace{- \frac{N}{2}\log\frac{|\mathbf{\Sigma}|}{|\mathbf{\Sigma}_T|}}_{\text{noise "lost"}} - \frac{N(P-L)}{2}\log 2\pi,
\end{aligned}
$$

*where $|| \cdot ||_{\mathbf{\Sigma}} = ||\mathbf{\Sigma}^{-\frac{1}{2}} \cdot ||$. When the likelihood is optimised with respect to $\mathbf{H}$, the correction terms will prevent the projection $\mathbf{T}$ from discarding a component of the data $\mathbf{Y}$ and the noise $\mathbf{\Sigma}$ that is "too large". For example, for the ILMM, if these correction terms were ignored, then after optimising we would find that $\mathbf{TY} = 0$ and $\mathbf{\Sigma}_T = 0$, because the density of a zero-mean Gaussian is highest at the origin, and becomes higher as the variance becomes smaller; it is exactly $\mathbf{TY} = 0$ and $\mathbf{\Sigma}_T = 0$ that the two penalties prevent from happening.*

*Proof.* By the definition of multivariate Gaussian density:

$$
\begin{aligned}
\log\mathcal{N}(\mathbf{y}_n \mid 0, \mathbf{\Sigma}) &= -\frac{P}{2}\log 2\pi - \frac{1}{2}\log|\mathbf{\Sigma}| - \frac{1}{2}\mathbf{y}_n^\top\mathbf{\Sigma}^{-1}\mathbf{y}_n \\
\log\mathcal{N}(\mathbf{Ty}_n \mid 0, \mathbf{\Sigma}_T) &= -\frac{L}{2}\log 2\pi - \frac{1}{2}\log|\mathbf{\Sigma}_T| - \frac{1}{2}\mathbf{y}_n^\top\mathbf{T}^\top\mathbf{\Sigma}_T^{-1}\mathbf{Ty}_n \\
&= -\frac{L}{2}\log 2\pi - \frac{1}{2}\log|\mathbf{\Sigma}_T| - \frac{1}{2}\mathbf{y}_n^\top\mathbf{T}^\top\left(\mathbf{H}^\top\mathbf{\Sigma}^{-1}\mathbf{H}\right)\mathbf{Ty}_n,
\end{aligned}
$$

Thus,

$$
\log\mathcal{N}(\mathbf{y}_n \mid 0, \mathbf{\Sigma}) - \log\mathcal{N}(\mathbf{Ty}_n \mid 0, \mathbf{\Sigma}_T) = -\frac{(P-L)}{2}\log 2\pi - \frac{1}{2}\log\frac{|\mathbf{\Sigma}|}{|\mathbf{\Sigma}_T|} - \frac{1}{2}\mathbf{y}_n^\top\left(\mathbf{\Sigma}^{-1} - \mathbf{T}^\top\mathbf{H}^\top\mathbf{\Sigma}^{-1}\mathbf{HT}\right)\mathbf{y}_n,
$$

For the last term:

$$
\mathcal{T}_{\text{quadratic}} = \mathbf{y}_n^\top\left(\mathbf{\Sigma}^{-1} - \mathbf{T}^\top\mathbf{H}^\top\mathbf{\Sigma}^{-1}\mathbf{HT}\right)\mathbf{y}_n = (\mathbf{\Sigma}^{-\frac{1}{2}}\mathbf{y}_n)^\top\left[I_P - \mathbf{\Sigma}^{\frac{1}{2}}\mathbf{T}^\top\mathbf{H}^\top\mathbf{\Sigma}^{-1}\mathbf{HT}\mathbf{\Sigma}^{\frac{1}{2}}\right](\mathbf{\Sigma}^{-\frac{1}{2}}\mathbf{y}_n).
$$

We realise that $\mathbf{P} = \boldsymbol{\Sigma}^{-\frac{1}{2}}\mathbf{HT}\boldsymbol{\Sigma}^{\frac{1}{2}}$ is an orthogonal projection matrix onto column space of $\boldsymbol{\Sigma}^{-\frac{1}{2}}\mathbf{H}$ by showing that:

$$\mathbf{P}^2 = \mathbf{P} \quad \text{and} \quad \mathbf{P}^\top = \mathbf{P}.$$

$$\mathbf{P}^2 = (\boldsymbol{\Sigma}^{-\frac{1}{2}}\mathbf{HT}\boldsymbol{\Sigma}^{\frac{1}{2}})(\boldsymbol{\Sigma}^{-\frac{1}{2}}\mathbf{HT}\boldsymbol{\Sigma}^{\frac{1}{2}}) = \boldsymbol{\Sigma}^{-\frac{1}{2}}\mathbf{HT}\boldsymbol{\Sigma}^{\frac{1}{2}} = \mathbf{P} \quad \text{where we used the fact that} \quad \mathbf{TH} = I_L$$

$$\begin{aligned}
\mathbf{P}^\top &= \boldsymbol{\Sigma}^{\frac{1}{2}}\mathbf{T}^\top\mathbf{H}^\top\boldsymbol{\Sigma}^{-\frac{1}{2}} \\
&= \boldsymbol{\Sigma}^{\frac{1}{2}}\boldsymbol{\Sigma}^{-1}\mathbf{H}(\mathbf{H}^\top\boldsymbol{\Sigma}^{-1}\mathbf{H})^{-1}\mathbf{H}^\top\boldsymbol{\Sigma}^{-\frac{1}{2}} \quad \text{where we use the definition of } \mathbf{T} \\
&= \boldsymbol{\Sigma}^{-\frac{1}{2}}\mathbf{H}(\mathbf{H}^\top\boldsymbol{\Sigma}^{-1}\mathbf{H})^{-1}\mathbf{H}^\top\boldsymbol{\Sigma}^{-1}\boldsymbol{\Sigma}^{\frac{1}{2}} \\
&= \boldsymbol{\Sigma}^{-\frac{1}{2}}\mathbf{HT}\boldsymbol{\Sigma}^{\frac{1}{2}}.
\end{aligned}$$

Therefore,

$$\begin{aligned}
\mathcal{T}_{quadratic} &= (\boldsymbol{\Sigma}^{-\frac{1}{2}}\mathbf{y}_n)^\top\left[I_P - \mathbf{P}^\top\mathbf{P}\right](\boldsymbol{\Sigma}^{-\frac{1}{2}}\mathbf{y}_n) \\
&= (\boldsymbol{\Sigma}^{-\frac{1}{2}}\mathbf{y}_n)^\top\left[I_P - \mathbf{P}\right](\boldsymbol{\Sigma}^{-\frac{1}{2}}\mathbf{y}_n) \\
&\quad \text{where we note that } (I_P - \mathbf{P}) \text{ is also orthogonal projection} \\
&= (\boldsymbol{\Sigma}^{-\frac{1}{2}}\mathbf{y}_n)^\top\left[(I_P - \mathbf{P})^\top(I_P - \mathbf{P})\right](\boldsymbol{\Sigma}^{-\frac{1}{2}}\mathbf{y}_n) \\
&= \|(I_P - \mathbf{P})\boldsymbol{\Sigma}^{-\frac{1}{2}}\mathbf{y}_n\|^2 \\
&= \|\boldsymbol{\Sigma}^{-\frac{1}{2}}(I_P - \boldsymbol{\Sigma}^{\frac{1}{2}}\mathbf{P}\boldsymbol{\Sigma}^{-\frac{1}{2}})\mathbf{y}_n\|^2 \\
&= \|(I_P - \mathbf{HT})\mathbf{y}_n\|_{\boldsymbol{\Sigma}}^2 \quad \text{where we use the definition of } \mathbf{P} \\
&= \|\mathbf{y}_n - \mathbf{HTy}_n\|_{\boldsymbol{\Sigma}}^2.
\end{aligned}$$

$\square$

**Orthogonal Instantaneous Linear Mixing Model (OILMM)**

**Definition J.9.** The OILMM is an ILMM where the basis $\mathbf{H}$ is a $P \times L$ matrix of the form $\mathbf{H} = \mathbf{US}^{\frac{1}{2}}$ with $\mathbf{U} \in \mathbb{R}^{P \times L}$ a matrix with orthonormal columns and $\mathbf{S} > 0$ is $L \times L$ diagonal matrix, and $\boldsymbol{\Sigma} = \sigma_y^2 I_P + \mathbf{HDH}^\top$ is a $P \times P$ matrix with $\mathbf{D} \geq 0$ being a $L \times L$ diagonal matrix.

*Remark* J.10. In OILMM, we require $L \leq P$, since the number of $P$-dimensional vectors that can be mutually orthogonal is at most $P$. Also, the $\mathbf{HDH}^\top$ term in $\boldsymbol{\Sigma}$ can be interpreted as heterogeneous noise deriving from the latent processes. Alternatively, this term can be discarded if we add $\delta[\mathbf{x} - \mathbf{x}']\mathbf{D}_{ll}$ as an additional term to $k_{\theta_l}$ of latent GPs, which is the approach we choose in our implementation.

**Proposition J.11.** *Consider the OILMM model, the projection and projected noise are given by*

$$\mathbf{T} = \mathbf{S}^{-\frac{1}{2}}\mathbf{U}^\top \quad \text{and} \quad \boldsymbol{\Sigma}_T = \sigma_y^2\mathbf{S}^{-1} + \mathbf{D}.$$

*Proof.* We use the technique that the term $\mathbf{D}$ in $\boldsymbol{\Sigma}$ can be absorbed into the kernel matrices of latent GPs. That is, $\mathbf{D}$ can be assumed to be 0 for the sake of simplicity in the reasoning below.

By definition,

$$\mathbf{H}^\top\boldsymbol{\Sigma}^{-1}\mathbf{H} = \mathbf{S}^{\frac{1}{2}}\mathbf{U}^\top\sigma_y^{-2}\mathbf{US}^{\frac{1}{2}} = \sigma_y^{-2}\mathbf{S}$$

$$\mathbf{T} = (\mathbf{H}^\top\boldsymbol{\Sigma}^{-1}\mathbf{H})^{-1}\mathbf{H}^\top\boldsymbol{\Sigma}^{-1} = (\sigma_y^2\mathbf{S}^{-1})(\sigma_y^{-2}\mathbf{S}^{\frac{1}{2}}\mathbf{U}^\top) = \mathbf{S}^{-\frac{1}{2}}\mathbf{U}^\top$$

$$\boldsymbol{\Sigma}_T = (\mathbf{H}^\top\boldsymbol{\Sigma}^{-1}\mathbf{H})^{-1} = \sigma_y^2\mathbf{S}^{-1}.$$

Finally, "pull $\mathbf{D}$ back out of kernel matrix of latent GPs", which we note is equivalent to adding it to $\boldsymbol{\Sigma}_T$. $\square$

*Remark* J.12 (Diagonal Projected Noise). As alluded to in the previous sections, under the OILMM, the projected noise $\boldsymbol{\Sigma}_T$ is diagonal: $\boldsymbol{\Sigma}_T = \sigma_y^2\mathbf{S}^{-1} + \mathbf{D}$. This property enables the model to decompose the high-dimensional multi-output problem into independent single-output problems, yielding significant computational advantages.

**Proposition J.13** (Log Likelihood). *For computing the log marginal likelihood, the OILMM offers computational benefits:*

$$\log p(\mathbf{Y}) = \underbrace{\sum_{l=1}^{L} \log \mathcal{N}((\mathbf{TY})_{l:} \mid 0, k_{\theta_l}(\mathbf{X}, \mathbf{X}) + (\sigma_y^2/\mathbf{S}_{ll} + \mathbf{D}_{ll})I_N)}_{\text{sum of Log Marginal Likelihood of Independent latent GPs}}$$

$$- \left[ \underbrace{\frac{N}{2} \log |\mathbf{S}| + \frac{N(P-L)}{2} \log 2\pi\sigma_y^2 + \frac{1}{2} \sum_{n=1}^{N} ||(I_P - \mathbf{HT})\mathbf{y}_n||_{\sigma_y^2 I_P}^2}_{\text{regularisation term}} \right].$$

*Proof.* We start with the general result for ILMM in Prop. J.8. Using the same technique which assumes $\mathbf{D} = 0$ by absorbing it into the kernel functions $k_{\theta_l}$:

$$\log \frac{|\mathbf{\Sigma}|}{|\mathbf{\Sigma}_T|} = \log \frac{|\sigma_y^2 I_P|}{|\sigma_y^2 \mathbf{S}^{-1}|} = (P - L) \log \sigma_y^2 + \log |\mathbf{S}|.$$

Thus, the term $||(I_P - \mathbf{HT})\mathbf{y}_n||_{\sigma_y^2 I_P}^2$ can be computed by $\sigma_y^{-2} * [\mathbf{y}_n - \mathbf{H}(\mathbf{Ty}_n)]$.square().sum() in `PyTorch`. $\qquad \square$

### Missing data

**Proposition J.14.** *For OILMM, consider $P_o$ observed output $\mathbf{y}_o \in \mathbb{R}^{P_o}$, which is a subset of all outputs $\mathbf{y}$. Therefore, $\mathbf{U}_o \in \mathbb{R}^{P_o \times L}$ denotes the selected rows from $\mathbf{U}$, and $\mathbf{H}_o = \mathbf{U}_o \mathbf{S}^{\frac{1}{2}}$, $\mathbf{\Sigma}_o = \sigma_y^2 I_{P_o} + \mathbf{H}_o \mathbf{D} \mathbf{H}_o^\top$.*

*The projection and projected noise are given by*

$$\mathbf{T}_o = \mathbf{S}^{-\frac{1}{2}} (\mathbf{U}_o^\top \mathbf{U}_o)^{-1} \mathbf{U}_o^\top = \mathbf{S}^{-\frac{1}{2}} \mathbf{U}_o^\dagger$$

$$\mathbf{\Sigma}_{T_o} \approx \sigma_y^2 \mathbf{S}^{-\frac{1}{2}} diag\left[ (\mathbf{U}_o^\top \mathbf{U}_o)^{-1} \right] \mathbf{S}^{-\frac{1}{2}} + \mathbf{D},$$

*where $\mathbf{U}_o^\dagger$ is the pseudo-inverse of $\mathbf{U}_o$.*

*Proof.* The proof follows the definition that:

$$\mathbf{T}_o = (\mathbf{H}_o^\top \mathbf{\Sigma}_o^{-1} \mathbf{H}_o)^{-1} \mathbf{H}_o^\top \mathbf{\Sigma}_o^{-1},$$
$$\mathbf{\Sigma}_{T_o} = (\mathbf{H}_o^\top \mathbf{\Sigma}_o^{-1} \mathbf{H}_o)^{-1}.$$

Again, we assume $\mathbf{D} = 0$ by absorbing it into the kernel matrix, and note that

$$\mathbf{H}_o^\top \mathbf{\Sigma}_o^{-1} \mathbf{H}_o = \sigma_y^{-2} \mathbf{S}^{\frac{1}{2}} \mathbf{U}_o^\top \mathbf{U}_o \mathbf{S}^{\frac{1}{2}},$$

So,

$$\mathbf{\Sigma}_{T_o} = (\mathbf{H}_o^\top \mathbf{\Sigma}_o^{-1} \mathbf{H}_o)^{-1} = \sigma_y^2 \mathbf{S}^{-\frac{1}{2}} (\mathbf{U}_o^\top \mathbf{U}_o)^{-1} \mathbf{S}^{-\frac{1}{2}}.$$

We recall that $\mathbf{D}$ must be extracted from the kernel matrices, which corresponds to adding it to $\mathbf{\Sigma}_{T_o}$. However, note that $\mathbf{\Sigma}_{T_o}$ is dense, because, unlike $\mathbf{U}$, the columns of $\mathbf{U}_o$ are not orthogonal. However, they may be approximately orthogonal, which motivates the approximation

$$\mathbf{\Sigma}_{T_o} = \sigma_y^2 \mathbf{S}^{-\frac{1}{2}} (\mathbf{U}_o^\top \mathbf{U}_o)^{-1} \mathbf{S}^{-\frac{1}{2}} + \mathbf{D} \approx \sigma_y^2 \mathbf{S}^{-\frac{1}{2}} diag\left[ (\mathbf{U}_o^\top \mathbf{U}_o)^{-1} \right] \mathbf{S}^{-\frac{1}{2}} + \mathbf{D}.$$

Moreover,

$$\mathbf{T}_o = (\mathbf{H}_o^\top \mathbf{\Sigma}_o^{-1} \mathbf{H}_o)^{-1} \mathbf{H}_o^\top \mathbf{\Sigma}_o^{-1} = (\sigma_y^2 \mathbf{S}^{-\frac{1}{2}} (\mathbf{U}_o^\top \mathbf{U}_o)^{-1} \mathbf{S}^{-\frac{1}{2}})(\sigma_y^{-2} \mathbf{S}^{\frac{1}{2}} \mathbf{U}_o^\top) = \mathbf{S}^{-\frac{1}{2}} \mathbf{U}_o^\dagger.$$

$\qquad \square$

**Proposition J.15** (Log Likelihood under missing data). *Denote* $\mathbf{Y}_o \in \mathbb{R}^{P_o \times N_o}$, *where* $P_o \leq P$ *and* $N_o \leq N$.

$$\log p(\mathbf{Y}_o) \approx \underbrace{\sum_{l=1}^{L} \log \mathcal{N}\left((\mathbf{T}_o\mathbf{Y}_o)_{l:} \mid 0, k_{\theta_l}(\mathbf{X}, \mathbf{X}) + \left(\sigma_y^2 * \left(\frac{diag[(\mathbf{U}_o^\top \mathbf{U}_o)^{-1}]}{\mathbf{S}}\right)_{ll} + \mathbf{D}_{ll}\right) I_{N_o}\right)}_{\text{sum of Log Marginal Likelihood of Independent Latent GPs}}$$

$$- \underbrace{\left[\frac{N_o}{2}\log|\mathbf{S}| - \frac{N_o}{2}\log\left|diag\left[(\mathbf{U}_o^\top \mathbf{U}_o)^{-1}\right]\right| + \frac{N_o(P_o - L)}{2}\log 2\pi\sigma_y^2 + \frac{1}{2}\sum_{n=1}^{N_o}||\mathbf{y}_n - \mathbf{H}_o\mathbf{T}_o\mathbf{y}_n||^2_{\sigma_y^2 I_{P_o}}\right]}_{\text{regularisation term}}.$$

*Proof.* The approximation arises from the misspecification of $\boldsymbol{\Sigma}_{T_o}$, as $\boldsymbol{\Sigma}_{T_o}$ is dense, which couples the latent processes. For efficient computation, we discard the off-diagonal entries of $(\mathbf{U}_o^\top \mathbf{U}_o)^{-1}$, thereby enforcing independence among the latent GPs. Proceeding from Prop. J.8, $\mathbf{D}$ is assumed to be 0 by absorbing it into the kernel matrix:

$$\log\frac{|\boldsymbol{\Sigma}_o|}{|\boldsymbol{\Sigma}_{T_o}|} = \log\frac{\left|\sigma_y^2 I_{P_o}\right|}{\left|\sigma_y^2 \mathbf{S}^{-\frac{1}{2}}diag\left[(\mathbf{U}_o^\top \mathbf{U}_o)^{-1}\right]\mathbf{S}^{-\frac{1}{2}}\right|} = \log\frac{\left|\sigma_y^2 I_{P_o}\right|}{\left|\sigma_y^2 I_L \mathbf{S}^{-1}diag\left[(\mathbf{U}_o^\top \mathbf{U}_o)^{-1}\right]\right|}$$

$$= (P_o - L)\log\sigma_y^2 + \log|\mathbf{S}| - \log\left|diag\left[(\mathbf{U}_o^\top \mathbf{U}_o)^{-1}\right]\right|.$$

$\square$

*Remark* J.16. Standard scalable GP techniques (e.g. SVGP) can be used to approximate the log marginal likelihood terms, with the Gaussian likelihood noise $\boldsymbol{\Sigma}_{T_o} \in \mathbb{R}^{L \times L}$ diagonal. By default, SVGPs are employed as latent processes.

$$\log p(\mathbf{Y}_o) \approx \sum_{l=1}^{L} \log p\left((\mathbf{T}_o\mathbf{Y}_o)_{l:}\right) - \sum_{n=1}^{N_o} n^{th} \text{ regularisation term}$$

$$\geq \sum_{l=1}^{L} \text{ELBO}((\mathbf{T}_o\mathbf{Y}_o)_{l:}) - \sum_{n=1}^{N_o} n^{th} \text{ regularisation term}.$$

**Prediction** The procedure of making predictions for all $P$ outputs on an unseen location $\mathbf{x}_*$ can be broken down into two steps: latent GP prediction and reconstruction. The first step is to obtain $L$ SVGP predictive means and variances at $\mathbf{x}_*$, which we denote as $\mu(\mathbf{x}_*) \in \mathbb{R}^L$ and $\nu(\mathbf{x}_*) \in \mathbb{R}^L$. Then, we obtain the predictive mean and variance in observation space by

$$\text{predictive mean} = \mathbf{H}\mu \in \mathbb{R}^P,$$
$$\text{predictive marginal variances} = (\mathbf{H} \circ \mathbf{H})\nu + \sigma_y^2 1_P + \text{diag}(\mathbf{H}\mathbf{D}\mathbf{H}^\top) \in \mathbb{R}^P,$$

where $\circ$ denotes the Hadamard product.

# K. Methodological Comparison with Baseline Models

This section provides a comparison between the T-LVMOGP and baseline models from a methodological perspective.

**Compare to SV-LMC and GS-LVMOGP:** In SV-LMC, the output functions are generated from some shared latent GPs, inducing a low-rank coregionalisation structure. GS-LVMOGP generalises it by replacing the linear kernel with any kernel function. However, both models inherit a sum-of-separable cross-output covariance structure, which might limit the modelling power. T-LVMOGP constructs cross-output correlations more flexibly by leveraging a learnt embedding space paired with a base kernel. This approach avoids rigid structural assumptions while maintaining computational efficiency. Furthermore, in Appendix D we demonstrate that the kernel of T-LVMOGP generalises that of GS-LVMOGP, underscoring T-LVMOGP's superior capacity.

**Compare to OILMM:** As in Appendix J.2, OILMM is a special type of LMC with a lower computational cost by exploiting its structure. OILMM relies on restrictive assumptions about the basis vectors and the observation noise matrix, which may limit its expressiveness. OILMM lacks native support for missing data, and adopting it for such cases requires modifications that undermine its computational benefits. OILMM is strictly tied to a Gaussian likelihood, limiting its applicability to non-Gaussian data. T-LVMOGP reformulates MOGPs as scalar GPs; it naturally supports missing data by marginalising them out in the ELBO. T-LVMOGP accommodates non-Gaussian likelihood, such as the ZINB likelihood for Spatial Transcripomics.

**Compare to SGPRN:** Though SGPRN is computationally efficient when the output space has a grid structure by exploiting tensor algebra, its applicability is restricted, as many real-world datasets cannot be naturally represented as multidimensional arrays. Moreover, SGPRN has cubic complexity with respect to the number of inputs $N$, limiting its scalability to large datasets. T-LVMOGP does not rely on explicit structural assumptions for cross-output covariances. Instead, it constructs covariances in a more flexible manner via an embedding space and a base kernel. We develop a scalable procedure that supports mini-batch training on both input and output, allowing T-LVMOGP to scale to datasets with large $N$ and $P$.

**Compare to G-MOGP:** G-MOGP employs an attention mechanism to model cross-output correlations, which requires constructing $P \times P$ attention weight matrices. As $P$ grows, this introduces substantial memory and computational overhead. Moreover, G-MOGP uses an independent SVGP for each output, each of which introduces $MD_X + M + M^2$ variational parameters. When $P$ is large, the total number of variational parameters becomes substantial, which further increases the training cost. T-LVMOGP defines the variational distribution $q(\mathbf{u})$ over the learned embedding space, shared across all outputs. This shared parameterisation can be viewed as an amortised variational construction, thereby improving parameter efficiency.

## L. Limitations and Future Work

In the current formulation, we employ a mean-field approximation for the variational distribution of the latent variables, factorising $q(\mathbf{H})$ across outputs to ensure computational efficiency. While this enhances scalability, it may limit posterior expressiveness by ignoring potential coupling among outputs. Future work could address this trade-off by adopting structured variational distributions (Mishkin et al., 2018; Tomczak et al., 2020), which capture complex dependencies without incurring prohibitive computational costs. Furthermore, although the parameter complexity of $q(\mathbf{H})$ scales linearly with the number of outputs $P$, future work could investigate imposing additional constraints within the latent space, such as amortisation (Kingma & Welling, 2014). Such constraints would reduce the parameterisation required for estimation, thereby further improving scalability for high-dimensional output spaces.

