# OpenReview forum: "Transformed Latent Variable Multi-Output Gaussian Processes"
_ICML.cc/2026/Conference — ICML 2026 regular_

### Official Review · Reviewer_6zJP · 2026-03-12

**Soundness:** 4
**Presentation:** 3
**Significance:** 4
**Originality:** 3
**Overall Recommendation:** 5
**Confidence:** 4

**Summary:**

This paper introduces the Transformed Latent Variable Multi-Output Gaussian Process (T-LVMOGP), a novel and scalable framework that addresses the critical challenge of modeling high-dimensional outputs in MOGPs. The core innovation lies in the construction of a flexible multi-output deep kernel. By employing a Lipschitz-regularized residual connection neural network (RCNN) to jointly map input features and output-specific latent variables into a unified embedding space, the framework elegantly transforms a complex multi-output problem into a more tractable scalar Gaussian process. Placing inducing points directly in this learned embedding space and leveraging stochastic variational inference (SVI) enables seamless integration with existing scalable GP frameworks, allowing for mini-batch training over both inputs and outputs. The incorporation of spectral normalization and residual connections to maintain distance-awareness and mitigate overfitting—a known pitfall in Deep Kernel Learning (DKL)—is both theoretically sound and practically effective. The paper makes a substantial academic contribution through its rigorous mathematical derivations, innovative architectural design, and comprehensive, persuasive empirical results on diverse, real-world datasets, including those from climate science and spatial transcriptomics.

**Compliance With Llm Reviewing Policy:**

Affirmed.

**Key Questions For Authors:**

1. How does the model ensure that the latent variable $h_p$ captures output-specific information rather than being ignored by the neural network? Is there any risk of collapse to an independent-outputs model, and could architectural inductive biases (e.g., multiplicative interactions) help?

2. Given that non-stationarity is induced solely by the input-dependent mapping $\phi_{\theta}$, how well does the model perform when the correlation structure itself is highly non-stationary? Could a non-stationary base kernel provide further benefits?

**Limitations:**

yes

**Strengths And Weaknesses:**

Strengths:

1. Innovative Architecture: The deep kernel formulation with Lipschitz-regularized RCNN enables expressive yet stable cross-output correlations without imposing rigid structural assumptions.

2. Scalability: By leveraging the SVGP framework and placing inducing points in the embedding space, the model scales to datasets with massive output dimensions and supports mini‑batch training over both inputs and outputs.

3. Theoretical Rigor: The use of spectral normalization and residual connections is well justified, and connections to prior LV‑MOGP models are clearly established.

4. Empirical Validation: Extensive experiments on EEG, SARCOS, climate, and spatial transcriptomics data demonstrate consistent outperformance over strong baselines in both accuracy and computational efficiency.

Weaknesses:

1. Although the core innovation of the model lies in constructing a deep joint embedding space, the experimental section lacks intuitive, low-dimensional visualizations (e.g., t-SNE or UMAP) of the learned output latent variables. Providing such an analysis—for instance, on the Copernicus Marine dataset—would substantially enhance the interpretability of the framework. Demonstrating that physically proximal monitoring stations exhibit natural clustering in the nonlinear embedding space would offer compelling evidence of the model's capacity to learn meaningful output representations.

2. Section 3.3 (Variational Inference) and the corresponding Appendix B contain several crucial derivations that are difficult to follow in their current form, which may hinder reproducibility.  A more detailed, step-by-step derivation—perhaps starting with the full-data bound, then introducing the factorization over latent variables, and finally applying mini-batching with explicit handling of missing data—would significantly improve the paper's clarity and ensure that the inference procedure is fully understood by readers seeking to implement the model.

---

> ### Author Rebuttal · Authors · 2026-03-30
>
> Thank you for the feedback and for taking the time to review our paper. Please find responses to your comments.
>
> ---
> ## 1. More details about ELBO derivation.
> ---
> Consider the latent variables $\\{ \\mathbf{H}, \\mathbf{f}, \\mathbf{u} \\}$ and observations $\\mathbf{Y}$. We introduce the variational distribution $q(\\mathbf{H}, \\mathbf{u}, \\mathbf{f}) = q(\\mathbf{H}) q(\\mathbf{u}) p(\\mathbf{f} \\mid \\mathbf{u}, \\mathbf{H}), $ and the joint distribution is $
> p(\\mathbf{Y}, \\mathbf{H}, \\mathbf{u}, \\mathbf{f}) = p(\\mathbf{Y} \\mid \\mathbf{f}) \\, p(\\mathbf{H}) \\, p(\\mathbf{u}) \\, p(\\mathbf{f} \\mid \\mathbf{u}, \\mathbf{H}). $
>
> $$
> \\begin{aligned}
> \\log p(\\mathbf{Y})
> &= \\log \\int p(\\mathbf{Y}, \\mathbf{H}, \\mathbf{u}, \\mathbf{f}) \\, d\\mathbf{H} \\, d\\mathbf{u} \\, d\\mathbf{f} \\\\
> &= \\log \\int q(\\mathbf{H}, \\mathbf{u}, \\mathbf{f})
> \\frac{p(\\mathbf{Y}, \\mathbf{H}, \\mathbf{u}, \\mathbf{f})}{q(\\mathbf{H}, \\mathbf{u}, \\mathbf{f})} d\\mathbf{H} \\, d\\mathbf{u} \\, d\\mathbf{f} \\\\
> &\\ge
> \\int q(\\mathbf{H}, \\mathbf{u}, \\mathbf{f})
> \\log \\left[
> \\frac{p(\\mathbf{Y}, \\mathbf{H}, \\mathbf{u}, \\mathbf{f})}
> {q(\\mathbf{H}, \\mathbf{u}, \\mathbf{f})}
> \\right] d\\mathbf{H} \\, d\\mathbf{u} \\, d\\mathbf{f},
> \\end{aligned}
> $$
>
> where the inequality follows from Jensen's inequality. The remaining derivation is given in Appendix B of the paper.
>
> Note that $q(\\mathbf{H})$ and $p(\\mathbf{H})$ factorise across outputs as in Section 3.2 (Lines 191--196):
>
> $$
> \\text{KL}\\!\\bigl[q(\\mathbf{H}) \\Vert p(\\mathbf{H})\\bigr]
> = \\int \\left( \\prod_{p=1}^{P} q(\\mathbf{h}_p) \\right)
> \\log \\left[
> \\frac{\\prod_p q(\\mathbf{h}_p)}
> {\\prod_p p(\\mathbf{h}_p)}
> \\right]
> \\, d\\mathbf{h}_1 \\cdots d\\mathbf{h}_P
> $$
>
> $$
> = \\sum_{p=1}^{P} \\int q(\\mathbf{h}_p)
> \\log \\left[
> \\frac{q(\\mathbf{h}_p)}{p(\\mathbf{h}_p)}
> \\right]
> \\, d\\mathbf{h}_p
> $$
>
> $$
> = \\sum_{p=1}^{P}
> \\operatorname{KL}\\!\\bigl[q(\\mathbf{h}_p) \\Vert p(\\mathbf{h}_p)\\bigr].
> $$
>
> For datasets with missing observations, let
> $\\Omega \\subseteq \\{1, \\ldots, P\\} \\times \\{1, \\ldots, N\\}$
> denote the index set of all observed training entries. Under this notation, the data-fit term in the ELBO becomes
> $\\frac{NP}{|\\Omega|} \\sum_{(n,p) \\in \\Omega} \\mathbb{E}_{q(\\mathbf{h}_p)} \\bigl[\\mathcal{V}^{n,p} (\\mathbf{h}_p)\\bigr].$
>
> We plan to release the full source code for T-LVMOGP, together with the implementations of all baseline models, to support the reproducibility of this work.
>
> ---
> ## 2. Does the latent variable $h_p$ capture output-specific information? Is there any risk of collapse to an independent-output model?
> ---
> Yes. The latent variable $h_p$ is the only output-specific component in T-LVMOGP. If the neural network were to ignore $h_p$, the model would be unable to produce output-dependent embeddings and hence could not learn meaningful cross-output covariances. In that case, predictive performance would deteriorate substantially.
>
> To verify this empirically, we consider an ablated setting in which all latent variables in T-LVMOGP are fixed to the zero vector and are not updated during training, while all other components are learned as usual. On the EEG and SARCOS datasets, we obtain the following results:
>
> | Setting | EEG: Learn $h_p$ | EEG: Fix $h_p$ | SARCOS: Learn $h_p$ | SARCOS: Fix $h_p$ |
> |---|---:|---:|---:|---:|
> | MSE | **0.115 $\\pm$ 0.025** | 0.558 $\\pm$ 0.005 | **0.022 $\\pm$ 0.000** | 0.798 $\\pm$ 0.001 |
> | NLL | **0.814 $\\pm$ 0.310** | 150.4 $\\pm$ 75.06 | **-0.485 $\\pm$ 0.009** | 1.306 $\\pm$ 0.001 |
>
> These results show that when the latent variables contain no output-specific information, T-LVMOGP fails to capture meaningful cross-output correlations and produces severely degraded predictions. By contrast, as shown in Section 5, the full model achieves strong predictive performance, indicating that the learned latent variables $h_p$ do capture useful output-specific information.
>
> We also include an independent GP (Ind-GP) baseline in the EEG and ERA5 experiments. T-LVMOGP consistently outperforms Ind-GP by a large margin, suggesting that the model does not collapse to an independent-output solution during training.
>
> ---
> ## 3. How well does the model perform when the correlation structure itself is highly non-stationary? Could a non-stationary base kernel provide further benefits?
> ---
> In our current implementation, we use an ARD-RBF kernel as the base kernel, which is stationary. Therefore, any non-stationarity in T-LVMOGP is introduced through the learned transformation $\\Phi_{\\theta}$. Since $\\Phi_{\\theta}$ is regularised by spectral normalisation to have a bounded Lipschitz constant, we do not expect T-LVMOGP to exhibit extremely strong non-stationary behaviour.
>
> We agree, however, that adopting a non-stationary base kernel could further improve the model's ability to capture highly non-stationary correlation structures. This is an interesting direction for future work.

---

> > ### Author Rebuttal · Reviewer_6zJP · 2026-04-03
> >
> > The reviewer thanks the authors for their responses.

---

> > > ### Author Response · Authors · 2026-04-04
> > >
> > > We are pleased that our responses have fully addressed your questions. We sincerely appreciate your comments on the ELBO derivation and on the role of $h_p$ in capturing output-specific information, which have helped us further clarify both the presentation and the technical soundness of our work. In the revised manuscript, we will provide additional details on the ELBO derivation and include further ablation results to more clearly demonstrate the role of $h_p$ in capturing output-specific information.

---

### Official Review · Reviewer_tyNS · 2026-03-12

**Soundness:** 4
**Presentation:** 4
**Significance:** 4
**Originality:** 3
**Overall Recommendation:** 5
**Confidence:** 4

**Summary:**

This work proposes a novel deep kernel multi-output Gaussian process model that builds on four key features that are incorporated into the MOGP framework: 1) additional latent variable vectors that are associated to each of the outputs, 2) a deep kernel approach that uses a carefully regulated neural network that maps inputs together with output-specific latent variables to output-specific embedding features, 3) a base kernel in the learned embedding feature space, and 4) variational training of the constructed model with inducing points in the embedding feature space that can be trained using mini batching across inputs and out dimensions.  The proposed model is claimed to learn non-trivial correlations across output dimensions implicitly via the learned output-specific latent variables and the learned embedding via the regularized neural network, and the base kernel. While the model is not as straightforward as simply MOGPs, the modeling approach seems reasonable when thought in the context of learnable embeddings that implicitly define the correlations across outputs (and across samples). The proposed model is evaluated on several real datasets with competitive results. The proposed model does require careful fine-tuning, especially for the trainable neural network based embedding to avoid overfitting.

**Compliance With Llm Reviewing Policy:**

Affirmed.

**Key Questions For Authors:**

A number of different MOGP methods have been proposed and this manuscript makes a good effort to give a comprehensive review of these methods.  Empirical evaluation wrt selected previous methods shows better accuracy across several datasets and different prediction tasks. A critical component in the proposed method is the careful regularization of the neural network based embedding mapping for which authors propose to use residual based neural nets and spectral normalisation. Authors acknowledge that a careful tuning of the neural net’s Lipschitz constant is important and study that (both with or without regularization, and different levels of smoothness) as an ablation of their model. Authors also demonstrate that the model can be applied, although with reduced accuracy, with the neural network being replaced by the identity mapping. I am interested in knowing the importance of some of the other hyper parameters.  For example, how does the dimension of the latent variable h affect the results? Can you do additional ablations where you varying e.g. D_H and M. Some of these parameters can also have an important role in providing accurate predictions and users should be able to set them. Do you suggest a simple cross-validation for calibrating the hyperparameters?  If you increase D_H, does the neural network based embedding become less and less important?  Could you eventually remove the neural net completely (i.e. use identity mapping) by sufficiently increasing the dimension of the output specific latent variables?

**Limitations:**

No specific limitations.

**Strengths And Weaknesses:**

Soundness: The proposed method is relatively straightforward and, in my opinion, a well-constructed method. Given the length of the Appendix, I did not have time to review all details in Appendix, but the parts of the method description related to the proposed method appear correct. (Appendix provides a full description of previous methods as well — I am not sure those are needed or should be repeated here given that they are described in the respective original publications.)

Presentation: The proposed methods are described clearly and with good amount of details.

Significance: Accurate probabilistic modeling of high-dimensional data is central in many applications. This manuscripts proposes a new method for this important task, thus emphasising the significance of the method and manuscript. A challenge is to have probabilistic modeling method that can accurately model correlated output such that the computational method is computationally scalable. This manuscript makes a contribution to specific task

Originality: To my knowledge this manuscript contains original ideas by proposing an MOGP that implements the correlation structures differently compared to earlier works.

---

> ### Author Rebuttal · Authors · 2026-03-30
>
> Thank you very much for your positive evaluation of our paper. Please find responses to your comments below.
>
> ---
> ## 1. Additional ablations with varying $D_H$ and $M$.
> ---
> We conduct additional ablation studies of T-LVMOGP on the SARCOS dataset by varying $D_H$ and $M$, while keeping all other hyperparameters the same as those used in Section 5.2 of the paper. The results are reported in the tables below.
>
> |$D_H$|1|3|5|7|9|
> |-|-|-|-|-|-|
> |MSE|0.024±0.001|0.024±0.000|**0.022±0.000**|**0.022±0.000**|0.023±0.000|
> |NLL|-0.462±0.012|-0.463±0.006|-0.485±0.009|**-0.493±0.009**|-0.481±0.008|
>
> |$M$|100|150|200|250|300|
> |-|-|-|-|-|-|
> |MSE|0.025±0.001|0.024±0.001|0.022±0.000|0.022±0.001|**0.021±0.002**|
> |NLL|-0.429±0.012|-0.459±0.016|-0.485±0.009|-0.496±0.013|**-0.505±0.029**|
>
> In terms of $D_H$, we find that varying $D_H$ has only a limited effect on test predictive performance, with $D_H = 7$ yielding the best overall result. As for $M$, when varying the number of inducing points from 100 to 300, we observe a consistent improvement in predictive performance as $M$ increases, which is in line with our expectations.
>
> ---
> ## 2. Could you eventually remove the neural net (i.e. use identity mapping) by increasing $D_H$?
> ---
> Note that T-LVMOGP uses an ARD-RBF kernel as its base kernel. If the neural network is removed, that is, if the identity mapping is used, the resulting covariance reduces to a separable cross-output structure. By contrast, in the full T-LVMOGP, the neural network induces a generally non-separable cross-output covariance, even when $D_H$ is relatively small. In this sense, the degenerate version of T-LVMOGP recovers the covariance structure of LV-MOGP with $Q=1$. This further suggests that T-LVMOGP with the identity mapping has weaker expressive capacity than the full model with a neural network component, even if $D_H$ is increased.
>
> To verify this empirically, we further evaluate T-LVMOGP with the identity mapping on the SARCOS dataset while varying $D_H$ from 1 to 13. The results are reported in the table below.
>
> |$D_H$|1|3|5|7|9|11|13|
> |-|-|-|-|-|-|-|-|
> |MSE|0.031±0.000|0.030±0.000|0.030±0.000|0.030±0.000|0.030±0.001|0.030±0.000|0.030±0.000|
> |NLL|-0.306±0.007|-0.327±0.009|**-0.336±0.01**|-0.331±0.004|-0.333±0.003|-0.328±0.006|-0.331±0.001
>
> We find that the test MSE is largely insensitive to the choice of $D_H$. The test NLL varies modestly across different values of $D_H$, with the best NLL achieved at $D_H = 5$. Nevertheless, even the best performance under the identity mapping, with MSE $0.030$ and NLL $-0.336$, remains inferior to that obtained with RCNN ($D_H=5$), which achieves MSE $0.022$ and NLL $-0.485$. These results suggest that the contribution of the neural network component cannot be recovered simply by increasing $D_H$.

---

> > ### Author Rebuttal · Reviewer_tyNS · 2026-04-03
> >
> > Thank you for the careful responses and clarifications as well as additional experiments. The relationship between the neural net / identity and latent dim. is now clear. I still think that this work makes a meaningful and important contribution and I will keep my score.

---

> > > ### Author Response · Authors · 2026-04-04
> > >
> > > Thank you for the follow-up and for confirming that your concerns have been addressed. We sincerely thank the reviewer for recognising the significance and contribution of our work. We are also grateful for the thoughtful questions concerning the latent dimensionality $D_H$, the identity mapping, and the number of inducing points $M$, which motivated additional ablation studies and helped us better clarify these aspects of the model. For completeness, we will include these results in the appendix of the revised manuscript.

---

### Official Review · Reviewer_94SE · 2026-03-13

**Soundness:** 3
**Presentation:** 3
**Significance:** 2
**Originality:** 3
**Overall Recommendation:** 4
**Confidence:** 4

**Summary:**

The authors propose a multi-output Gaussian process (MOGP) framework that introduces a deep kernel to model cross-output covariances. A Lipschitz-regularized residual neural network with spectral normalization is used to improve generalization. The MOGP framework is flexible and scalable to high-dim outputs.

**Compliance With Llm Reviewing Policy:**

Affirmed.

**Final Justification:**

This article is well-written and well-structured, and the author addressed my concerns in the rebuttal section. However, I feel the topic is somewhat outdated, and the methodology is, in some sense, incremental. Given that I gave it a score of 4 in the first round of peer review, this will be my final evaluation of this paper.

**Key Questions For Authors:**

- In Fig.1, I think $h_*$ should be revised to $h$, given that $H$ is shared for all input $x$

- Are the parameters in the base kernel also jointly learned through the proposed algorithm? It would be helpful if the authors could present all model & variaional parameters together with the training procedure.

- I am trying to understand the role of $H$. In this framework, as shown in Fig. 1, Fig 2, and also the kernel function in Section 3.1, it seems that we can directly omit $h_p$, and only take $x_n$ as input, and get the embedding output. In this case, the whole framework also seems to work. The benefit is that in this case, you can get rid of the regularization term (KL of $H$) in the objective function and do not need to optimize additional parameters. Can you clarify this point?

**Limitations:**

Yes

**Strengths And Weaknesses:**

The paper is well structured and well-written, and provides strong empirical evidence to show the effectiveness of the proposed method.

However:
- Its contribution is somewhat incremental.
- A key technical concern must be addressed: the complexity analysis is suspect. The claim that the complexity does not depend on $Q$ is counterintuitive and lacks rigorous justification.
- Since the authors claim that "the existing methods do not adequately address scalability for high-dimensional output spaces," I suggest adding a complexity comparison table in the paper (not in the Appendix) to more directly show the contribution of this paper. Please explain why the proposed method is advantageous to those methods that also scale linearly with the output dimension $P$.

---

> ### Author Rebuttal · Authors · 2026-03-29
>
> Thank you for your comments. Here are our responses.
>
> ---
> ## 1. Why does the computational complexity not depend on $Q$?
> ---
> $Q$ denotes the number of coregionalisation matrices in LMC and GS-LVMOGP. In these models, the cross-output covariance is designed as a sum of separable terms, and $Q$ specifies the number of such components. As discussed in Section 3.1, T-LVMOGP does not assume a sum-of-separable structure. Instead, it models cross-output correlations implicitly via a learned embedding space and a base kernel. Consequently, our model does not introduce a counterpart of $Q$, and its complexity does not depend on $Q$.
>
> ---
> ## 2. I suggest adding a table comparing complexity in the paper (not in the appendix).
> ---
> Yes, we agree. We will summarise Appendix I into a table in the updated version.
>
> ---
> ## 3. Explain why the proposed method is advantageous to baseline methods.
> ---
> **Compare to SV-LMC and GS-LVMOGP:**
>
> In SV-LMC, the output functions are generated from some shared latent GPs, inducing a low-rank coregionalisation structure. GS-LVMOGP generalises it by replacing the linear kernel with any kernel function. However, both models inherit a sum-of-separable cross-output covariance structure, which might limit the modelling power.
>
> T-LVMOGP constructs cross-output correlations more flexibly by leveraging a learned embedding space paired with a base kernel. This approach avoids rigid structural assumptions while maintaining computational efficiency. Furthermore, we demonstrate in Appendix D that the kernel of T-LVMOGP generalises that of GS-LVMOGP, underscoring T-LVMOGP's superior capacity.
>
> **Compare to OILMM:**
>
> As in Appendix J.2, OILMM is a special type of LMC with a lower computational cost by exploiting its structure. OILMM relies on restrictive assumptions on basis vectors and observation noise matrix, which may limit its expressiveness. OILMM lacks native support for missing data, and adopting it for such cases requires modifications that undermine its computational benefits. OILMM is strictly tied to a Gaussian likelihood, limiting its applicability to non-Gaussian data.
>
> T-LVMOGP reformulates MOGPs as scalar GPs; it naturally supports missing data by marginalising them out in the ELBO. T-LVMOGP accommodates non-Gaussian likelihood, such as the ZINB likelihood for Spatial Transcripomics.
>
> **Compare to SGPRN:**
>
> Though SGPRN is computationally efficient when the output space has a grid structure by exploiting tensor algebra, its applicability is restricted, as many real-world datasets cannot be naturally represented as multidimensional arrays. Moreover, SGPRN has cubic complexity w.r.t the number of inputs $N$, limiting its scalability to large datasets.
>
> T-LVMOGP does not rely on explicit structural assumptions for cross-output covariances. Instead, it constructs covariances in a more flexible manner via an embedding space and a base kernel. We develop a scalable procedure that supports mini-batch training over both inputs and outputs, allowing T-LVMOGP to scale to datasets with large both $N$ and $P$.
>
> **Compare to G-MOGP:**
>
> G-MOGP employs an attention mechanism to model cross-output correlations, which requires constructing $P \times P$ attention weight matrices. As $P$ grows, this introduces substantial memory and computational overhead. Moreover, G-MOGP uses an independent SVGP for each output, each introducing $M D_X + M + M^2$ variational parameters. When $P$ is large, the total number of variational parameters becomes substantial, which further increases the training cost.
>
> T-LVMOGP defines the variational distribution $q(\mathbf{u})$ over the learned embedding space, shared across all outputs. This shared parameterisation can be viewed as an amortised variational construction, leading to greater parameter efficiency.
>
> ---
> ## 4. In Fig. 1, can we revise $h_*$ to $h$? Explain the role of $H$.
> ---
> No, they are different. We use $h_*$ to denote a latent variable distinct from $h$, corresponding to a different output. $H$ denotes the collection of all $P$ latent variables. Importantly, $H$ is not shared across outputs in the sense of using the same latent vector for every output. For a given output $p$, only its associated latent variable $h_p \in H$ is paired with input $x \in X$. In other words, $h_p$ is the key output-specific component that distinguishes one output from another. We cannot omit $h_p$. Without $h_p$, different outputs would be mapped to indistinguishable embeddings under the shared neural network $\Phi_{\theta}$, and the model would therefore be unable to learn meaningful cross-output correlations.
>
> ---
> ## 5. Are the parameters in the base kernel jointly learned?
> ---
> Yes. All model parameters are jointly learned via gradient-based optimisation, including the base kernel, the likelihood function, the inducing points, the variational distribution over inducing variables, the neural network parameters, and the variational distribution over the latent variables.

---

> > ### Author Rebuttal · Reviewer_94SE · 2026-04-01
> >
> > The reviewer thanks the authors for their response.
> >
> > One follow-up question: how do you choose the dimension of h_p? I was wondering if different dimensions would affect the performance? Any comments or results about this?
> >
> > I will keep my score.

---

> > > ### Author Response · Authors · 2026-04-01
> > >
> > > We thank the reviewer for carefully reviewing our rebuttal and for taking the time to engage further with our response. We are also pleased that some of the concerns raised in the original review have been resolved. Here is our response to your follow-up question.
> > >
> > > ---
> > > In T-LVMOGP, we treat the latent variable dimension $D_H$ as a small hyperparameter that controls the complexity of the output-specific representation. In practice, we set $D_H$ to a low value (typically $D_H \leq 5$ in our experiments) to keep the latent representation compact and reduce the number of learnable parameters.
> > >
> > > Empirically, we did not observe strong sensitivity to the choice of $D_H$ within a reasonable range. To examine this directly, we conducted an ablation study on the SARCOS dataset, varying $D_H$ from 1 to 9 (also see our reply to Reviewer tyNS):
> > >
> > > |$D_H$|1|3|5|7|9|
> > > |-|-|-|-|-|-|
> > > |MSE|0.024 ± 0.001|0.024 ± 0.000|0.022 ± 0.000|0.022 ± 0.000|0.023 ± 0.000|
> > > |NLL|-0.462 ± 0.012|-0.463 ± 0.006|-0.485 ± 0.009|-0.493 ± 0.009|-0.481 ± 0.008|
> > >
> > > The results show that changing $D_H$ only has a limited effect on test predictive performance. This suggests that the model is relatively robust to the precise choice of latent dimensionality.
> > >
> > > Overall, our current evidence suggests that using a low-dimensional latent variable is sufficient in practice, and increasing $D_H$ does not by itself lead to substantial performance gains.

---

### Official Review · Reviewer_oD9b · 2026-03-16

**Soundness:** 3
**Presentation:** 3
**Significance:** 3
**Originality:** 3
**Overall Recommendation:** 5
**Confidence:** 4

**Summary:**

The paper proposes Transformed Latent Variable Multi-Output Gaussian Processes (T-LVMOGP), a scalable framework for modelling large numbers of correlated outputs with GPs. The key challenge in multi-output GPs is specifying the cross-output covariance function. Inspired from deep kernel learning they propose, applying a shared base kernel on this idea of a learned embedding. Each output is associated with a latent vector, which is concatenated with the input and passed through a residual neural network to produce an embedding. The framework contends that the output correlations are implictly captured through the learned representation rather than a manually designed multi-output kernel. The stochastic variational GP machinery is used for scalable inference, they derive an ELBO compatible with their framework and propagate gradients towards learning all unknows (latents per output, residual network params, kernel hypers, variational params) in one joint step. Empirically, the framework is evaluated on oft used datasets in multi-output GP literature and compared across baselines.

**Compliance With Llm Reviewing Policy:**

Affirmed.

**Final Justification:**

The rebuttal provided a clear example and explanation of how their framework differs and improves over the canonical linear model of coregionalisation, the de-facto approach to multi-output GPs.

I've chosen to update my score 4 -> 5.

**Key Questions For Authors:**

1. For every pair $\tilde{x}\_{n,p} = \Phi_{\theta}( x_{n}, h_{p} )$, the same network is used across all outputs? so, the weights are shared, the architecture is shared and the base kernel is shared, the only thing that differs is the latent vector $h_{p}$?
2. The cross-output correlation appears because the embeddings differ driven by the latent $h_{p}$?

**Limitations:**

Yes, limitations have been discussed.

**Strengths And Weaknesses:**

Strengths:

- Well-written and clearly presented derivations and baselines.
- The framework is conceptually simple and modular: it combines latent output embeddings, deep kernel learning, and sparse variational GPs in a unified architecture. This design allows the method to leverage existing GP inference machinery while extending it to high-dimensional multi-output settings.
- The framework offers a flexible alternative to traditional multi-output GP kernels, which often rely on restrictive structures such as low-rank coregionalization or separable kernels.

Weaknesses:

- The experimental evaluation primarily focuses on standard benchmark datasets commonly used in the multi-output GP literature perhaps to aid cross-comparisons.  However, they do not clearly demonstrate scenarios where the proposed architecture of a deep-kernel like framework within the multi-ouput setting provides a decisive advantage.
- The. mathematical crux of the paper is having this shared base kernel operate on learned embeddings, but there is not much beyond metrics on how this approach is captures cross-output correlations better than say LMC. I think the core idea needs to be demonstrated (perhaps on a synthetic dataset) to drive home the fundamental point that the cross-output dependendies are captured in a more superior way under this modelling framework.

---

> ### Author Rebuttal · Authors · 2026-03-28
>
> We sincerely appreciate your detailed and positive assessment of our work. We provide detailed responses to your questions below.
>
> ---
> ## 1. T-LVMOGP better captures cross-output correlations than LMC.
> ---
> As described in Section 3.1, T-LVMOGP models cross-output dependencies implicitly through a learned embedding space. This construction allows each input-output pair to interact flexibly during training, without imposing rigid assumptions on the cross-output covariance. In contrast, LMC assumes a sum-of-separable covariance structure by design, which can be restrictive when the true dependence pattern is not well approximated by a sum of separable components.
>
> To illustrate this difference, we consider a simple synthetic two-output regression problem. The outputs are defined as $y_1(x) = \sin(x) + \epsilon_1, y_2(x) = \sin(\pi + x) + \epsilon_2,$
> where $\epsilon_1, \epsilon_2 \sim \mathcal{N}(0, 0.01)$ and $x \in [0, 2\pi]$. The second output is therefore a phase-shifted version of the first. In particular, $y_1(x) \approx y_2(x+\pi), \: x \in [0,\pi],$ and $y_1(x) \approx y_2(x-\pi), \: x \in [\pi, 2\pi].$ By contrast, for $x \in (0,\pi) \cup (\pi,2\pi)$, we generally do not have $y_1(x) \approx y_2(x)$.
>
> An appropriate multi-output model should therefore learn:
> * (1) relatively large cross-covariances between $y_1(x)$ and $y_2(x+\pi)$ for $x \in [0,\pi]$, and between $y_1(x)$ and $y_2(x-\pi)$ for $x \in [\pi,2\pi]$;
> * (2) relatively small cross-covariances between $y_1(x)$ and $y_2(x)$ over $x \in (0,\pi) \cup (\pi,2\pi)$.
>
> We refer to the first case as **value-matched** and the second as **input-matched**. Under the ground-truth correlation structure, a desirable model should assign higher covariance to value-matched pairs than to input-matched pairs.
>
> We uniformly sample $64$ inputs from $[0,2\pi]$ and fit both T-LVMOGP and LMC to this dataset, using $M=5$ inducing points in all cases. After training, we compute the average normalised covariance magnitudes (over 10 random trails):
>
> |Setting|T-LVMOGP|LMC $(Q=1)$|LMC ($Q=2$)|LMC ($Q=3$)|
> |-|-|-|-|-|
> |Value-matched|**0.92 ± 0.03**|0.48 ± 0.02|0.40 ± 0.15|0.36 ± 0.16|
> |Input-matched|0.72 ± 0.11|**0.99 ± 0.01**|**0.82 ± 0.26**|**0.74 ± 0.31**|
>
> The results show that T-LVMOGP assigns relatively larger covariance to value-matched pairs than to input-matched pairs, which is consistent with the true cross-output dependency. In contrast, for all LMC variants ($Q=1,2,3$), the input-matched covariances remain larger than the value-matched ones. This suggests that the sum-of-separable structure imposed by LMC can hinder its ability to represent this phase-shifted correlation pattern, whereas T-LVMOGP can capture it more naturally through its learned embeddings and the base kernel.
>
> ---
>
> ## 2. Is the same network used across all outputs? The only thing that differs across outputs is the latent vector $h_p$?
>
> ---
> Yes. T-LVMOGP uses a **single shared neural network** parameterised by $\theta$ for all outputs. Specifically, the network takes the pair $(x_n, h_p)$ as input and maps it to an embedding vector $\tilde{x}_{n,p}$, which is then passed to the base kernel. In this way, the embedding jointly incorporates both input information and output-specific information, as illustrated in Figs. 1 and 2.
>
> Yes. The only output-specific component is the latent vector $h_p$, which allows different outputs to be represented differently within the common embedding space.
>
> ---
> ## 3. The cross-output correlation appears because the embeddings differ driven by the latent $h_p$?
> ---
> Yes. Different outputs are associated with different latent vectors $h_p$. Although the mapping $\Phi_\theta$ is shared across all outputs, different values of $h_p$ lead to different embeddings. These output-dependent embeddings are then passed to the base kernel, through which the model induces cross-output correlations.

---

> > ### Author Rebuttal · Reviewer_oD9b · 2026-04-04
> >
> > Thank you for the response, the example of how T-LVMOGP better captures cross-output correlations than LMC is an important crux that should be included in the revised paper.
> >
> > I am happy to raise my score to 5.

---

> > > ### Author Response · Authors · 2026-04-04
> > >
> > > We sincerely thank you for taking the time to read our rebuttal carefully. We greatly appreciate your positive evaluation and your decision to raise the score.
> > >
> > > We are also especially grateful for your insightful comment regarding how T-LVMOGP compares with LMC in capturing cross-output correlations. We agree that this point is central to the paper, and your feedback helped us better identify this important aspect. In the revised version of the paper, we will incorporate this synthetic experiment and the corresponding discussion, so that the advantage of T-LVMOGP in modelling more flexible cross-output dependency structures is better conveyed in the main manuscript.

---

### Decision · Program_Chairs · 2026-04-30

**Decision:**

Accept (regular)

**Comment:**

Multi-Output Gaussian Processes (MOGPs) are probabilistic models that provide a principled framework for modelling correlated outputs. To maintain tractability, current methods resort to restrictive assumptions, such as low-rank or sum-of-separable kernels, which can limit expressiveness. This paper proposes the Transformed Latent Variable MOGP, a novel framework that scales MOGPs to a massive number of outputs. The proposed method constructs a flexible multi-output deep kernel by mapping inputs and output-specific latent variables into an embedding space using a Lipschitz-regularized neural network. Combined with stochastic variational inference, such a model effectively scales to high-dimensional output settings. Across diverse benchmarks, the proposed model outperforms baselines in both predictive accuracy and computational efficiency. The reviewers have indicated several strong points of this submission such as well-written and clearly presented derivations and baselines; a conceptually simple and modular framework; it provides strong empirical evidence to show the effectiveness of the proposed method; the proposed methods are described clearly and with good amount of details; significance, as accurate probabilistic modeling of high-dimensional data is central in many applications; and originality and with theoretical rigor. Some issues were raised during their review. However, the authors have correctly addressed them during the rebuttal phase.  An explanation of how the proposed framework differs and improves over the canonical linear model of corgionalization, the de facto approach to multi-output GPs, was given.  Some reviewers feel that the topic is somewhat outdated, and the methodology is, in some sense, incremental. However, I believe this will be a nice contribution that will receive the attention of the community working on multi-output GPs.